# Allosteric modulation of the CXCR4:CXCL12 axis by targeting receptor nanoclustering via the TMV-TMVI domain

Eva M García-Cuesta[1], Pablo Martínez[1], Karthik Selvaraju[2], Gabriel Ulltjärn[2], Adrián Miguel Gómez Pozo[3], Gianluca D'Agostino[1], Sofia Gardeta[1], Adriana Quijada-Freire[1], Patricia Blanco Gabella[3], Carlos Roca[3], Daniel del Hoyo[4], Rodrigo Jiménez-Saiz[1,5,6,7], Alfonso García-Rubia[3], Blanca Soler Palacios[1], Pilar Lucas[1], Rosa Ayala-Bueno[1], Noelia Santander Acerete[1], Yolanda Carrasco[8], Carlos Oscar Sorzano[4], Ana Martinez[3,9], Nuria E Campillo[3], Lasse D Jensen[2], Jose Miguel Rodriguez Frade[1], César Santiago[10], Mario Mellado[1]*

[1]Chemokine Signaling group, Department of Immunology and Oncology, Centro Nacional de Biotecnología/CSIC, Campus de Cantoblanco, Madrid, Spain; [2]Division of Diagnostics and Specialist Medicine, Department of Health, Medical and Caring Sciences, Linköping University, Linköping, Sweden; [3]Centro de Investigaciones Biológicas Margarita Salas (CIB-CSIC), Madrid, Spain; [4]Biocomputing Unit, Centro Nacional de Biotecnología (CNB-CSIC), Campus de Cantoblanco, Madrid, Spain; [5]Department of Immunology, Instituto de Investigación Sanitaria Hospital Universitario de La Princesa (IIS-Princesa), Madrid, Spain; [6]Department of Medicine, McMaster Immunology Research Centre (MIRC), Schroeder Allergy and Immunology Research Institute, McMaster University, Hamilton, Canada; [7]Faculty of Experimental Sciences, Universidad Francisco de Vitoria (UFV), Madrid, Spain; [8]B Lymphocyte Dynamics, Department of Immunology and Oncology, Centro Nacional de Biotecnología (CNB)/CSIC, Campus de Cantoblanco, Madrid, Spain; [9]Neurodegenerative Diseases Biomedical Research Network Center (CIBERNED), Instituto de Salud Carlos III, Madrid, Spain; [10]X-ray Crystallography Unit, Department of Macromolecules Structure, Centro Nacional de Biotecnología/CSIC, Campus de Cantoblanco, Madrid, Spain

*For correspondence:
mmellado@cnb.csic.es

**Abstract** CXCR4 is a ubiquitously expressed chemokine receptor that regulates leukocyte trafficking and arrest in both homeostatic and pathological states. It also participates in organogenesis, HIV-1 infection, and tumor development. Despite the potential therapeutic benefit of CXCR4 antagonists, only one, plerixafor (AMD3100), which blocks the ligand-binding site, has reached the clinic. Recent advances in imaging and biophysical techniques have provided a richer understanding of the membrane organization and dynamics of this receptor. Activation of CXCR4 by CXCL12 reduces the number of CXCR4 monomers/dimers at the cell membrane and increases the formation of large nanoclusters, which are largely immobile and are required for correct cell orientation to chemoattractant gradients. Mechanistically, CXCR4 activation involves a structural motif defined by residues in TMV and TMVI. Using this structural motif as a template, we performed in silico molecular modeling followed by in vitro screening of a small compound library to identify negative allosteric modulators of CXCR4 that do not affect CXCL12 binding. We identified AGR1.137, a small molecule that abolishes CXCL12-mediated receptor nanoclustering and dynamics and blocks the ability

of cells to sense CXCL12 gradients both in vitro and in vivo while preserving ligand binding and receptor internalization.

## eLife assessment

This is an **important** study that describes an elegant modelling driven approach to design of allosteric antagonists for CXCR4 that have a selective effect on receptor nanocluster formation, cell polarisation and chemotaxis, but spare binding of CXCL12 to the receptor and inhibition of adenylate cyclase. This enables selective targeting of processes dependent upon cell polarisation and chemotaxis without impacting signalling effects and may avoid some of the toxicity associated with antagonists that target CXCL12 binding and thus block all CXCR4 signalling. The revised manuscript offers **convincing** evidence to support the claims. The modelling work is better described and additional data has been presented that better illustrates the unique features of the new antagonist. The in vivo studies in the zebrafish model open a path to studies in mammalian models.

## Introduction

CXC chemokine receptor 4 (CXCR4) is a homeostatic G-protein-coupled receptor (GPCR) that is widely expressed in both embryonic and adult tissues (*Zlotnik and Yoshie, 2012*). It is also ubiquitously expressed in the hematopoietic system, where it plays a pivotal role in leukocyte trafficking and arrest in specific niches in homeostasis and disease. CXCR4 is essential for adaptive and innate immune responses, as well as in the organization and maintenance of the bone marrow (BM; *Nie et al., 2008*). Indeed, CXCR4 and its unique chemokine ligand CXC motif chemokine 12 (CXCL12) are largely responsible for the migration (*Wright et al., 2002*), homing (*Lapidot and Kollet, 2002*), and survival (*Broxmeyer et al., 2003*) of hematopoietic stem cells in the BM.

Results from studies in conditional *Cxcr4*-knockout mice point to a relevant role for CXCR4 in several non-hematopoietic tissues, for example, in regulating central nervous system development (*Cash-Padgett et al., 2016*), and in vasculature development in the gastrointestinal tract (*Tachibana et al., 1998*) and the kidney (*Takabatake et al., 2009*). In addition, CXCR4, together with CCR5, serve as primary co-receptors (with CD4) for HIV-1 fusion and entry into target cells (*Berger et al., 1999*).

CXCR4 expression is frequently elevated in many cancers, including breast (*Müller et al., 2001*), ovarian (*Hall and Korach, 2003*), prostate (*Taichman et al., 2002*), melanoma (*Scala et al., 2006*), and neuroblastoma (*Geminder et al., 2001*), where it participates in tumor growth, tumor cell interactions with the microenvironment (*Orimo et al., 2005*), vasculogenesis, angiogenesis (*Liang et al., 2007*), and metastasis (*Müller et al., 2001*). Indeed, increased CXCR4 expression in metastatic lesions correlates with tumor progression and with preferential metastatic sites of the primary tumor (*Bohn et al., 2009*). Studies in mice have suggested that CXCR4 is a good target for cancer therapy, as blockade of its signaling impairs metastasis in several models (*Müller et al., 2001*; *Kim et al., 2008*). Despite the essential role of the CXCR4/CXCL12 axis in physiology and pathology, the only commercially available CXCR4 antagonist approved for clinical use is plerixafor (AMD3100), which is indicated for the mobilization of stem cells from the BM to the peripheral blood in autologous transplantation (*De Clercq, 2019*).

Single particle tracking in total internal reflection fluorescence (SPT-TIRF) experiments and stimulated emission depletion super resolution microscopy (STED) have demonstrated that CXCR4 is organized at the cell membrane as monomers, dimers and small aggregates (groups of ≥3 receptors) termed nanoclusters, and that CXCL12 binding decreases the percentage of monomers/dimers and increases the formation of large nanoclusters (*Martínez-Muñoz et al., 2018*). This mechanism is essential for CXCR4 signaling, which allows cells to orient themselves correctly in response to CXCL12 gradients (*García-Cuesta et al., 2022*). A triple-mutant CXCR4 (K239L/V242A/L246A; CXCR4mut), located in the N-terminal region of transmembrane (TM) VI, dimerizes but neither forms nanoclusters in response to CXCL12 nor supports CXCL12-induced directed cell migration, although it can induce some $Ca^{2+}$ flux and is internalized after ligand binding (*Martínez-Muñoz et al., 2018*). A cluster has been identified in CXCR4, defined by TMV and TMVI residues that form a hydrophobic lock connecting the signal from the orthosteric ligand-binding site to the downstream conserved domains involved in G-protein coupling and signaling (*Wescott et al., 2016*; *Zhou et al., 2019*).

Here, we used this signal domain of CXCR4 as a base to search for new CXCR4 antagonists. We performed an in silico screening of a small aromatic compound library for the identification of allosteric modulators that block directed cell migration, without interfering with ligand binding and, thus, allowing other CXCL12-mediated functions. Using molecular modeling analyses, we identified a novel class of structurally related small molecules that modulate some CXCR4 functions by binding to a novel regulatory cleft formed by the TMV and TMVI transmembrane helices. One of the selected compounds, AGR1.137, abolished CXCL12-mediated receptor nanoclustering and dynamics and the ability of the cells to specifically sense CXCL12 gradients, but did not alter ligand binding or receptor internalization. Notably, AGR1.137 had a minimal effect on ERK1/2 and AKT phosphorylation, which may reduce the side effects associated with full CXCR4 inhibition. Finally, using a zebrafish tumor model, we observed that AGR1.137 treatment of HeLa cells reduced tumorigenesis and metastasis, suggesting that AGR1.137 also impairs cell sensitivity to CXCL12 gradients by blocking CXCR4 nanoclustering in vivo. In conclusion, our results establish the importance of residues within the TMV–VI cleft for CXCR4 nanoclustering and CXCL12-mediated directional migration, and identify a negative allosteric modulator that exhibits both in vitro and in vivo activity.

## Results

### Screening for small compounds targeting CXCR4 that block CXCL12-induced CXCR4 nanoclustering

The propagation domain of CXCR4, which links ligand binding to the residues involved in G-protein association, includes residues bounded by the transmembrane helices TMV and TMVI, which enclose a cleft of 870 $Å^3$ (*Wescott et al., 2016*) is surface-exposed for interactions with the plasma membrane, as it contains residues involved in lipid binding (*Di Marino et al., 2023*), and also includes residues implicated in receptor activation and in the transmission of conformational changes through the TM helix domains (*Wescott et al., 2016*; *Figure 1A*). Additionally, the cleft contains the residues K239, V242, and L246, the mutation of which prevents CXCL12-mediated receptor oligomerization (*Martínez-Muñoz et al., 2018*). We thus screened for allosteric modulators that fit into this cleft and the cavity formed by TMV and TMVI but do not interfere with the CXCL12 binding site. Virtual screening was performed using the Medicinal and Biological Chemistry (MBC) library (CIB-CSIC) (*Sebastián-Pérez et al., 2017*) comprises more than 2000 small heterocyclic compounds with drug-like properties. As a template for modeling CXCR4 we used the atomic structure of the receptor solved in complex with the inhibitor IT1t (PDB code: 3ODU). As this included T4 lysozyme inserted between helices TMV and TMVI to increase the stability of the protein (*Figure 1A*), a common strategy to facilitate crystallogenesis of GPCRs (*Zou et al., 2012*), we first generated (*Di Marino et al., 2023*; *Wu et al., 2010*) a CXCR4 homology model using the SWISS_MODEL server (*Waterhouse et al., 2018*). The compounds were then prepared and docked against the modeled CXCR4 structure using 'Glide', and those identified were ranked by docking score, that is, by the minimal interaction energy given by Glide and visualization of the poses. The results were confirmed using a second strategy based on the CXCR4 model predicted by AlphaFold (*Jumper et al., 2021*) and the sequence available under UniProt entry P61073.

The top 40 candidates with the best docking score in the region of interest were chosen and were selected for functional analysis (*Figure 1—source data 1*). We first screened their ability to block CXCL12-induced cell dependent migration in Transwell chambers.

Jurkat cells that were either untreated or treated with different concentrations (1–100 µM) of the selected small compounds (30 min, 37 °C, 5% $CO_2$) were allowed to migrate towards CXCL12. To enhance their possible inhibitory effects, the compounds were maintained in the upper chamber containing the cells throughout the chemotaxis experiment *Figure 1—figure supplement 1*.

Based on these results, we selected two compounds, AGR1.135 and AGR1.137, which showed reproducible dose-dependent inhibitory effects on CXCR4-induced cell migration (*Figure 1B*) and had the highest docking score on the cleft between TMV and TMVI (–51.4 and –37.2 kcal /mol, respectively) and had the highest docking score on the cleft between TMV and TMVI (–51.4 and –37.2 kcal /mol, respectively). As a control for further analysis, we used AGR1.131, which shares a common scaffold with the other compounds and interacts with TM domains of CXCR4, but did not affect CXCL12-promoted cell migration. It also theoretically targeted the same motif on CXCR4 as AGR1.135 and AGR1.137 (-39.8 kcal/mol), but showed a better docking score on a pocket between TMI and TMVII (–43.6

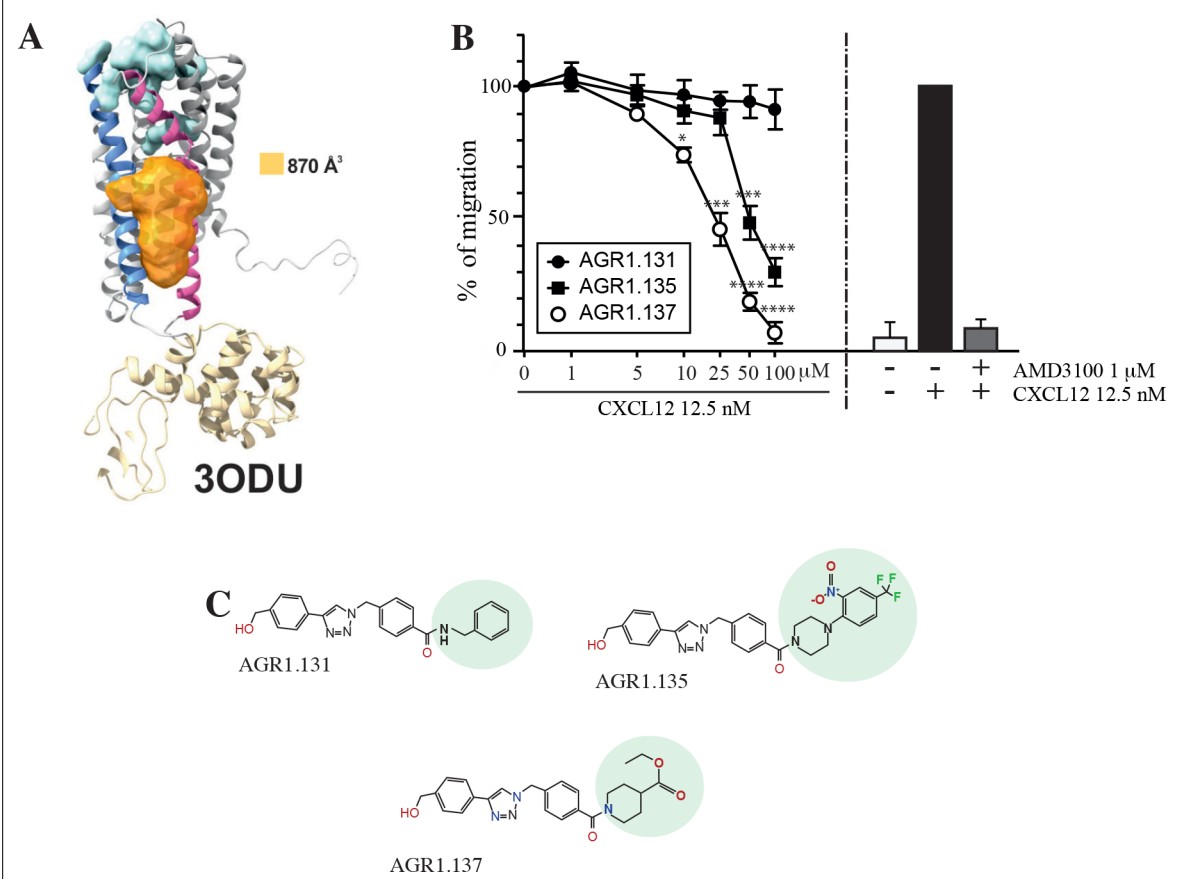

**Figure 1.** Screening for small compound antagonists acting at the oligomerization site of CXCR4. (**A**) Cartoon and surface representation of CXCR4 and the cleft identified between TMV and TMVI. The protein structure is shown in gray, with TMV and TMVI colored in blue and pink, respectively. In green, residues involved in CXCL12 binding. The cavity identified by SurfNet is shown in orange. Volume for the cavity is measured and shown in $Å^3$. T4 lysozyme inserted between TMV and TMVI in the crystallized version of CXCR4 (PDB: 3ODU) is also shown in yellow. (**B**) Dose-response curve of the selected antagonists in Jurkat cell migration experiments in response to 12.5 nM CXCL12. Data are shown as percentage of migrating cells (mean ± SD; n=5; *≤0.05, *** p≤0.001, **** p≤0.0001). (**C**) Chemical structure of the selected compounds (AGR1.131, AGR1.135 and AGR1.137). The differences between the lateral chains of the three compounds are shaded in green.

The online version of this article includes the following source data and figure supplement(s) for figure 1:

**Source data 1.** Compounds with minimal interaction energy in the area of interest.

**Source data 2.** Compound synthesis and characterization.

**Figure supplement 1.** Small compound-mediated inhibition of CXCL12-induced cell migration.

**Figure supplement 2.** Effect of small compounds on Jurkat cell cycle.

---

kcal/mol). Under these experimental conditions, none of the selected compounds were toxic against Jurkat cells, as shown by propidium iodide incorporation and cell cycle analysis (*Figure 1—figure supplement 2*). The structures of the three selected compounds was verified by NMR analysis (*Figure 1—source data 2*) and were characterized by the presence of a common core (4-(1-benzyl-1H-1,2,3-triazol-4-yl)phenyl) methanol, linked through a benzamide group to a lateral chain bearing amines of different length and chemical nature. AGR1.131 bears a simple benzylamine, AGR1.135 contains a complex lateral chain due to the presence of the 1-(2-nitro-4-(trifluoromethyl)phenyl)piperazine, and AGR1.137 incorporates ethyl piperidine-4-carboxylate (*Figure 1C*).

## AGR1.135 and AGR1.137 block CXCL12-mediated CXCR4 nanoclustering and dynamics

We next evaluated whether the selected compounds altered CXCL12-mediated CXCR4 nanoclustering and dynamics. To do this, we utilized SPT-TIRF on CXCR4-deficient Jurkat cells (JK[-/-]) transiently

transfected with CXCR4 fused to the AcGFP monomeric protein (CXCR4-AcGFP) and pretreated or not with 50 µM of the selected small compounds (*Figure 2—videos 1–8*), a concentration that inhibited CXCL12-induced cell migration by 50–75% (*Figure 1B*). The compounds had no effect on DMSO-stimulated cells, and we observed mainly receptor monomers and dimers (~82–98%), with a very low percentage of complexes with >3 receptors (~2–18%; *Figure 2A*). Accordingly, the basal mean spot intensity (MSI) of CXCR4 was similar in all cases (~1500 a.u.; *Figure 2B*). Under these experimental conditions, most of the particles corresponded to mobile particles (~88%; *Figure 2C*), and none of the selected compounds affected the short time-lag diffusion coefficient ($D_{1-4}$) for CXCR4 trajectories, with a median value of ~0.02 µm$^2$/s (*Figure 2D*).

We next examined how pre-treatment with the selected compounds affected receptor dynamics upon CXCL12 stimulation. Pre-treatment with AGR1.135 and AGR1.137, but not with AGR1.131, significantly altered CXCL12-mediated receptor nanoclustering (32–69% of nanoclusters with ≥3 receptors/particle in untreated and in AGR1.131-treated cells *versus* 7–22% in cells pretreated with AGR1.135 or AGR1.137; *Figure 2A*), with an MSI of 3219 a.u. for CXCR4 in untreated cells and 3,618 a.u. in AGR1.131-treated cells *versus* 1611 a.u. in AGR1.135- and 1403 a.u. in AGR1.137-treated cells (*Figure 2B*). This effect was also evident when we evaluated the dynamic parameters of CXCR4. Under control conditions (untreated or AGR1.131-treated cells), CXCL12 significantly increased the percentage of immobile particles (12% in untreated cells *versus* 25% in untreated cells +CXCL12 and 18% in AGR1.131-treated cells +CXCL12) (*Figure 2C*), whereas AGR1.135 and AGR1.137 pretreatment had no effect on the percentage of immobile particles after CXCL12 stimulation (14% for both AGR1.135 and AGR1.137). Moreover, the expected reduction in CXCR4 diffusivity triggered by CXCL12 in untreated (median $D_{1-4}$ = 0.0088 µm$^2$/s) or AGR1.131-treated (median $D_{1-4}$ = 0.0099 µm$^2$/s) cells was abolished with AGR1.135 ($D_{1-4}$ = 0.0169 µm$^2$/s) and AGR1.137 ($D_{1-4}$ = 0.0165 µm$^2$/s) treatments (*Figure 2D*).

Altogether, these results suggest that AGR1.135 and AGR1.137 behave as negative allosteric modulators of CXCR4 and alter the CXCL12-mediated receptor nanoclustering and dynamics.

## AGR1.135 and AGR1.137 incompletely abolish CXCR4-mediated responses in Jurkat cells

CXCR4 internalizes in response to CXCL12, a process that mediates receptor desensitization (*Haribabu et al., 1997*). This process is associated with clathrin-mediated endocytosis of chemokine receptors (*Venkatesan et al., 2003*), and requires receptor aggregation. Our previous results indicate that CXCR4 nanoclustering and the aggregates involved in receptor internalization are independent processes (*Martínez-Muñoz et al., 2018*). Indeed, CXCL12 triggers CXCR4mut internalization but fails to promote receptor nanoclustering (*Martínez-Muñoz et al., 2018*). We thus tested whether the selected antagonists affected CXCL12-mediated CXCR4 internalization. Jurkat cells treated with vehicle or the selected compounds were stimulated with CXCL12 and receptor internalization was evaluated by flow cytometry using anti-CXCR4 antibodies. Neither AGR1.135 nor AGR1.137 treatment altered the internalization of CXCR4 that was observed in untreated Jurkat cells or in AGR1.131-treated cells (*Figure 3A*). These data suggest that none of the allosteric modulators block CXCL12 binding to CXCR4, supporting the in silico screening strategy of preserving ligand binding integrity. Indeed, using flow cytometry analysis and CXCL12-ATTO 700 (*Ameti et al., 2018*) we confirmed that none of the selected compounds blocked ligand binding, whereas this was blocked by the ligand-binding inhibitor AMD3100 (*Figure 3B*, *Figure 3—figure supplement 1*). Taken together, the data indicate that the selected compounds do not abrogate ligand binding or receptor internalization, but blocked other functions such as cell migration.

Because CXCL12 promotes Gi-protein activation, we analyzed the effects of the selected compounds on CXCL12-mediated inhibition of cAMP production, a canonical signaling pathway downstream of CXCR4 activation. In contrast to AMD3100, an orthosteric antagonist of CXCR4, none of the compounds affected CXCL12-mediated Gi protein activation (*Figure 3C*). We next evaluated their effects on other signaling pathways such as ERK1/2 phosphorylation and PI3K activation (*Ganju et al., 1998*). Jurkat cells treated with vehicle or the selected compounds were activated with CXCL12 (50 nM) for different time periods and cell lysates were analyzed by western blotting using anti-P-ERK1/2 and -P-AKT antibodies. Neither AGR1.135 nor AGR1.137 nor the control AGR1.131 treatment inhibited CXCL12-mediated activation of the two signaling pathways (*Figure 3D–F*, *Figure 3—figure*

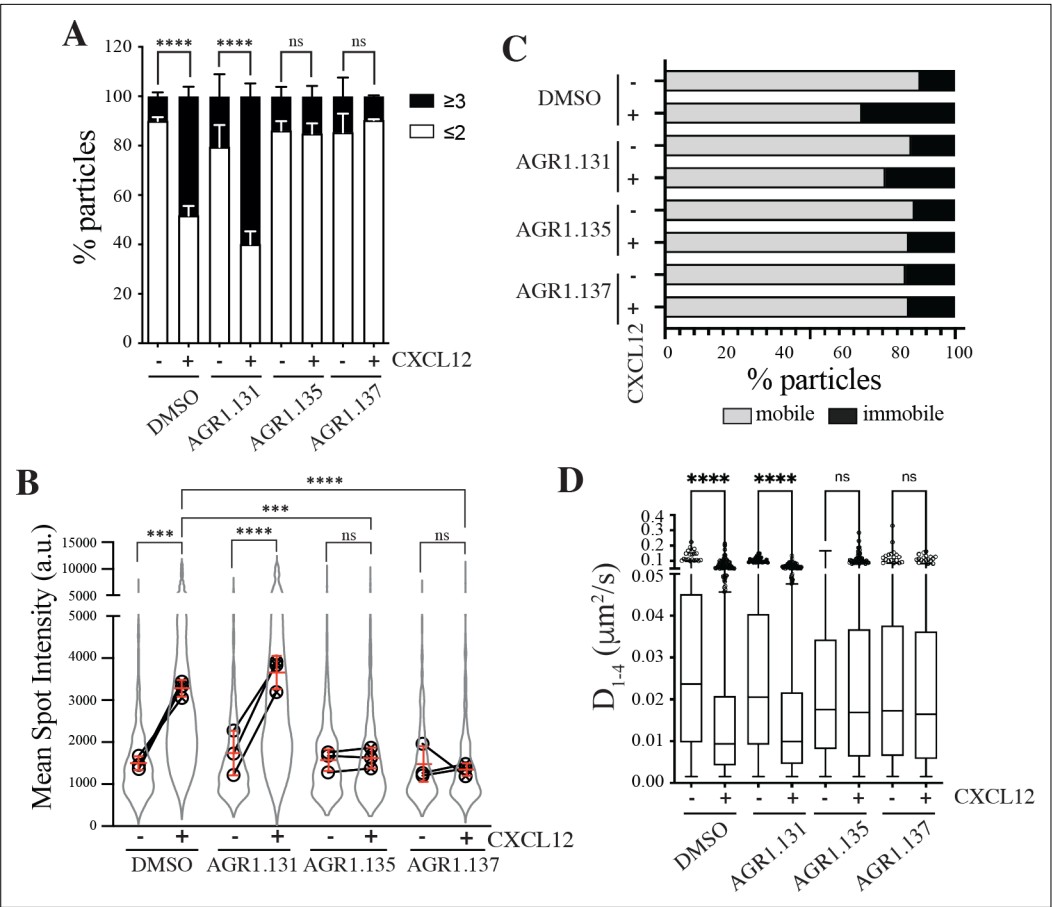

**Figure 2.** AGR1.135 and AGR1.137 alter CXCL12-mediated CXCR4 dynamics and nanoclustering. Single-particle tracking analysis of JK$^{-/-}$ cells transiently transfected with CXCR4 (JK$^{-/-}$-X4) treated with DMSO (control), AGR1.131, AGR1.135 or AGR1.137 on fibronectin (FN)- or FN +CXCL12-coated coverslips (DMSO: 581 particles in 59 cells on FN; 1365 in 63 cells on FN +CXCL12; AGR1.131: 1019 particles in 71 cells on FN; 1291 in 69 cells on FN +CXCL12; AGR1.135: 862 particles in 70 cells on FN; 1003 in 77 cells on FN +CXCL12; AGR1.137: 477 particles in 66 cells on FN; 566 in 64 cells on FN +CXCL12) n=3. (**A**) Frequency of CXCR4-AcGFP particles containing monomers plus dimers (≤2) or nanoclusters (≥3),± SEM, calculated from mean spot intensity values of each particle as compared with the value of monomeric CD86-AcGFP (n=3; n.s., not significant; ****p≤0.0001). (**B**) Intensity distribution (arbitrary units, a.u.) from individual CXCR4-AcGFP trajectories on unstimulated and CXCL12-stimulated JK$^{-/-}$-X4 cells pretreated or not with the indicated compounds or vehicle (DMSO). Graph shows the distribution of all trajectories (gray outline), with the mean value of each experiment (black circles) and the mean of all experiments ± SD (red lines) (n=3; n.s., not significant; ***p≤0.001, ****p≤0.0001). (**C**) Percentage of mobile and immobile CXCR4-AcGFP particles at the membrane of cells treated as indicated. (**D**) Diffusion coefficients ($D_{1-4}$) of mobile single particle trajectories at the membrane of cells treated as indicated, represented by box-and-whiskers plots (Tukey method) with median (black line) indicated. (n.s., not significant, ****p≤0.0001).

The online version of this article includes the following video(s) for figure 2:

**Figure 2—video 1.** Related to *Figure 2*.
https://elifesciences.org/articles/93968/figures#fig2video1

**Figure 2—video 2.** Related to *Figure 2*.
https://elifesciences.org/articles/93968/figures#fig2video2

**Figure 2—video 3.** Related to *Figure 2*.
https://elifesciences.org/articles/93968/figures#fig2video3

**Figure 2—video 4.** Related to *Figure 2*.
https://elifesciences.org/articles/93968/figures#fig2video4

**Figure 2—video 5.** Related to *Figure 2*.
https://elifesciences.org/articles/93968/figures#fig2video5

*Figure 2 continued*

**Figure 2—video 6.** Related to *Figure 2*.
https://elifesciences.org/articles/93968/figures#fig2video6

**Figure 2—video 7.** Related to *Figure 2*.
https://elifesciences.org/articles/93968/figures#fig2video7

**Figure 2—video 8.** Related to *Figure 2*.
https://elifesciences.org/articles/93968/figures#fig2video8

---

*supplement 2*), whereas AMD3100 treatment as a positive control did so (*Figure 3—figure supplement 2*). Overall, the data indicate that the selected compounds inhibit CXCL12-induced chemotaxis in Transwell assays, but do not affect G protein activation or other ligand-mediated signaling pathways such as ERK1/2 or PI3K.

Alterations in actin cytoskeletal dynamics are linked to deficiencies in ligand-mediated receptor nanoclustering and to defects in the ability of cells to sense chemoattractant gradients. Cells treated with latrunculin A, an actin polymerization inhibitor (*Martínez-Muñoz et al., 2018*), and cells expressing CXCR4WHIM$^{R334X}$, which fail to properly control actin dynamics (*García-Cuesta et al., 2022*; *Gardeta et al., 2022*), have defective CXCL12-mediated nanoclustering and are unable to properly sense chemoattractant gradients. Using flow cytometry and phalloidin staining of Jurkat cells treated with vehicle or the selected compounds, we detected altered actin polymerization in cells treated with AGR1.135, with AGR1.137, and with AMD3100 (*Figure 4A*). These data were corroborated by immunostaining and confocal analysis (*Figure 4B and C*). Whereas Jurkat cells treated with vehicle or AGR1.131 were correctly polarized after CXCL12 activation, AGR1.135 and AGR1.137 treatments promoted a reduction in the number of polarized cells. As a control, AMD3100 treatment also abolished CXCL12-mediated cell polarization (*Figure 4B and C*). None of the compounds affected anti-CD3 mediated actin polymerization, ruling out a direct effect of the modulators on actin or actin-binding proteins *Figure 4—figure supplement 1*.

CXCR4 nanoclustering has been recently associated with the migratory phenotype of T cell blasts (*García-Cuesta et al., 2022*; *Gardeta et al., 2022*). To explore how the allosteric modulators affect the phenotype of migrating cells, we added T cell blasts previously treated with vehicle or with the selected compounds to a 2D lipid bilayer system with embedded ICAM-1, alone or with CXCL12. We observed that treatment with AGR1.135, AGR1.137, or AMD3100 (control), but not with AGR1.131, abolished CXCL12-induced cell migration and cell adhesion (*Figure 4D and E*).

As the antagonistic effects of AGR1.135 and AGR1.137 are also compatible with a partial agonist behavior, we used Transwell chambers to evaluate the ability of the selected compounds to promote Jurkat cell migration. None of the compounds induced cell migration (*Figure 4—figure supplement 2*). Overall, our results suggest that AGR1.135- and AGR1.137-driven modulation of CXCR4 nanoclustering blocks certain receptor-associated functions, including actin dynamics, directional cell migration, integrin-mediated adhesion and migration, while leaving other functions unaffected (receptor internalization, inhibition of cAMP production and ERK1/2 activation).

## AGR1.135 and AGR1.137 antagonists act by direct binding on CXCR4

We developed a fluorescence spectroscopy strategy and FRET assays to formally examine the ability of the compounds to bind CXCR4, but our efforts were unsuccessful. AGR1.135 exhibited a yellow color that interfered with these techniques, and AGR1.137 did not alter the FRET efficiency of CXCR4 dimers (*Figure 5*). Therefore, we designed an in silico approach including a final step of molecular dynamics (MD) analysis, which allowed us to define the CXCR4 residues involved in the binding of the compounds, and to generate point mutant receptors to determine the inhibitory effect of AGR1.135, AGR1.137, and AGR1.131 (control) on CXCL12-mediated chemotaxis. We initially performed a binding-site search using PELE software, a Monte Carlo-based technique (*Borrelli et al., 2005*), to identify the most promising CXCR4 binding sites of the selected compounds. AGR1.135 and AGR1.137 exhibited one of the most stable trajectories (less than 0.25 Å$^2$ RMSD fluctuation) upon binding to the cleft formed by TMV and TMVI (*Figure 6A and B*). By contrast, the most stable trajectories for AGR1.131 corresponded to the binding to a contiguous region located between TMI and TMVII (*Figure 6C*).

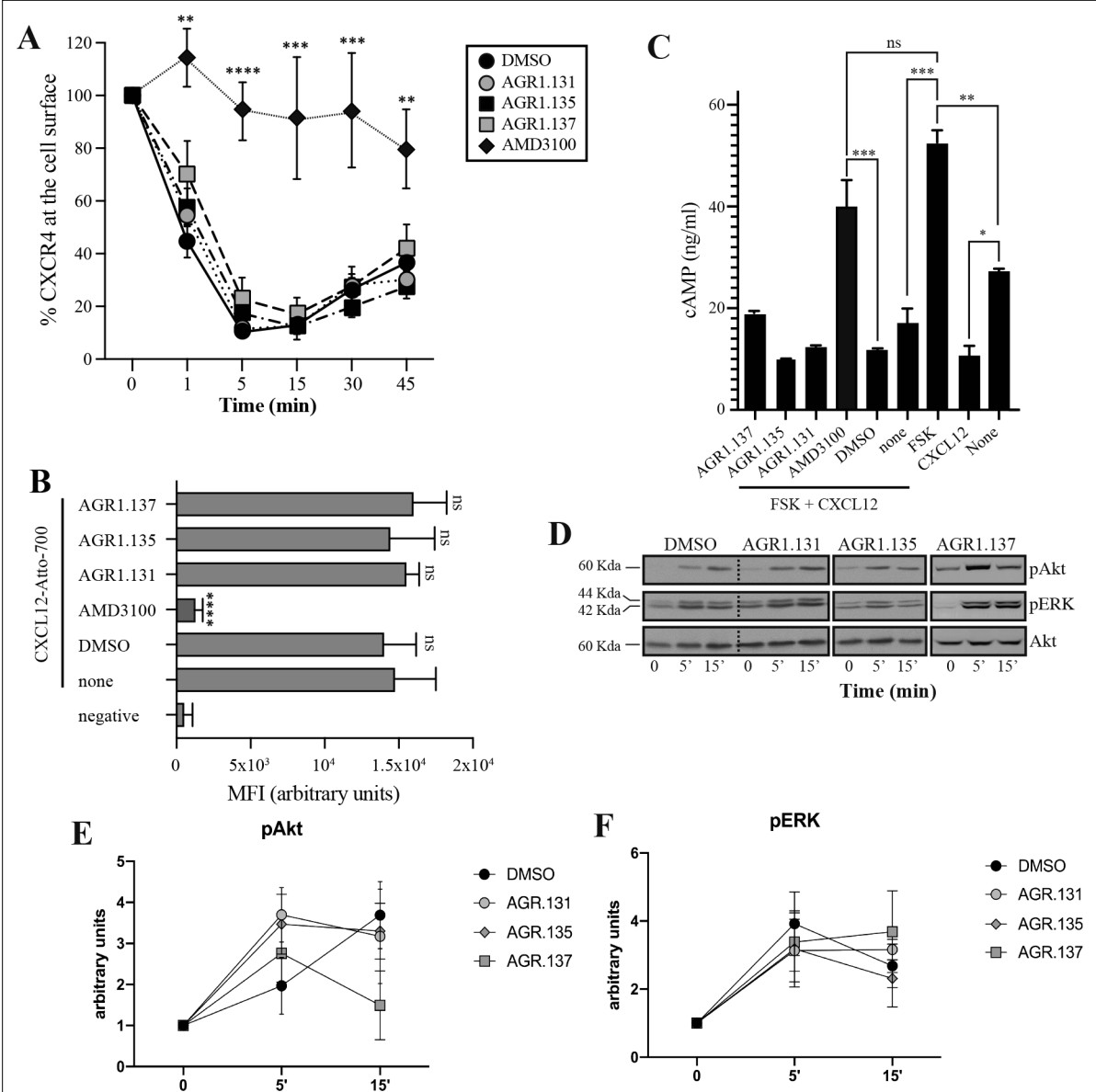

**Figure 3.** AGR1.135 and AGR1.137 treatments do not block CXCL12-mediated signaling pathways. (**A**) Cell surface expression of CXCR4 in Jurkat cells after stimulation with CXCL12 (12.5 nM) at different time points and analyzed by flow cytometry using an anti-CXCR4 antibody in nonpermeabilized cells. Results show mean ± SEM of the percentage of CXCR4 expression at the cell surface (n=4; n.s. not significant; **p≤0.01, ***p≤0.001, **** p≤0.0001). (**B**) CXCL12-Atto-700 binding on untreated or Jurkat cells pre-treated with DMSO (vehicle), AGR1.131, AGR1.135, AGR1.137 or with AMD3100 as a control, followed by flow cytometry analysis. Results are expressed as mean fluorescence intensity (MFI) values (arbitrary units). Negative corresponds to basal cell fluorescence in the absence of CXCL12-Atto-700 (mean ± SD, n=3 n.s. not significant; ****p≤0.0001). (**C**) Cells untreated or pretreated with the small compounds (50 µM, 30 min 37 °C) or AMD3100 (1 µM, 30 min 37 °C) were stimulated with CXCL12 (50 nM, 5 min, 37 °C) followed by forskolin (10 µM, 10 min, 37 °C). Cells were then lysed and cAMP levels were determined (mean ± SD; n=3; n.s. not significant; *p≤0.05, **p≤0.01, ***p≤0.001). (**D**) Western blot analysis of phospho (p) Akt and pERK in Jurkat cells pre-treated with DMSO, AGR1.131, AGR1.135 or AGR1.137, in response to CXCL12. As a loading control, membranes were re-blotted with an anti-Akt antibody. Representative experiments are shown (n=4). (**E, F**) Densitometry evaluation of the specific bands in D (mean ± SD; n=4).

The online version of this article includes the following source data and figure supplement(s) for figure 3:

**Source data 1.** PDF file containing original western blots for *Figure 3D*, indicating the relevant bands and treatments.

**Source data 2.** Original files for western blot analysis displayed in *Figure 3D*.

**Figure supplement 1.** Flow cytometry analysis of the binding of CXCL12-ATTO700 to JK cells.

**Figure supplement 2.** Western blot analysis of AMD3100-treated JK cells.

*Figure 3 continued on next page*

Figure 3 continued

**Figure supplement 2—source data 1.** PDF file containing original western blots for *Figure 3—figure supplement 2*, indicating the relevant bands and treatments.

**Figure supplement 2—source data 2.** Original files for western blot analysis displayed in *Figure 3—figure supplement 2*.

Once the binding site of the compounds was confirmed, we performed docking studies to identify the best poses as a starting point for further studies.

Results showed that AGR1.135 and AGR1.137 shared a similar binding mode (*Figure 6A and B*) and might interact with CXCR4 in a molecular cavity between TMV and TMVI. Specifically, AGR1.135 formed hydrogen bonds with residues G207, Y256 and R235, although a pi-cation interaction was also possible in the latter case (*Figure 6A*). Additionally, AGR1.135 could interact hydrophobically with several residues in TMV and TMVI, comprising two independent cavities of 434 and 1,381 Å$^3$, respectively (*Figure 6—figure supplement 1*), which were not connected to the orthosteric site as indicated by SURFNET analysis (*Laskowski, 1995*) included in the PDBsum server (*Laskowski et al., 1997*; *Figure 6—figure supplement 1*). AGR1.137 could use the carboxyl group of V124 in TMIII and overlap with AGR1.135 binding in the cavity, interacting with the other 19 residues scattered between TMV and VI to create several interaction surfaces of 790 and 580 Å$^3$ (*Figure 6B*; *Figure 6—figure supplement 1*). AGR1.137 did not have the phenyl ring present in AGR1.135, likely explaining why it did not interact with residue R235 (*Figure 1C*, *Figure 6B*). Regarding AGR1.131, its best binding site was the pocket between helices TMI and TMVII (*Figure 6C*), which is likely the reason why this compound had no effect on receptor oligomerization. These results were also confirmed using the CXCR4 model predicted by AlphaFold (*Jumper et al., 2021*) and the sequence available under UniProt entry P61073.

To confirm the putative binding sites for AGR1.135 and AGR1.137, we generated several point mutations in CXCR4 using the identified residues that were not critical for signal transduction processes. These included the G207I, L208K, R235L, F249L and Y256F mutants. We also included the mutations I204K, S260A as controls, as they are located in the same region (TMV-TMVI cleft) but did not interfere with compound binding or the CXCL12 binding site. All CXCR4 mutants were normally expressed at the cell surface, as demonstrated by flow cytometry (*Figure 7—figure supplement 1*), and were fully functional in CXCL12-mediated cell chemotaxis assays (*Figure 7—figure supplement 1*).

We next used transient transfection of JK$^{-/-}$ cells with each of the mutants or with wild-type CXCR4 treated with AGR1.131, AGR1.135 or AGR1.137 (50 µM) prior to assessing migration to CXCL12 gradients in Transwell chambers. The antagonistic effect of AGR1.135 was reversed in cells transfected with CXCR4 carrying the specific point mutations L208K or Y256F (*Figure 7A*). Similarly, the antagonistic effect of AGR1.137 was reversed in cells expressing G207I, L208K, R235L, F249L, or Y256F mutants (*Figure 7A*). Collectively, these data indicate that the selected compounds bind directly to CXCR4, that the L208 and Y256 residues in the CXCR4 TMVI domain are critical for the allosteric modulation activities of AGR1.135 and AGR1.137, and that other residues, G207, R235 and F249, are also important for the effect of AGR1.137 (*Figure 7A–D*).

## AGR1.137 treatment reduces tumor volume and dissemination in a zebrafish xenograft model

Finally, to determine the in vivo activity of the selected compounds, we considered the role of CXCR4 in tumor growth and metastasis (*Müller et al., 2001*; *Orimo et al., 2005*) and the transparency of zebrafish larvae to develop a tumor xenograft (ZTX) model. ZTX models have proven to be a powerful complementary in vivo system for research in oncology and tumor biology (*Rouhi et al., 2010*; *Xiao et al., 2020*), particularly for early tumor invasion and dissemination.

We used HeLa cells, which are derived from a human cervical tumor that expresses functional CXCR4 (*Figure 8A and B*; *Figure 8—figure supplement 1*), and which have been used previously in ZTX models (*Brown et al., 2017*; *You et al., 2020*). HeLa cells respond to CXCL12 gradients in directional cell migration assays (*Figure 8A and B*) and, as in Jurkat cells, their treatment with AGR1.137 but not with AGR1.131 (control) abolished directional cell migration toward a CXCL12 gradient (*Figure 8A and B*, *Figure 8—videos 1–4*). As a control for specificity, CXCL2-driven HeLa cell

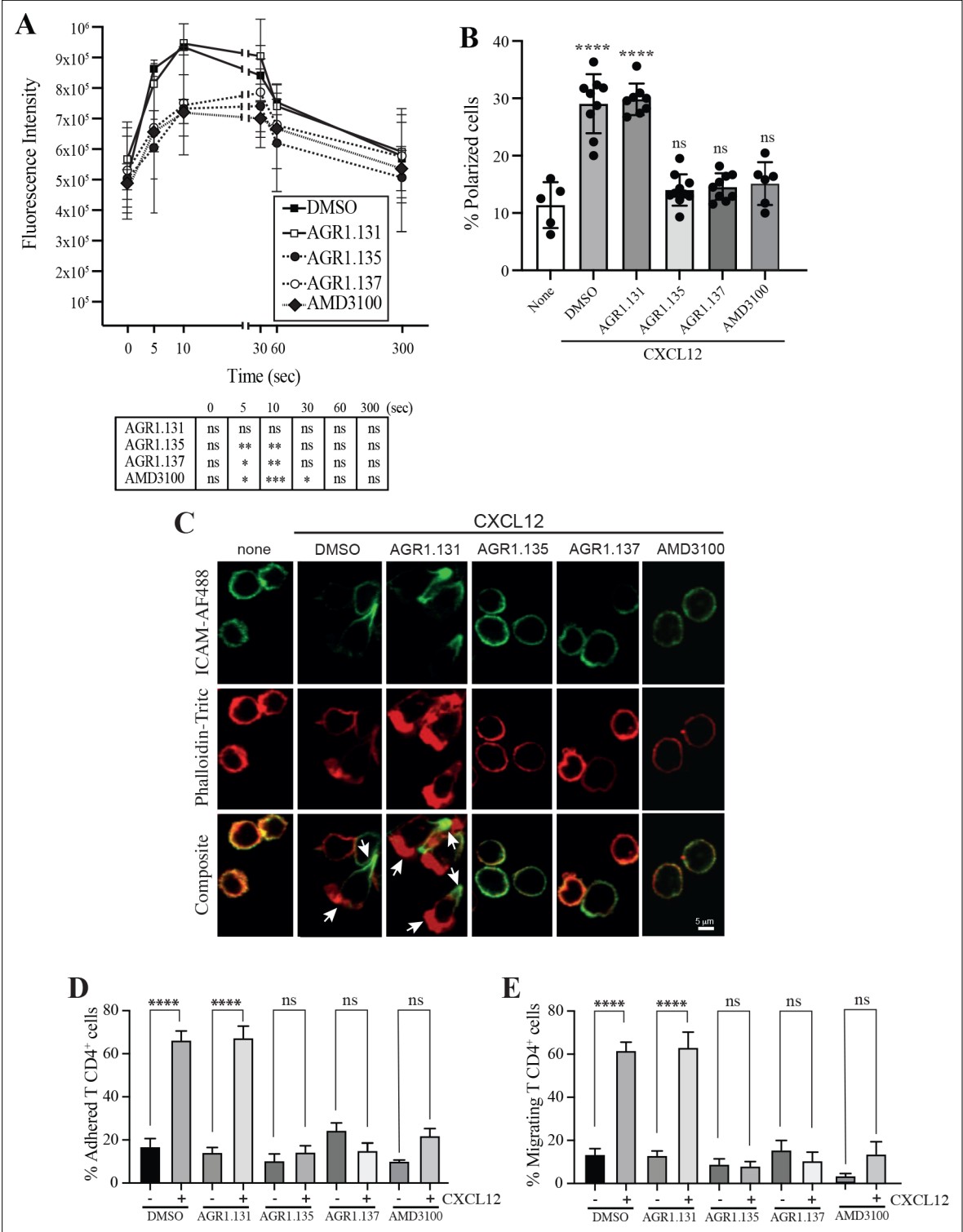

**Figure 4.** AGR1.135 and AGR1.137 treatment alter CXCL12-mediated actin polymerization. (**A**) Actin polymerization in response to CXCL12 as determined by F-actin (phalloidin-TRITC) staining in Jurkat cells treated with DMSO (vehicle) or the indicated modulators. Statistical significance of the different time points in comparison with the control (DMSO) is shown in the table (mean ± SD; n=3; n.s. not significant; *p≤0.05, **p≤0.01, ***p≤0.001). (**B**) Percentage of polarized T cell blasts adhered to fibronectin and treated or not with CXCL12 in the presence of the indicated antagonists, as analyzed by immunostaining with anti-ICAM3-Alexa fluor 488 and phalloidin-TRITC. More than 500 cells were analyzed in each condition. Data are presented as percentage of polarized cells (mean ± SD; n=3; n.s. not significant; ****p<0.0001). (**C**) Representative images of T cell blasts adhered to fibronectin and treated or not with CXCL12 in the presence of the indicated antagonists, as analyzed by immunostaining with anti-ICAM3-Alexa fluor 488 and phalloidin-

*Figure 4 continued on next page*

*Figure 4 continued*

TRITC as in B. (**D**) CD4+ T cells pretreated with AGR1.131, AGR1.135 or AGR1.137 were perfused in flow chambers coated with ICAM-1-containing lipid bilayers, alone or CXCL12-coated, and analyzed for cell contacts with the substrate. Data are presented as percentage of adhered cells (mean ± SD; n=3; n.s. not significant; ****p≤0.0001). (**E**) Cells in D were analyzed for cell migration. Data are presented as percentage of migrating cells (mean ± SD; n=3; n.s. not significant; ****p≤0.0001).

The online version of this article includes the following figure supplement(s) for figure 4:

**Figure supplement 1.** AGR1.135 and AGR1.137 do not alter anti-CD3-induced actin polimerization.

**Figure supplement 2.** AGR1.131, AGR1.135 and AGR1.137 are not CXCR4 agonists.

migration was unaffected by AGR1.137 treatment but was blocked by the specific CXCR2 inhibitor AZD5069 (*Figure 8C and D*).

DiI-labeled HeLa cells treated with vehicle, AMD3100, AGR1.135, AGR1.137, or AGR1.131 (control) were implanted subcutaneously into the perivitelline space of 2-day-old zebrafish larvae. Images of each tumor-bearing larva were taken immediately after implantation and 3 days later, and the relative change in tumor size was determined. AGR1.135 was discarded because most of the treated larvae died, indicating a toxic effect. By contrast, AGR1.137 was not toxic to embryos, as shown by visual inspection of the larvae at the end of the experiment, and no other non-lethal toxic phenotypes (e.g. pericardial edema, head and tail necrosis, head or tail malformation, brain hemorrhage, or yolk sac edema) were evident. AGR1.137 10 µM had no effect but 50 µM significantly reduced the size of high-intensity red fluorescent tumors (~40%), whereas the control AGR1.131 had no effect on tumor size as compared with vehicle-treated fish, and AMD3100 reduced tumor size to a similar extent as AGR1.137 (*Figure 8E*). These data demonstrate an antiproliferative effect of AGR1.137 on HeLa cells. As a control, AGR1.137 also reduced HeLa cell proliferation in in vitro assays mimicking the in vivo experimental conditions in zebrafish (addition of compounds every 24 hr for 3 days; *Figure 8—figure supplement 1*). We next evaluated the ability of the compounds to block cell dissemination by determining the number of labeled HeLa cells that appeared in the main metastatic niche at the caudal hematopoietic plexus. AGR1.137 50 µM treatment of HeLa cells reduced cell dissemination by ~40% (*Figure 8F*), strongly suggesting that this compound exerts anti-metastatic effects. As control AMD3100 (10 µM) treatment also reduced cell dissemination (~30%). The combined ability of AGR1.137 to reduce proliferation and dissemination in vivo is likely to have therapeutic value.

## Discussion

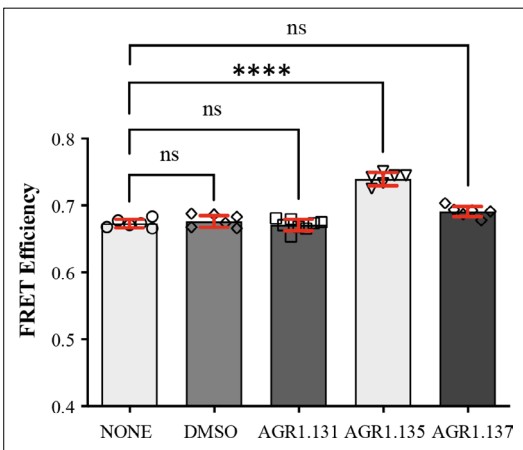

**Figure 5.** FRET analysis of CXCR4 in the presence of AGR1.131, AGR1.135 and AGR1.137. FRET efficiency in HEK-293 cells transiently transfected with CXCR4-YFP/ CXCR4-CFP (ratio 15:9), in the presence of 50 µM AGR1.131, 50 µM AGR1.135, 50 µM AGR1.137 or vehicle (DMSO). Data shows FRET efficiency (a.u.) (mean ± SD; n=6; n.s. not significant, **** p≤0.0001).

The CXCL12/CXCR4 axis is involved in myriad functions, including leukocyte recruitment, embryogenesis, vascular development, hematopoiesis, heart development, nervous system organization, tumor progression, autoimmune diseases and, together with CD4, CXCR4 is a co-receptor used by HIV-1 to infect immune cells (*Pozzobon et al., 2016*). This evidence supports the clinical interest in developing specific antagonists to modulate or directly block CXCR4 functions.

Plerixafor (AMD3100), which blocks ligand binding, is the first FDA-approved CXCR4 antagonist, and is used in peripheral blood stem cell transplantation regimens (*Fruehauf et al., 2012*). Unfortunately, its clinical use is limited by poor pharmacokinetics and adverse effects associated with long-term administration (*Choi et al., 2010*; *Hendrix et al., 2004*). These limitations and the poor clinical success of other chemokine receptor antagonists have prompted the search for alternative strategies to block chemokine receptor functions.

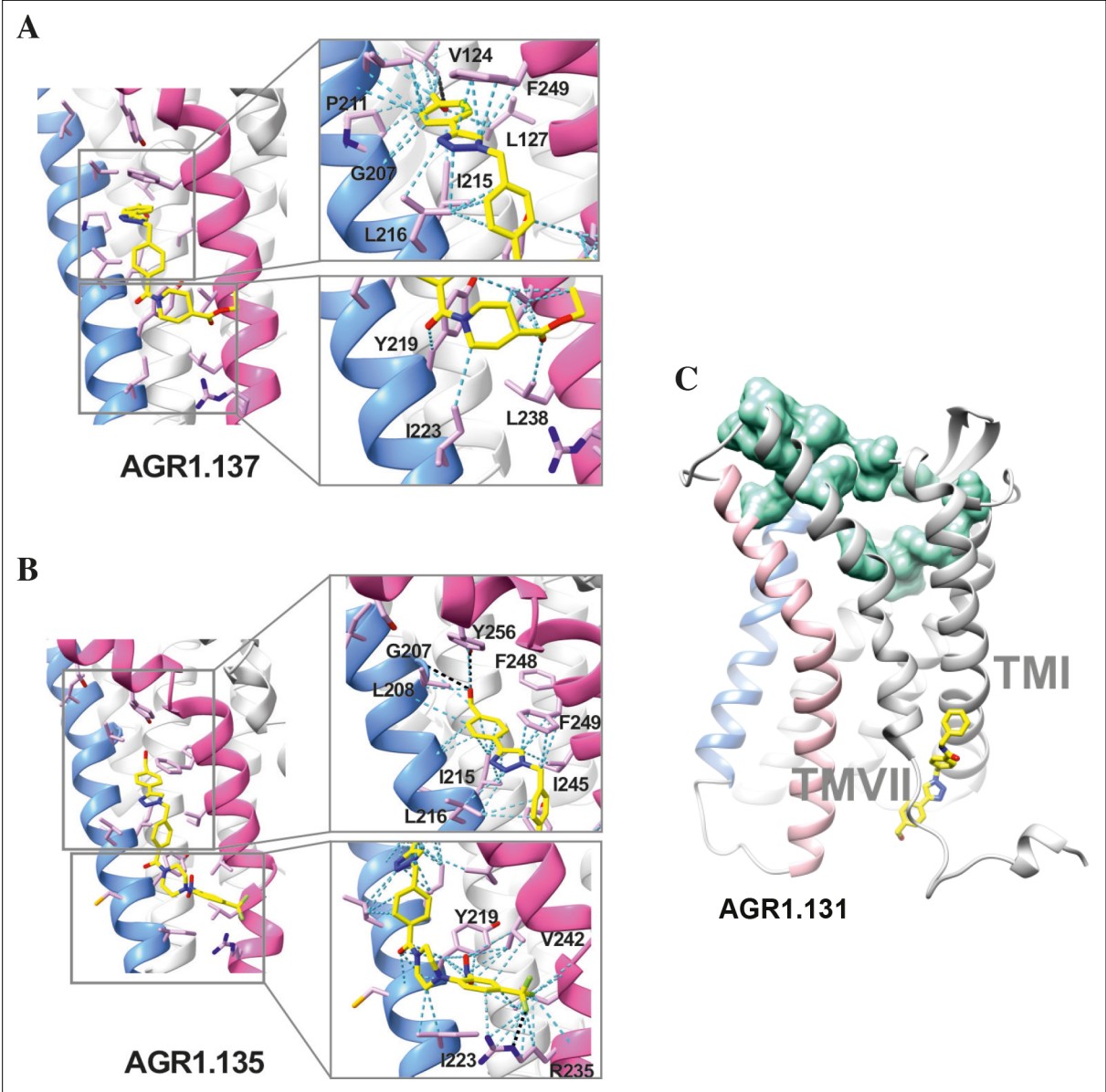

**Figure 6.** Ribbon and sticks representation of the CXCR4 modulators bound to the receptor. In the ribbon and sticks representations, the TMV helix is colored in blue and TMVI in pink. Figures show the binding of ligands AGR1.135 (**A**) and AGR1.137 (**B**) Zoom images showing the residues involved in the interaction of both compounds are also shown (right panels).(**C**) Binding of AGR1.131 represented as sticks with carbon atoms in yellow, oxygen in red, nitrogen in blue and fluorine in green. CXCR4 residues involved in CXCL12 engagement and initial signal transmission are represented as green spheres.

The online version of this article includes the following figure supplement(s) for figure 6:

**Figure supplement 1.** Cartoon and surface representation of CXCR4 and the clefts identified between TMV and TMVI.

As for other GPCRs, CXCR4 exists in multiple conformations on the plasma membrane and interacts with other chemokine receptors (*Muñoz et al., 2012*) and with cell surface proteins such as the T-cell receptor (*Kumar et al., 2006*), CD4 and other receptors (*Martínez-Muñoz et al., 2018*; *Pozzobon et al., 2016*). Advanced imaging-based techniques such as SPT-TIRF and super-resolution microscopy have revealed the existence of CXCR4 monomers, dimers and oligomers at the cell membrane, diffusing into the lipid bilayer. CXCL12 binding induces receptor nanoclustering and reduces the proportion of monomers and dimers, which is essential for the full activation of CXCR4 signaling and the correct orientation of the cell towards chemokine gradients (*Martínez-Muñoz et al., 2018*; *García-Cuesta et al., 2022*; *Gardeta et al., 2022*). CXCR4R[334X], a truncated mutant receptor

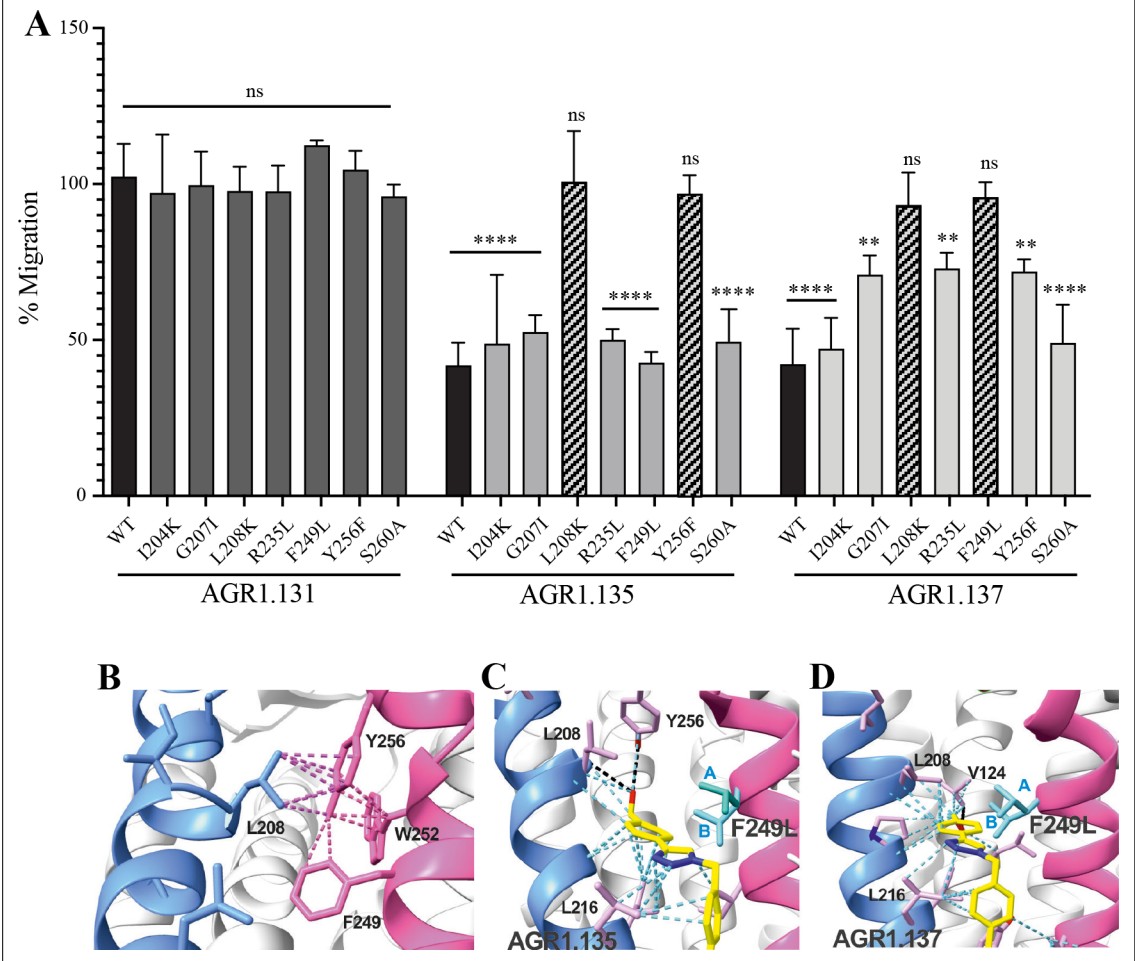

**Figure 7.** The antagonistic behavior of AGR1.135 and AGR1.137 depends on specific residues of CXCR4. (**A**) CXCL12-induced migration of AGR1.131-AGR1.135- or AGR1.137- pretreated JK$^{-/-}$ cells transiently transfected with CXCR4wt or the different mutants described. Data are shown as the mean percentage ± SD of input cells that migrate (n=4; n.s. not significant, **p≤0.01, ****p≤0.0001). (**B**) Cartoon representation of Y256 and its intramolecular interactions in the CXCR4 X-ray solved structure 3ODU. (**C, D**) Cartoon representation of the interaction of CXCR4 F249L mutant with AGR1.135 and AGR1.137, respectively. The two most probable conformations of leucine rotamers are represented in cyan. Van der Waals interactions are depicted in cyan dashed lines and hydrogen bonds in black dashed lines.

The online version of this article includes the following figure supplement(s) for figure 7:

**Figure supplement 1.** Expression and function of CXCR4 and CXCR4 mutants coupled to AcGFP in transfected JK-/- cells.

that causes WHIM syndrome, fails to form nanoclusters in the presence of CXCL12 and is unable to sense chemoattractant gradients, although it remains capable of triggering Ca$^{2+}$ flux and other signaling pathways (*García-Cuesta et al., 2022*). Due to the inability of CXCR4 to associate with the actin cytoskeleton, β-arrestin-1-deficient cells do not exhibit CXCL12-mediated receptor nanoclustering (*García-Cuesta et al., 2022*). Similar observations are seen CXCR4-expressing cells treated with latrunculin A, an inhibitor of actin polymerization (*Martínez-Muñoz et al., 2018*; *Pozzobon et al., 2016*). Collectively, these observations suggest that receptor nanoclustering is a ligand-mediated process that requires activation of a signaling pathway involving β-arrestin-1 and proper actin cytoskeletal dynamics.

CXCR4 contains a pocket formed by two TM domains, TMV and TMVI, and mutations in this pocket can affect the dynamics of the conformational changes triggered by ligand binding. The mechanism involves the initial interaction of CXCL12 with the extracellular region of CXCR4, which promotes conformational changes at a series of hydrophobic residues located mainly on TMV and TMVI that continue downward in the TM domains of the receptor, ultimately allowing G-protein interaction and activation (*Wescott et al., 2016*). Interestingly, a complete alignment of GPCR class A chemokine

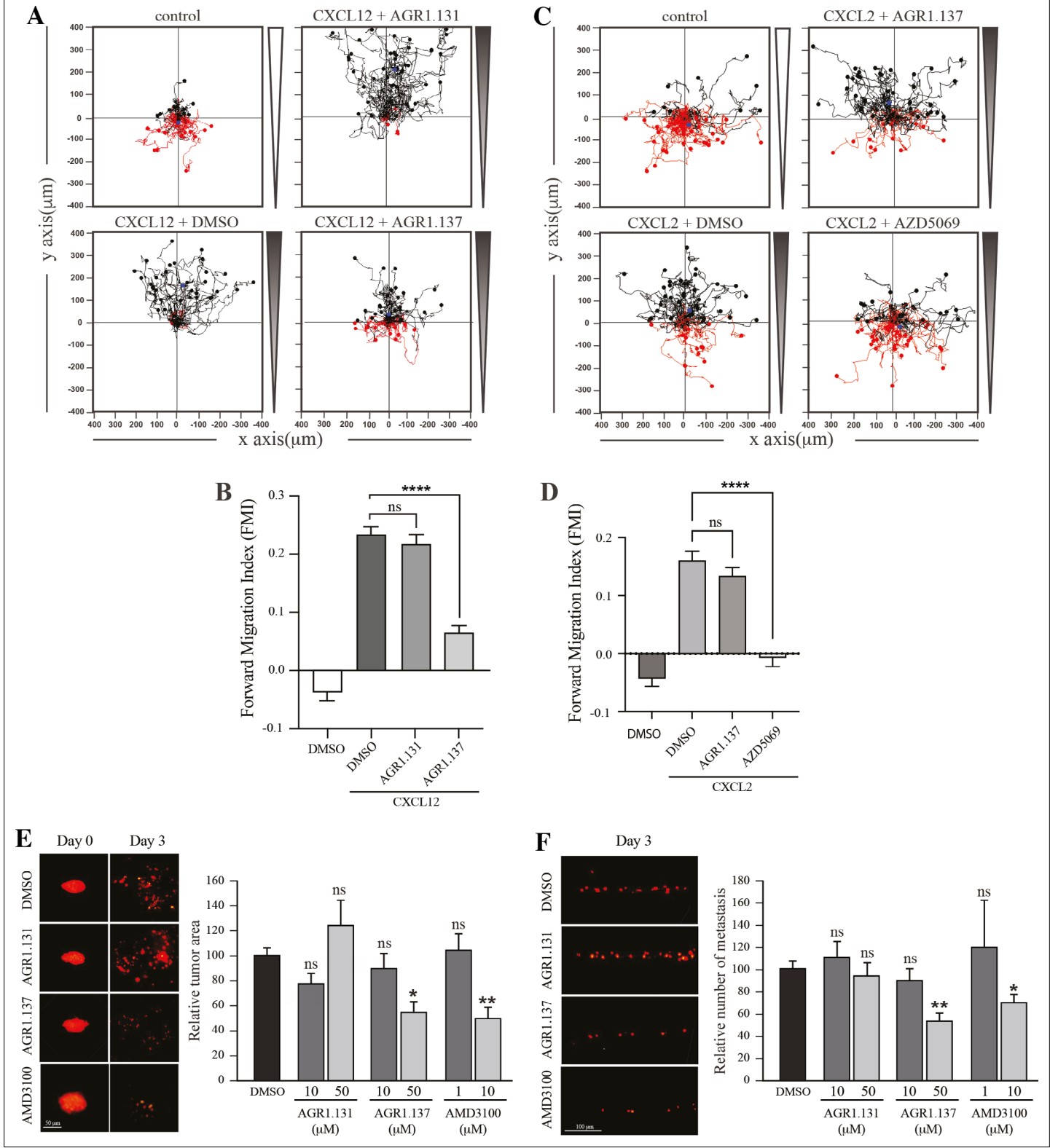

**Figure 8.** AGR1.137 reduces tumorigenesis and metastasis in a zebrafish model. (**A–D**) Migration of HeLa cells treated with vehicle (DMSO) or with the selected small compounds or inhibitor as indicated, on μ-chambers in response to a CXCL12 (**A, B**) or CXCL2 (**C, D**) gradient (n=2, in duplicate, with at least 50 cells tracked in each condition). Panels (**A**) and (**C**) show representative spider plots with the trajectories of tracked cells migrating along the gradient (black) or moving in the opposite direction (red). Black and red dots in the plots represent the final position of each single tracked cell. (**B, D**) Quantification of the Forward Migration Index of experiments performed in (**A**) and (**C**) (mean ± SD; n=3; n.s. not significant, ****p≤0.0001). (**E**)

*Figure 8 continued on next page*

*Figure 8 continued*

Representative fluorescent images of DiI-labeled HeLa cells in zebrafish larvae treated with vehicle (DMSO), AGR1.131 50 µM, AGR1.137 50 µM or AMD3100 10 µM at 0 or 3 days post-implantation and treatment. Quantitation of the relative tumor size at day 3 compared with that of day 0 normalized to the relative tumor size in the DMSO control group, is shown for each experimental group (mean ± SD; n=20; n.s. not significant, **p≤0.01). (**F**) Representative fluorescent images of the caudal hematopoietic plexus of larvae from the same groups as shown in E at 3 days postimplantation. Quantitation of the relative amount of metastasized cells in each group relative to DMSO-controls is shown (mean ± SD; n=20, n.s. not significant, * p≤0.05, *** p≤0.001).

The online version of this article includes the following video and figure supplement(s) for figure 8:

**Figure 8—video 1.** Related to *Figure 8A*.

https://elifesciences.org/articles/93968/figures#fig8video1

**Figure 8—video 2.** Related to *Figure 8A*.

https://elifesciences.org/articles/93968/figures#fig8video2

**Figure 8—video 3.** Related to *Figure 8A*.

https://elifesciences.org/articles/93968/figures#fig8video3

**Figure 8—video 4.** Related to *Figure 8A*.

https://elifesciences.org/articles/93968/figures#fig8video4

**Figure supplement 1.** Characterization of HeLa cells.

receptors reveals that hydrophobic residues in this region that are not in highly conserved positions can have any side-chain length at any site depending on the receptor, likely conferring a specificity that could be exploited for the discovery and/or design of new antagonists (*Ma et al., 2021*; *Figure 9*). A recent report using meta-dynamic approaches highlights the importance of the TM regions for dimerization and oligomerization of CXCR4 and CCR5 (*Di Marino et al., 2023*).

The majority of GPCR-targeted drugs are functionally active by binding to the orthosteric site of the receptor. Nonetheless, these orthosteric compounds have off-target effects and poor selectivity due to highly homologous receptor orthosteric sites and abrogation of spatial and/or temporal endogenous signaling patterns. The use of negative allosteric modulators can tune receptor-associated functions without affecting the orthosteric binding site. For example, the negative allosteric modulator 873140 blocks CCL3 binding to CCR5 but does not alter CCL5 binding (*Watson et al., 2005*). Maraviroc, an anti-HIV-1 molecule is an allosteric modulator of CCR5 that regulates CCR5 dimer populations, their localization to the cell membrane, and their subsequent subcellular trafficking (*Jin et al., 2018*). Allosteric modulators of the kappa opioid receptor have been demonstrated to have desirable

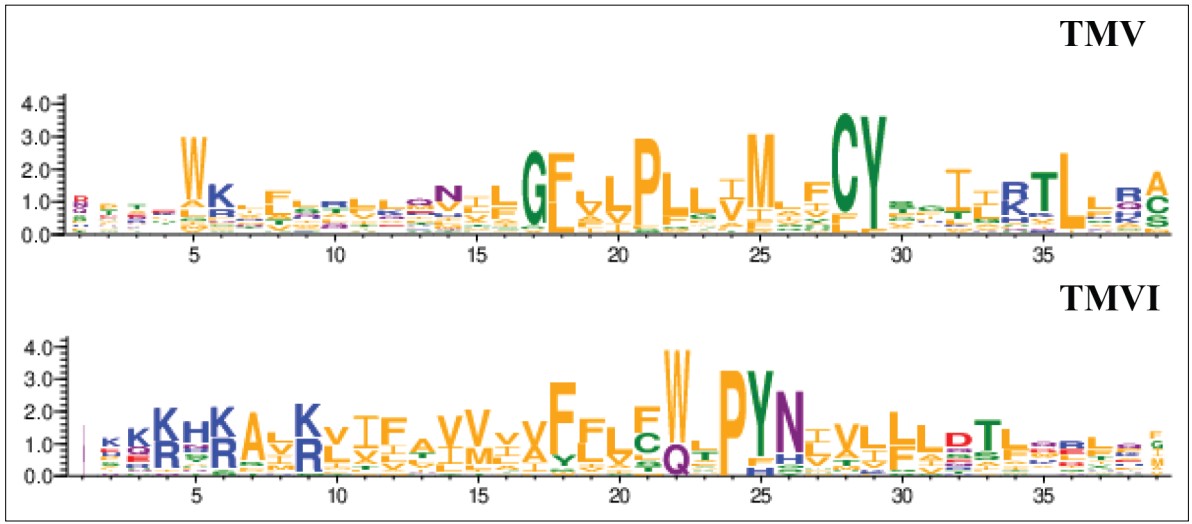

**Figure 9.** Residue conservation among all GPCR class A chemokine receptors for TMV and TMVI. Graphs were produced in the WebLogo server after aligning all receptors sequence in the sequence alignment tool included in GPCRdb. Font size for each residue represented in one letter code stands for the degree of conservation among all chemokine receptors. Residues are colored by their chemistry being green for polar, orange for hydrophobic, red for acidic, blue for basic and purple for neutral.

antinociceptive and antipruritic effects without sedative and dissociative effects in rodent models (*Brust et al., 2016*).

To identify new allosteric modulators, we focused on an in silico screen targeting the pocket in CXCR4 created by TMV and TMVI, without affecting the residues involved in CXCL12 binding (*Martínez-Muñoz et al., 2018*; *Pozzobon et al., 2016*). Subsequent functional analysis of the selected compounds identified AGR1.135, AGR1.137, and AGR1.131. The latter, although selected with the same in silico criteria, showed a better docking score on a cleft between TMI and TMVII and did not block CXCL12-mediated chemotaxis. In addition, the compounds had no effect on CXCR2-mediated cell migration, supporting their specificity for CXCR4. The three compounds had a common core of (4-(1-benzyl-1H-1,2,3-triazol-4-yl)phenyl) methanol but different side chains carrying amines of different length and chemical nature, which may explain their different biological activity. AGR1.131 seems to be more flexible than the other two compounds. It has an amide group and an additional CH2 linking the phenyl ring. The absence of the piperidine or piperazine rings in AGR1.131 may be related to its lack of activity on CXCR4. AGR1.137 is expected to be less hydrophilic than AGR1.135, as it contains only one tertiary amine while AGR1.135 contains two. These tertiary amines are susceptible to protonation to form a quaternary ammonium salt. In addition, the ethyl ester group present in AGR1.137 is less polar than the nitro group in AGR1.135, a feature that could affect the solubility of the compounds as well as their interactions on the receptor.

SPT-TIRF analysis showed that AGR1.135 and AGR1.137 abolished the ability of CXCL12 to induce CXCR4 nanoclustering and altered receptor dynamics at the cell membrane. Moreover, as expected from the selection criteria, these compounds had no effect on CXCL12 binding, receptor internalization, cAMP-production, or ERK1/2 or PI3K activation. The results also suggested the existence of different CXCR4 conformational states responsible for the activation of different signaling pathways. It is well known that GPCRs are thought to reside on the plasma membrane in equilibrium between distinct states, depending on complex allosteric interactions and ligand-induced conformational changes, as well as cell-specific parameters (*Vauquelin and Van Liefde, 2005*; *Wess et al., 2008*). The emerging view is that GPCRs are highly dynamic proteins, and ligands with varying pharmacological properties differentially modulate the balance between multiple conformations. In the absence of agonists, the receptor population is dominated by conformations closely related to those observed in inactive-state crystal structures (*Manglik et al., 2015*). While agonist binding drives the receptor population towards conformations similar to those in active-state structures, a mixture of inactive and active conformations remains, reflecting 'loose' or incomplete allosteric coupling between the orthosteric and transducer pockets (*Dror et al., 2011*). Surprisingly, for some GPCRs, and under some experimental conditions, a substantial fraction of unliganded receptors are already in an active-like conformation, which may be related to their level of basal or constitutive signaling (*Staus et al., 2019*; *Ye et al., 2016*). These basal activities can be modulated by ligands of varying potency. Full agonists can induce the maximal signaling response, whereas partial agonists and inverse agonists promote submaximal signaling and decrease basal activity, respectively. In addition, some ligands are known to be biased by selectively activating specific receptor-associated pathways at the expense of others, supporting the existence of distinct receptor conformations (*Kenakin, 2013*). Studies on the β2-adrenergic receptor support the idea that different agonists stabilize different receptor conformations (*Yao et al., 2006*), raising the possibility that allosteric ligands shift the balance to favor a particular receptor conformation. Biased ligands or allosteric modulators may thus achieve their distinctive signaling profiles by modulating this distribution of receptor conformations (*Wingler and Lefkowitz, 2020*). For example, some analogs of angiotensin II do not appreciably activate Gq signaling (e.g. increase IP3 and $Ca^{2+}$), but still induce receptor phosphorylation, internalization, and mitogen-activated protein kinase (MAPK) signaling (*Wei et al., 2003*). Some of these ligands activate Gi and $G_{12}$ in bioluminescence resonance energy transfer experiments (*Namkung et al., 2018*). A similar observation has been described for CCR5, where some chemokine analogs promote G protein subtype-specific signaling bias (*Lorenzen et al., 2018*). Structural analyses of various GPCRs in the presence of different ligands vary considerably in the extent and nature of the conformational changes in the orthosteric ligand-binding site upon agonist binding (*Venkatakrishnan et al., 2016*). However, these changes modify conserved motifs in the interior of the receptor core and induce common conformational changes in the intracellular site involved in signal transduction.

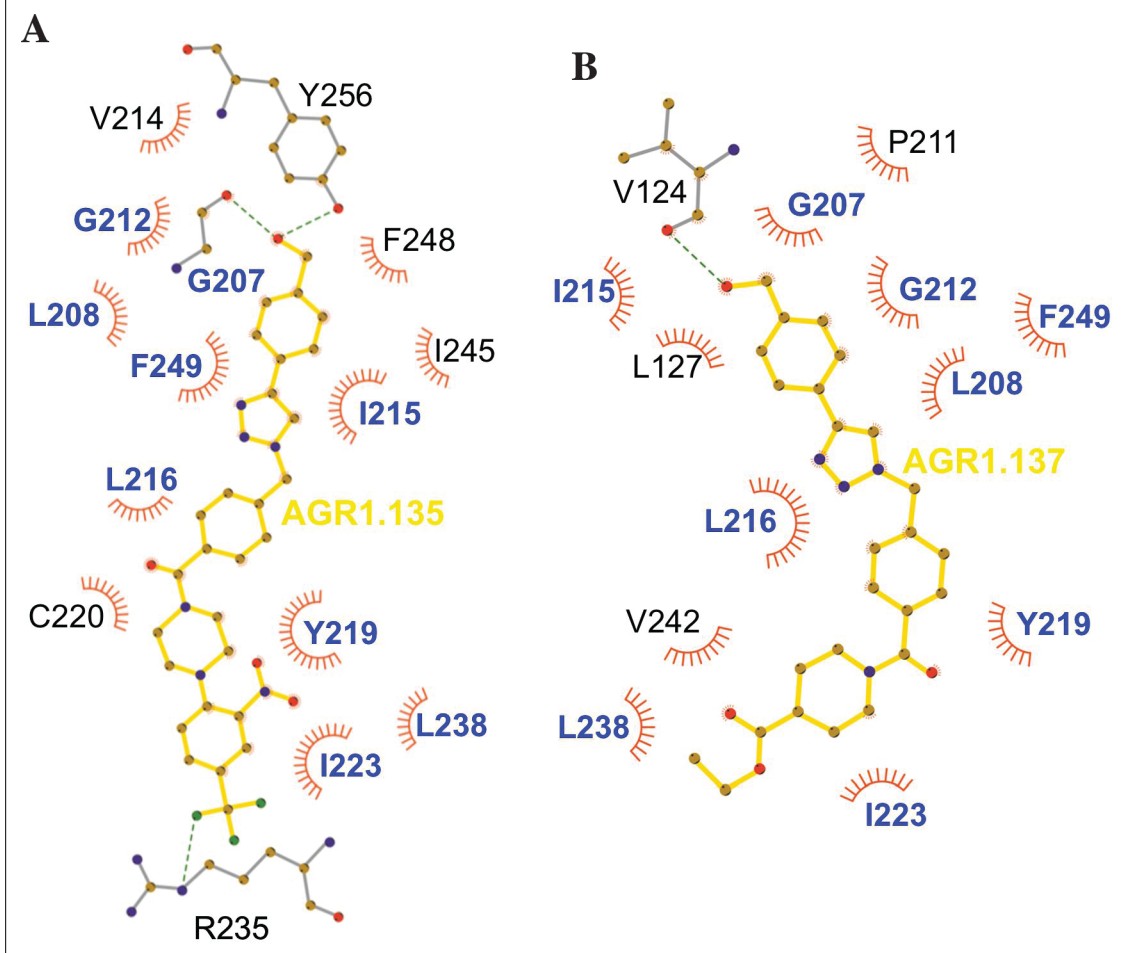

**Figure 10.** Ligplot representation of antagonists interactions with CXCR4. Ligplot representations of AGR1.135 (**A**) and AGR1.137 (**B**), with hydrogen bonds represented in green dotted lines and non-bonded contacts in red spoke arcs. Residues labeled in blue font are present in the interface of both AGR1.135 and AGR1.137. CXCR4 residues forming a hydrogen bond with the ligands are shown in gray.

We failed to detect a direct interaction between the compounds and CXCR4. Therefore, we employed an indirect strategy using docking and MD to confirm this interaction and to predict the most promising CXCR4 binding residues involved in the binding of the selected compounds. This methodology was combined with a mutation strategy to confirm the binding specificity of the allosteric modulators. AGR1.135 and AGR1.137 lost antagonism on cells expressing CXCR4L208K or CXCR4Y256F. LigPlot +analysis (*Laskowski and Swindells, 2011*) indicated that AGR1.135 interacts directly with the receptor through a hydrogen bond with Y256 (*Figure 10*). When this residue was mutated to F, one of the anchors for the compound was lost, weakening the potential interaction in the region of the upper anchor point. It is not clear how the Y256F mutation affects the binding of AGR1.137, but other potential contacts cannot be ruled out since this part of the compound is identical in both AGR1.135 and AGR1.137. This is particularly important for its neighboring residues in the alpha helix, F249, L208, as shown in the 3ODU structure, which appear to be directly involved in the interaction of both compounds with CXCR4. Alternatively, we cannot rule out an interaction between Y256 and other TMs or lipids stabilizing the overall structure, which could reverse the effect of the mutant at a later stage.

The inhibitory behavior of AGR1.137 was also lost in cells expressing CXCR4F249L. Both allosteric modulators showed stable trajectories bound to CXCR4, being able to occupy the cavity formed by TMV and TMVI and maintaining specific hydrogen-bond interactions and hydrophobic contacts along the interaction surface. Moreover, there were differences between the two modulators in terms of binding; whereas AGR1.135 showed a more elaborated interaction due to three hydrogen

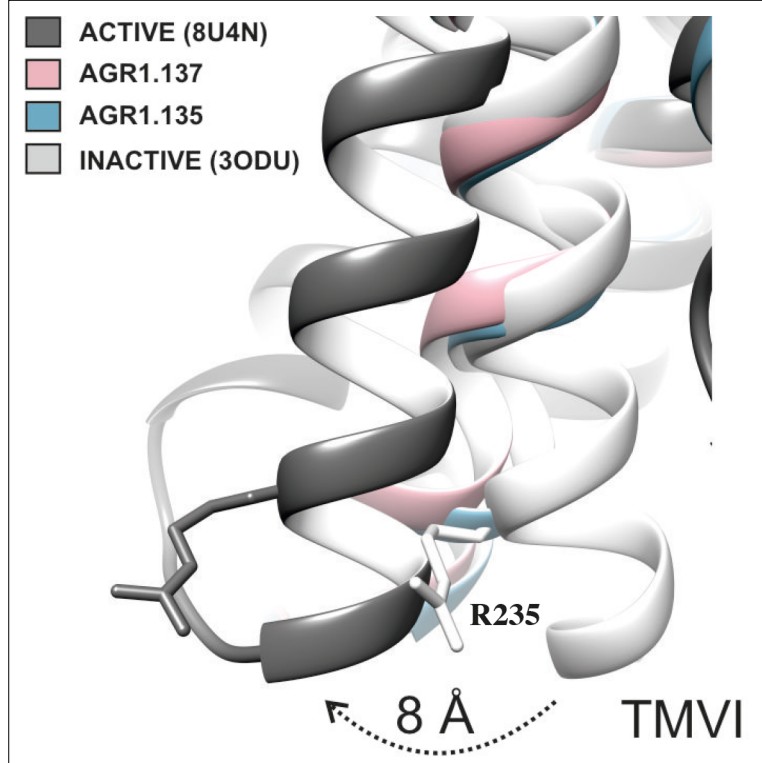

**Figure 11.** TMVI transition from inactive to active position. Ribbon representation of TMVI of CXCR4 structures superimposed for inactive (PDB code 3ODU), intermediate active (AGR1.135, AGR1.137) and active (PDB 8U4N) upon interaction with no, negative allosteric modulators and CXCL12-induced Gi-protein interaction, respectively. Arginine 235 is depicted in stick representation as a reference for the movement of the protein.

bonds that curved the position of the molecule as it fitted into the cavity formed by TMV and TMVI, AGR1.137 bound to the carboxyl group of V124 in TMIII, penetrating deeper into the receptor and then following the TMV–TMVI cleft down to the bottom of the cavity. In both cases, hydrophobic interactions with several residues of the receptor allowed the compounds to attach to the cleft and partially abolish the movement of TMVI required for complete signal transmission (*Wescott et al., 2016*). AGR1.137 allowed for a different conformation of the TMVI helix, one that is more shifted towards an active conformation (*Figure 11*). By contrast, predictions indicated that AGR1.135 established hydrogen bonds with residues G207, Y256, and R235, complemented by hydrophobic interactions in TMV and TMVI, comprising a total surface area of 400 Å$^2$ with a length of 20 Å. AGR1.135 appeared in a more stretched conformation than AGR1.137 due to its ability to share a hydrogen bond with R235. AGR1.137 could use the carboxyl group of V124 in TMIII and overlap with AGR1.135 binding in the cavity, interacting with the other 19 residues scattered between TMV and VI to create an interaction surface of 370 Å$^2$ along 20 Å. AGR1.137 did not have the phenyl ring present in AGR1.135, resulting in a shorter moiety with greater difficulty in reaching the lower part of TMVI where R235 sits. These differences may explain the different behavior of the two compounds in cells expressing CXCR4F249L and suggest a closer interaction between the triazole group of AGR1.137 and the F249 residue on CXCR4. The main difference between AGR1.135 and AGR1.137 is the way in which the triazole group interacts with F249 and L216. For AGR1.137, the three groups are aligned in a parallel organization, which appears to be more effective. This is because there is only one hydrogen bond with V124, which significantly contributes to a more flexible interaction. Contrastingly, AGR1.135 has two anchor points on either side of the link, which, together with the interaction with F248, results in a more rigid position in the cleft. When F249 is changed to L and the two most likely rotamers resulting from the mutation are shown, rotamer B appears in close proximity to the compound, which may cause the binding to either be displaced or to adopt an alternative conformation that is easier to bind into the cleft. As mentioned above, it is likely that AGR1.135 can displace

the mutant rotamer and bind more easily to the cleft due to its higher affinity. This region has also been implicated in the attachment of the orthosteric chemokine binding pocket to the lipid bilayer. F249 in CXCR4 as well as L132, V214 and L216 are involved in cholesterol binding (*Di Marino et al., 2023*). Thus, AGR1.137 may also affect cholesterol binding to CXCR4, thereby altering receptor oligomerization. The X-ray structure of CXCR4 with PDB code 3ODU (*Wu et al., 2010*) shows experimentally the presence of two fatty acid molecules in contact with both TMV and TMVI. Although further experiments are needed to clarify the mechanism, our results suggest that cholesterol and/ or other lipids might also play an important role in CXCR4 oligomerization and function, as seen for other GPCRs (*Jakubík and El-Fakahany, 2021*). We have recently shown that the lipid composition of the cell membrane modulates CXCR4 oligomerization. The chemokine field is very complex, and most receptors can form dimers (homo- and heterodimers) as well as oligomers (*Martínez-Muñoz, 2011*). It is therefore possible that the compounds might also affect the function of other chemokine receptors that heterodimerize with CXCR4. Nonetheless, it should be noted that the residues involved in CXCR4 oligomerization might differ from those involved in dimeric structures (*Di Marino et al., 2023*). These observations highlight the flexibility of the cavity and its potential for modulation. They also suggest that this cavity could be used to guide the development of a new generation of antagonists that, without affecting CXCL12 binding, allosterically modulate some GPCR-mediated functions without altering others.

CXCR4 overexpression contributes to tumor growth, invasion, angiogenesis, metastasis, relapse, and therapeutic resistance (*Chatterjee et al., 2014*). As might be expected, CXCR4 antagonism has been shown to disrupt tumor-stromal interactions, sensitize cancer cells to cytotoxic drugs, and reduce tumor growth and metastatic burden (*Chatterjee et al., 2014*). We used a ZTX model with HeLa cells expressing functional CXCR4 to evaluate the in vivo relevance of the selected antagonists. This is a well-studied model to assess tumor progression and metastasis (*Brown et al., 2017*). Our results demonstrate that AGR1.137 treatment of HeLa cells induce a dose-dependent reduction tumor size by 40% and limits cell dissemination in the absence of toxicity, suggesting that this compound has both anti-proliferative and anti-metastatic properties. While the effect of AGR1.137 on cell dissemination may be a consequence of its effect on tumorigenesis, it also abolished directed HeLa cell migration in vitro, supporting the inhibition of cell metastasis. The data support that AGR1.137 reduces tumor size and metastasis to a similar extent as AMD3100, but the allosteric modulator is highly hydrophobic and therefore it will be very important to improve solubility, increase affinity and optimize delivery, to facilitate its use, particularly in vivo. AMD3100, which is FDA-approved for clinical use, is an orthosteric inhibitor of CXCR4 that blocks the entire signaling cascade triggered by CXCL12. Indeed, we observed that AMD3100 treatment blocked CXCL12 binding, cAMP inhibition, calcium flux, cell adhesion and cell migration, whereas the effect of AGR1.137 was limited to CXCL12-mediated directed cell migration. Although AMD3100 was well tolerated by healthy volunteers in a single-dose study, it also promotes some mild and reversible events such as increases in white blood cell counts and variations in urinary calcium just outside the normal range (*Hendrix et al., 2000*). For the treatment of viral infections, continuous daily dosing of AMD3100 was impractical due to severe side effects including cardiac arrhythmias (*De Clercq, 2015*). It is therefore critical to control the timing of administration of AMD3100 in the context of therapeutic use. In addition, side effects after long-term administration are a potential problem. Shorter-term usage and lower doses would be fundamental keys to success in clinical use (*Liu et al., 2015*). The use of new drugs, such as the negative allosteric modulator AGR1.137, which block cell migration but do not affect other CXCL12-mediated signaling pathways, would, at least in theory, be more specific and produce fewer side effects.

Our data support the notion that CXCR4 is a flexible protein that can adopt a spectrum of conformations depending on several factors such as the presence of a bound ligand, the lipid composition of the cell membraneand the presence of other interacting proteins. As with many GPCRs, the stabilization of distinct CXCR4 states is a key element to modulate its function (*Bourque et al., 2017*; *Nygaard et al., 2013*). Our data show that by targeting the pocket between TMV and TMVI in CXCR4, AGR1.137 can block (both in vitro and in vivo) CXCL12-mediated receptor nanoclustering and cell sensitivity to chemoattractant gradients without altering ligand binding, thus preserving other signaling events. Ultimately, these results demonstrate the value of stabilizing specific CXCR4 conformations and considering targets other than the ligand-binding site for the design of negative allosteric modulators to block a specific set of CXCR4 functions.

# Materials and methods

## Cells and reagents

HeLa cells (CCL-2) and HEK293T cells (CRL-11268) were obtained from the American Type Culture Collection (Rockville, MD). Jurkat human leukemia CD4[+] cells were kindly provided by Dr. J. Alcamí (Centro Nacional de Microbiología, Instituto de Salud Carlos III, Madrid, Spain). Where indicated, Jurkat cells lacking endogenous *CXCR4* (JK[-/-]) (*García-Cuesta et al., 2022*) were transiently transfected with plasmids expressing wild-type or mutant CXCR4-AcGFP receptors (20 µg), as described (*García-Cuesta et al., 2022*). CXCR4 mutants were generated by PCR using the QuikChange site-directed mutagenesis kit (Stratagene, La Jolla, CA) with full-length CXCR4-AcGFP serving as a template, and specific primers (*Table 1*).

Human peripheral blood mononuclear cells isolated from buffy coats were activated in vitro for 1 week with 20 U/ml of IL-2 (Teceleukin; Roche, Nutley, NJ) and 5 µg/ml phytohemagglutinin (Roche) to generate T cell blasts (*Gardeta et al., 2022*).

The following antibodies were used: monoclonal mouse anti-human CXCR4 (clone 44717) and phycoerythrin-conjugated human CXCR4 (clone 12G5; both from R&D Systems, Minneapolis, MN); purified mouse anti-human CD3 (clone HIT3a, BD Biosciences, Franklin Lakes, NJ); goat F(ab')2 anti-mouse IgG-PE (Southern Biotech, Birmingham, AL); and anti-phospho-AKT (Ser473), anti-AKT (#9272), anti-phospho-ERK1,2 (#9191) and anti-ERK (#9102) (all from Cell Signaling Technology, Danvers, MA). Human CXCL12, CXCL2 and CCL17 were obtained from PeproTech (Rocky Hill, NJ). Human CXCR4 was cloned into the pAcGFPm-N1 plasmid (Clontech Laboratories, Palo Alto, CA), as described (*Martínez-Muñoz et al., 2018*).

Unless otherwise indicated, cells were pre-treated with 50 µM of the selected small compounds, 10 µM AMD3100 (Merck, Darmstadt, Germany), 1 µM AZD5069 (MedChemExpress, Monmouth Junction, NJ) or vehicle (DMSO) as control (30 min, 37 °C, 5% $CO_2$).

Binding experiments were performed using CXCL12-Atto-700 (*Ameti et al., 2018*), kindly donated by Prof. Marcus Thelen (Institute for Research in Biomedicine, Università della Svizzera italiana, Bellinzona, Switzerland).

## Compounds database

All the compounds used in the present study were synthesized in the Centro de Investigaciones Biológicas Margarita Salas (CIB-CSIC, Madrid) following previously described procedures (*Sebastián-Pérez et al., 2017*). Synthetic and analytical data are shown as Supplementary information. All the compounds have a purity ≥95% determined by high-performance liquid chromatography. Compounds are collected in the MBC library (CIB-CSIC) (*Sebastián-Pérez et al., 2017*), which contains more than 2,000 drug-like compounds.

## Computational studies of CXCR4 modulators

### Protein modeling

The CXCR4 model was built on the SWISS-MODEL server (*Waterhouse et al., 2018*) using the human CXCR4 sequence and the crystallographic structure of a CXCR4 dimer in complex with the small molecule IT1t as template (PDB code: 3ODU *Wu et al., 2010*). At the time we performed these analyses, several crystallographic structures of CXCR4 in complex with different molecules and peptides were deposited in the PDB. None of them were fully resolved because the N- and C-terminal ends of CXCR4 are very flexible loops. In addition, the CXCR4 constructs contained T4 lysozyme inserted between helices TMV and TMVI to increase the stability of the protein; this was a common strategy to facilitate crystallogenesis of GPCRs (*Zou et al., 2012*). Therefore, a CXCR4 homology model was generated using the SWISS-MODEL server (*Waterhouse et al., 2018*). This program reconstructed the loop between TMV and TMVI, a domain of particular importance in this study and not present in any of the crystal structures available in the PDB.

The model was further optimized by adding hydrogens, ionizing the structure at pH 7.2, and adjusting size chain positions using the Maestro Protein Preparation Wizard tool included in the Schrödinger software package (*Madhavi Sastry et al., 2013*).

**Table 1.** Oligonucleotides used to generate the CXCR4 mutants.

| Mutation | Forward | Reverse |
|---|---|---|
| CXCR4 G207I | cac atc atg gtt att ctt atc ctg cct | agg cag gat aag aat aac cat gat gtg |
| CXCR4 R235L | ggc cac cag aag tta aag gcc ctc aag | CTT GAG GGC CTT TAA CTT CTG GTG GCC |
| CXCR4 Y256F | TGG CTG CCT TAC TTT ATT GGG ATC AGC | GCT GAT CCC AAT AAA GTA AGG CAG CCA |
| CXCR4 S260A | TACTACATTGGGATCgccATCGACTCCTTCATC | GATGAAGGAGTCGATggcGATCCCAATGTAGTA |
| CXCR4 L208K | CACATCATGGTTGGCAAGATCCTGCCTGGTATT | AATACCAGGCAGGATCTTGCCAACCATGATGTG |
| CXCR4 I204K | TTCCAGTTTCAGCACAAGATGGTTGGCCTTATC | GATAAGGCCAACCATCTTGTGCTGAAACTGGAA |
| CXCR4 G212F | GCCCTTATCCTGCCTTTCATTGTCATCCTGTCC | GGACAGGATGACAATGAAAGGCAGGATAAGGCC |
| CXCR4 F249L | GTCATCCTCATCCTGGCTTTCTGCCTGTTGGCTGCCT | AGGCAGCCAACAGGCCAGGAAAGCCAGGATGAGGATGAC |

## Virtual screening

Virtual screening of ligands was performed from the compounds included in the MBC library using the "Glide" module as a docking tool on the CXCR4 model using standard-position (SP) and extra-position (XP) scoring functions. The center of the catalytic pocket was selected as the centroid of the grid. In the grid generation, a scaling factor of 1.0 in van der Waals radius scaling and a partial charge cutoff of 0.25 were used. A rescoring of the SP poses of each compound was then performed with the XP scoring function of Glide. The XP mode in Glide was used in the virtual screening, the ligand sampling was flexible, epik state penalties were added, and an energy window of 2.5 kcal/mol was used for ring sampling. In the energy minimization step, the distance dependent dielectric constant was 4.0 with a maximum number of minimization steps of 100,000. In the clustering, poses were considered as duplicates and discarded if both RMS deviation is less than 0.5 Å and maximum atomic displacement is less than 1.3 Å.

Simulations were calculated with an Asus 1151 h170 LVX-GTX-980Ti workstation, with an Intel Core i7-6500 K Processor (12 M Cache, 3.40 GHz) and 16 GB DDR4 2133 MHz RAM, equipped with a Nvidia GeForce GTX 980Ti available for GPU (Graphics Processing Unit) computations.

The computational sequential workflow applied in this project included: (i) Protein model construction, (ii) Virtual screening (Glide), (iii) PELE, (iv) Docking (autoDock and Glide) and (v) MD (AMBER).

## PELE

A Protein Energy Landscape Exploration (*Borrelli et al., 2005*) study was carried out for the control molecule IT1t, which is a small druglike isothiourea derivative that has been crystallized in complex with CXCR4 (PDB code: 3ODU). PELE is a method to explore the energy landscape of a protein that can perform unconstrained ligand exploration to search its binding site. PELE combines a Monte Carlo stochastic approach with protein structure prediction algorithms. The script 'Unconstrained ligand exploration and binding search' was used to calculate 20 PELE trajectories for each ligand of the PELE program, yielding 20 PELE trajectories for each ligand. From the best five trajectories, two of them presented clearly better binding energies, and corresponded to almost the same predicted pose of the molecule. Although the predicted binding mode was not exactly the same as that in the crystal structure, the approximation was very good, allowing the approach validation. Although PELE is a suitable technique to find potential binding sites, the predicted poses must be subsequently refined using docking programs.

## Docking studies

All compounds were adjusted to a pH of 7.2±0.2 using the Maestro LigPrep tool *LigPrep, 2016*. The processed geometry of these ligands was optimized using Gaussian09 software (*Frisch, 2016*), which calculates RESP (Restrained ElectroStatic Potential) charges and other molecular properties. The method used for calculations was Hartree-Fock/6–311++G(2d,2p), using the tight option to ensure adequate convergence. Subsequently, all ligands were parameterized using Antechamber (*Wang et al., 2006*). In this work, docking calculations were performed using Autodock v4.2 (*Morris et al., 2009*).

All docking assays were performed using a box that contained the intracellular half of the protein. The CXCL12 binding pocket was excluded from the search based on the experimental results showing that none of these molecules prevent CXCL12 binding (unpublished results). The grid was formed by 126×126 × 126 points with a spacing of 0.310 Å (~39 × 39×39 Å3). Ligand posing was performed using a stochastic search with Lamarckian Genetic Algorithms (GAs). GAs use a population of poses and combine them to generate offspring that are evaluated with the scoring function to select the best ones to create the next generation. The GA parameters were 200 runs, a population size of 150 and 2,500,000 evaluations The results were analyzed clustering the best 200 poses with a Root Mean Square Deviation (RMSD) of 2.0 Å.

## Molecular dynamics

The most promising docking poses from the docking studies were further studied using MD simulations to determine their stability and interactions with the receptor. The CXCR4-IT1t complex was used as a control to validate the protocol. Simulations were performed using AMBER14 (*Case et al.,*

*2014*) with ff14SB (*Maier et al., 2015*) and lipid14 (*Dickson et al., 2014*) force fields in the NPT thermodynamic ensemble (constant pressure and temperature). All bonds involving hydrogen atoms were constrained with the SHAKE algorithm (*Lippert et al., 2007*). To stabilize CXCR4 and more accurately reproduce the real environment in the MD simulation, the system was embedded in a lipid bilayer using the Membrane Builder tool (*Jo et al., 2009*) from the CHARMM-GUI server. The membrane was composed of 175 molecules of the fatty acid 1-palmitoyl-2-oleoyl-sn-glycero-3-pho sphocholine (POPC) in each leaflet. The protein-membrane complex was solvated with TIP3 water molecules. Chloride ions were added up to a concentration of 0.15 M in water, and sodium ions were added to neutralize the system. Minimization was performed using 3500 Steepest Descent steps and 4500 Conjugate Gradient steps three times, firstly considering only hydrogens, then considering only water molecules and ions, and finally minimizing all atoms. Equilibration raises system temperature from 0 to 300 K at a constant volume, fixing everything but ions and water molecules. After thermalization, several density equilibration phases were performed. In the production phase 50 ns MD simulations without position restraints were calculated using a time step of 2 fs. Trajectories of the most interesting poses were extended to 150 ns. 'Stable trajectories' have been defined to refer to those trajectories of the ligand-protein complexes whose RMSD does not fluctuate more than 0.25 Å. To estimate the affinity of the ligands for CXCR4, the binding energy of each representative structure obtained with the principal components analysis was calculated using the PRODIGY-LIGAND server (*Vangone and Bonvin, 2017*).

## Virtual screening validation

We also used a strategy based on the model of CXCR4 predicted by AlphaFold (*Jumper et al., 2021*) and the sequence available under UniProt entry P61073 to validate our studies. We prepared the ligands using the OpenBabel (*O'Boyle et al., 2011*), with a gasteiger charge assignment, and generated 10 conformers for each input ligand using OpenBabel. We then prepared the target structure with Openmm, removing all water and possible heteroatoms, and adding all missing atoms. We next predicted the target binding pockets with fPocket (*Le Guilloux et al., 2009*), p2rank (*Krivák and Hoksza, 2018*), and AutoDock autosite (*Ravindranath and Sanner, 2016*). We chose only those pockets between TMV and TMVI. We merged the results of the three programs into so-called consensus pockets, as two pockets are said to be sufficiently similar if at least 75% of their surfaces are shared (*Del Hoyo et al., 2023*). From the consensus pockets, one pocket was significantly larger than the others and was, therefore, selected. We then docked the ligand conformers in this pocket using AutoDock GPU (*Santos-Martins et al., 2021*), LeDock (*Liu and Xu, 2019*), and Vina (*Eberhardt et al., 2021*). The number of dockings varied from 210 to 287 poses. We scored each pose with the Vina score using ODDT (*Wójcikowski et al., 2015*). Then, we clustered the different solutions into groups whose maximum RMSD was 1 Å. This resulted in 40 clusters; the representative of each cluster was the one with maximum Vina score, and confirmed that the selected compounds bound this pocket. When required, we calculated the binding affinity using Schrodinger's MM-GBSA procedure (*Greenidge et al., 2013*) in two ways: first, assuming that the ligand and target are fixed; second, with an energy minimization of all the atoms within a distance of 3 Å from the ligand.

## Cleft volume calculations

Once we detected that several compounds bound CXCR4 in the target region, the cleavage properties were calculated by subtracting the compound structure. The resulting PDB was analyzed using the PDBsum server (*Laskowski et al., 2018*). Volume calculations were obtained using the SURFNET server to analyze surface clefts (*Laskowski, 1995*). The theoretical interaction surface between the selected compounds and CXCR4 and the atomic distances between the protein residues and the compounds was calculated using the PISA server (*Krissinel and Henrick, 2007*).

## Transwell migration assay

Cells ($3\times10^5$) in 0.1 ml of RPMI medium containing 10 mM HEPES and 0.1% BSA were placed in the upper wells of 5 μm pore size transmigration chambers (Transwell, Costar, Corning, NY). CXCL12 (12.5 nM) or CCL17 (50 nM) in 0.6 ml of the same medium was added to the lower well. Cell migration was evaluated as described (120 min, 37 °C, 5% $CO_2$) (*García-Cuesta et al., 2022*).

## Cell cycle analysis

Cells ($5\times10^5$ cells/well) were collected from microplates and washed twice in PBS and then resuspended in 50 µl of detergent (DNA-Prep Reagent Kit; Beckman Coulter, Brea, CA) containing 10 ng/ml propidium iodide (DNA-Prep Reagent Kit; 30 min, 37 °C). Cell cycle phases were analyzed by flow cytometry on a Beckman Coulter FC500 flow cytometer and results were expressed as the percentage of stained cells. $H_2O_2$ (10%) and staurosporine (10 µM) treatments were used as positive controls of cell death.

## Single molecule TIRF imaging and analysis

Transfected cells expressing 8500–22,000 receptors/cell (<4.5 particles/µm²) were plated on glass-bottomed microwell dishes (MatTek Corp., Ashland, MA) coated with fibronectin (Sigma-Aldrich, St. Louis, MO: 20 µg/ml, 1 hr, 37 °C) as a substrate for cell adhesion and to facilitate image aquisition, as TIRF determinations are restricted to the cell membrane in contact with the coverslip. These are conditions corresponding to the unstimulated cells. To observe the effect of the ligand, we coated dishes with CXCL12 (100 nM, 1 hr, 37 °C); stimulated cells were incubated (20 min, 37 °C, 5% $CO_2$) before image acquisition. Experiments were performed at 37 °C with 5% $CO_2$ using a total internal reflection fluorescence microscope (Leica AM TIRF inverted microscope; Leica Microsystems, Wetzlar, Germany). Image sequences of individual particles (500 frames) were then acquired at 49% laser power (488 nm diode laser) with a frame rate of 10 Hz (100 ms/frame). The penetration depth of the evanescent field was 90 nm. Particles were detected and tracked using the U-Track2 algorithm (*Jaqaman et al., 2008*) implemented in MATLAB, as described (*Sorzano et al., 2019*). Mean spot intensity (MSI), number of mobile and immobile particles and diffusion coefficients ($D_{1-4}$) were calculated from the analysis of thousands of single trajectories over multiple cells (statistics provided in the respective figure captions) using described routines (*Sorzano et al., 2019*). The receptor number along individual trajectories was determined as reported (*Martínez-Muñoz et al., 2018*), using the intensity of the monomeric protein CD86-AcGFP as a reference (*Dorsch et al., 2009*). Briefly, for each particle detected we measured the intensity of pixels around the particle (in a 3×3 pixel matrix whose central position is given by the coordinates x, y obtained by U-Track2), in each frame along its trajectory. We also estimated the particle background (K0) for each frame. K0 for each frame was calculated by locating the cell in the video and analyzing the intensities of the cell background in each frame. K0 was chosen as the gray value at a given quantile of this distribution (80%). The intensity value for each particle is then given by the difference between particle intensity and the background in each frame. To minimize photon fluctuations within a given frame, we considered the particle intensity as the average value (background subtracted) obtained over the first 20 frames. To ensure that no photobleaching events occurred within this range of frames that would affect the quantitation of the data, we measured the photobleaching times (s) from individual CD86-AcGFP (electroporated in Jurkat CD4 + cells, *Dorsch et al., 2009*) particles that exclusively showed a single photobleaching step. Fitting of the distribution to a single exponential decay renders a $\tau 0$ value of ~5 s, which corresponds to 50 frames (100 ms/frame). The total number of receptors/particles was finally estimated by dividing the average particle intensity by the particle intensity arising from individual AcGFP molecules. To unequivocally identify the intensity emitted by an individual AcGFP, we used as a calibrator Jurkat CD4$^+$ cells electroporated with the monomeric CD86-AcGFP. We performed SPT experiments on CD86-AcGFP and analyzed the data in similar way as described above. Distribution of monomeric particle intensities was analyzed by Gaussian fitting, rendering a mean value of 980 ± 89 a.u. This value was then used as the monomer reference to estimate CXCR4-AcGFP particle size.

## Internalization and flow cytometry studies

Cells ($5\times10^5$ cells/well) were activated with CXCL12 (50 nM) at the indicated time points at 37 °C. After incubation, cells were washed twice with staining buffer and receptor internalization was determined by flow cytometry using an anti-CXCR4 mAb (clone 44717, 30 min, 4 °C), followed by secondary staining with PE-coupled goat anti-mouse IgG (30 min, 4 °C). Results are expressed as a percentage of the mean fluorescence intensity of treated cells relative to that of untreated cells. When required, cells were pretreated with the small compounds (50 µM, 30 min, 37 °C) or AMD3100 (1 µM, 30 min, 37 °C) before activation with CXCL12.

## Western blotting

Cells ($3 \times 10^6$) were activated with CXCL12 (50 nM) at the time points indicated and then lysed in RIPA detergent buffer supplemented with 1 mM PMSF, 10 µg/ml aprotinin, 10 µg/ml leupeptin and 10 µM sodium orthovanadate, for 30 min at 4 °C. Extracts were analyzed by western blotting using specific antibodies. Densitometric evaluation of blots was performed using ImageJ (NIH, Bethesda, MD). When required, cells were pretreated with the small compounds (50 µM, 30 min, 37 °C) or AMD3100 (1 µM, 30 min, 37 °C) before activation with CXCL12.

## cAMP determination

cAMP levels were determined using the cAMP-Glo Max Assay (Promega, Madison, WI, in cells $5 \times 10^5$ cells/well) untreated or pre-treated with CXCL12 (50 nM, 5 min, 37 °C) followed by the addition of forskolin (10 µM, 10 min, 37 °C). When required, cells were pretreated with the small compounds (50 µM, 30 min, 37 °C) or AMD3100 (1 µM, 30 min, 37 °C) before activation with CXCL12.

## CXCL12 binding

Cells ($5 \times 10^5$ cells/well) were incubated with CXCL12 ATTO-700 (*Ameti et al., 2018*), (30 min, 37 °C) and maintained at 4 °C before analyzing the bound fluorescence in a Beckman Coulter Cytoflex flow cytometer. Results are expressed as mean fluorescence intensity. When required, cells were pretreated with the small compounds (50 µM, 30 min, 37 °C) or AMD3100 (1 µM, 30 min, 37 °C) before activation with CXCL12.

## Fluorescence resonance transfer analysis by sensitized emission

HEK-293T cells transiently transfected at a fixed CXCR4-YFP:CXCR4-CFP ratio (15 µg and 9 µg, respectively) were treated or not with the compounds (50 µM, 30 min, 37 °C) or their vehicle (DMSO), and FRET efficiency was evaluated (n=3, mean ± SEM, $p \leq 0.001$). Emission light was quantified using the Wallac Envision 2104 Multilabel Reader (Perkin Elmer, Foster City, CA) equipped with a high-energy xenon flash lamp (donor: receptor fused to C-CFP, 8 nm bandwidth excitation filter at 405 nm; acceptor:receptor fused to YFP, 10 nm bandwidth excitation filter at 510 nm).

## Actin polymerization

Cells ($5 \times 10^5$ cells/well) were incubated with CXCL12 (50 nM, 37 °C) or anti-CD3 (1 µg/ml, 37 °C) at the indicated time periods and then fixed with 2% paraformaldehyde and transferred to ice for 10 min. Fixed cells were permeabilized with 0.01% saponin (10 min, 4 °C) and labeled with Phalloidin-TRITC (Merck, 30 min, 4 °C). After washing, bound fluorescence was analyzed by flow cytometry on a Beckman Coulter FC500 flow cytometer. When required, cells were pretreated with the small compounds (50 µM, 30 min, 37 °C) or AMD3100 (1 µM, 30 min, 37 °C) before activation with CXCL12.

## Immunofluorescence analyses

Cells on fibronectin (20 µg/ml, Sigma)-coated glass slides were stimulated or not with 100 nM CXCL12 (5 min at 37 °C), fixed with 4% paraformaldehyde (10 min), permeabilized with 0.25% saponin (10 min), and stained with phalloidin-TRITC (Sigma-Merck; 30 min), all at room temperature (RT). Preparations were analyzed using a Leica TCS SP8 confocal multispectral microscope. When required, cells were pretreated with the small compounds (50 µM, 30 min, 37 °C) or AMD3100 (1 µM, 30 min, 37 °C) before activation with CXCL12.

## Cell adhesion/migration on planar lipid bilayers

Planar lipid bilayers were prepared as reported (*Carrasco et al., 2004*). Briefly, unlabeled GPI-linked intercellular adhesion molecule 1 (ICAM-1) liposomes were mixed with 1,2-dioleoyl-phosphatidylcoline. Membranes were assembled in FCS2 chambers (Bioptechs, Butler, PA), blocked with PBS containing 2% FCS for 1 hr at RT, and coated with CXCL12 (200 nM, 30 min, RT). Cells ($3 \times 10^6$ cells/ml) in PBS containing 0.5% FCS, 0.5 g/l D-glucose, 2 mM $MgCl_2$ and 0.5 mM $CaCl_2$ were then injected into the pre-warmed chamber (37 °C). Confocal fluorescence, differential interference contrast (DIC) and interference reflection microscopy (IRM) images were acquired on a Zeiss Axiovert LSM 510-META inverted microscope with a 40×oil-immersion objective. Imaris 7.0 software (Bitplane, Zurich, Switzerland) and ImageJ 1.49 v were used for qualitative and quantitative analysis of cell dynamics

parameters, fluorescence and IRM signals. The fluorescence signal of the planar bilayer in each case was established as the background fluorescence intensity. The frequency of adhesion (IRM+ cells) per image field was estimated as [n° of cells showing IRM contact/total n° of cells (estimated by DIC)]×100; similarly, we calculated the frequency of migration (cells showing and IRM contact and moving over time). When required, cells were pretreated with the small compounds (50 µM, 30 min, 37 °C) or AMD3100 (1 µM, 30 min, 37 °C) before activation with CXCL12.

## Directional cell migration

Pre-treated cells were diluted to $10×10^6$ cell/ml in RPMI medium containing 1% BSA and 20 mM HEPES (chemotaxis medium) and then seeded into the channel of a chemotaxis slide (µ-Slide Chemotaxis System, 80326; Ibidi, Munich, Germany) (1 hr, 37 °C, 5% $CO_2$). The reservoirs were then filled with chemotaxis medium containing 50 µM AGR1.137 or AGR1.131, or vehicle (DMSO) as a control condition, and 100 nM CXCL12 or CXCL2 was added to the right reservoir. Phase-contrast images were recorded over 20 hrs with a time lapse of 15 min using a Microfluor inverted microscope (Leica) with a 10×objective and equipped with an incubation system set to 5% $CO_2$ and 37 °C. Single-cell tracking was evaluated by selecting the center of mass in each frame using the manual tracking plug-in tool in ImageJ. Spider plots, representing the trajectories of the tracked cells, forward migration index (FMI), and straightness values, were obtained using the Chemotaxis and Migration Tool (ImageJ).

## Drug efficacy assay in zebrafish

HeLa cells were labeled with 8 µg/ml Fast-DiI oil (Cat #D3899; Thermo Fisher Scientific, Waltham, MA) in RPMI medium supplemented with 2% FBS for 30 min at 37 °C. Cells were then washed and filtered, and cell viability was determined using trypan blue-exclusion. Transgenic zebrafish Tg(fli1a:EGFP)y1 (*Lawson and Weinstein, 2002*) were bred naturally and maintained in E3 embryo medium (deionized water containing 0.5 mM NaCl, 0.4 mM $CaCl_2$, 0.7 mM MgSO4 and 0.2 mM KCl, pH 7.2) supplemented with 0.2 mM PTU (E3/PTU) at 28.5 °C. DiI-labeled HeLa cells were implanted into the dorsal perivitelline space of 2-day-old embryos, as described (*Rouhi et al., 2010*). Tumors were imaged within 2 hrs of implantation and tumor-baring embryos were treated with either vehicle (DMSO), AMD-3100 (1 and 10 µM) or with AGR1.131 or AGR1.137 (10 and 50 µM) for 3 days, followed by re-imaging. Changes in tumor size were evaluated as tumor area at day 3 divided by tumor area at day 0, and metastasis was evaluated as the number of cells disseminated to the caudal hematopoietic plexus.

## Statistical analyses

All data were analyzed with GraphPad Prism software version 9 (GraphPad Inc, San Diego, CA). Cell migration in Transwell assays and in planar lipid bilayers and directional cell migration assays and cell polarization under the various conditions were analyzed to determine significant differences between means using one-way analysis of variance (ANOVA) followed by Tukey's multiple comparison test. A two-tailed Mann-Whitney non-parametric test was used to analyze the diffusion coefficient ($D_{1-4}$) of single particles. We used contingency tables to compare two or more groups of categorical variables, such as the percentages of mobile or immobile particles, and these were compared using a Chi-square test with a two-tailed p-value. Statistical differences were reported as n.s.=not significant $p > 0.05$, *$p \leq 0.05$, **$p \leq 0.01$, ***$p \leq 0.001$ and ****$p \leq 0.0001$.

## Acknowledgements

This work was supported by grants from the Spanish Ministry of Science and Innovation (PID2020-114980RB-I00) Agencia Estatal de Investigación/Fondo Europeo de Desarrollo Regional (AEI/FEDER), Unión Europea, and by a grant from the Regional Government of Madrid P2022/BMD-7274. CS is supported by MCIN /AEI /10.13039/501100011033/FEDER, UE (PID2022-140651NB-I00). NEC is supported by a grant of the Spanish Ministry of Economy, Industry and Competitivity (RTI2018-096100B-100). RJS is supported by FSE/FEDER through the Instituto de Salud Carlos III (ISCIII; CP20/00043). EG-C was supported by the program Apoyos Centros de Excelencia S.O. of the Spanish Ministry of Science and Innovation (SEV-2017-0712). PM and SG are included in the doctoral program of the Department of Molecular Biosciences, Universidad Autónoma de Madrid, and are supported by the Fondo de Personal Investigador (FPI) program of the Spanish Ministry for Science and Innovation

(BES2015-071302 and PRE2018-083201 respectively). RA-B is supported by the Garantía Juvenil program of the Regional Government of Madrid, Spain (CAM20_CNB_AI_07). We also acknowledge the technical help of the Advance Light Microscopy Unit at the CNB/CSIC. Compounds AGR1.31, AGR1.35 and AGR1.37 are included in the patent PCT/ES2022/070379.

## Additional information

### Competing interests

Eva M García-Cuesta, Ana Martinez, Nuria E Campillo, Jose Miguel Rodriguez Frade, César Santiago, Mario Mellado: Has a patent for compounds AGR1.131, AGR1.135 and AGR1.137 (PCT/ES2022/070376). The other authors declare that no competing interests exist.

### Funding

| Funder | Grant reference number | Author |
|---|---|---|
| Spanish National Plan for Scientific and Technical Research and Innovation | PID2020-114980RB-I00 | Eva M García-Cuesta<br>Pablo Martínez<br>Gianluca D'Agostino<br>Sofia Gardeta<br>Adriana Quijada-Freire<br>Blanca Soler Palacios<br>Pilar Lucas<br>Rosa Ayala-Bueno<br>Noelia Santander Acerete<br>Jose Miguel Rodriguez Frade<br>Mario Mellado |
| Comunidad de Madrid | P2022/BMD-7274 | Eva M García-Cuesta<br>Pablo Martínez<br>Gianluca D'Agostino<br>Sofia Gardeta<br>Adriana Quijada-Freire<br>Blanca Soler Palacios<br>Pilar Lucas<br>Rosa Ayala-Bueno<br>Noelia Santander Acerete<br>Jose Miguel Rodriguez Frade<br>Mario Mellado |
| Spanish National Plan for Scientific and Technical Research and Innovation | PID2022-140651NB-I00 | César Santiago |
| Instituto de Salud Carlos III | CP20/00043 | Rodrigo Jiménez-Saiz |
| Ministerio de Ciencia e Innovación | SEV-2017-0712 | Eva M García-Cuesta<br>Adriana Quijada-Freire |
| Spanish National Plan for Scientific and Technical Research and Innovation | BES2015-071302 | Pablo Martínez |
| Spanish National Plan for Scientific and Technical Research and Innovation | PRE2018-083201 | Sofia Gardeta |
| Comunidad de Madrid | CAM20_CNB_AI_07 | Rosa Ayala-Bueno |

The funders had no role in study design, data collection and interpretation, or the decision to submit the work for publication.

### Author contributions

Eva M García-Cuesta, Formal analysis, Investigation, Methodology, Writing – original draft, Writing – review and editing; Pablo Martínez, Karthik Selvaraju, Gabriel Ulltjärn, Gianluca D'Agostino, Sofia Gardeta, Adriana Quijada-Freire, Investigation; Adrián Miguel Gómez Pozo, Patricia Blanco Gabella,

Carlos Roca, Daniel del Hoyo, Blanca Soler Palacios, Pilar Lucas, Rosa Ayala-Bueno, Noelia Santander Acerete, Methodology; Rodrigo Jiménez-Saiz, Provided input into the project; Alfonso García-Rubia, Ana Martinez, Resources; Yolanda Carrasco, Provided input into the project; Carlos Oscar Sorzano, Resources, Methodology; Nuria E Campillo, Resources, Validation, Methodology; Lasse D Jensen, Investigation, Methodology; Jose Miguel Rodriguez Frade, Conceptualization, Data curation, Formal analysis, Supervision, Investigation, Methodology, Writing – original draft, Project administration, Writing – review and editing; César Santiago, Resources, Methodology, Writing – original draft, Writing – review and editing; Mario Mellado, Conceptualization, Supervision, Funding acquisition, Writing – original draft, Writing – review and editing

### Author ORCIDs
Rodrigo Jiménez-Saiz ⓘ https://orcid.org/0000-0002-0606-3251
Lasse D Jensen ⓘ https://orcid.org/0000-0003-2338-357X
Jose Miguel Rodriguez Frade ⓘ http://orcid.org/0000-0002-7753-1462
Mario Mellado ⓘ https://orcid.org/0000-0001-6325-1630

### Ethics
Human peripheral blood mononuclear cells isolated from buffy coats were obtained from the Centro de Transfusiones of the Madrid Community. The ethical approval was obtained fron the ethics committee of this institution.

Reviewer #2 (Public Review): https://doi.org/10.7554/eLife.93968.3.sa1
Author response https://doi.org/10.7554/eLife.93968.3.sa2

---

# Additional files

### Supplementary files
• MDAR checklist

### Data availability
All data generated or analysed during this study are included in the manuscript, figures, figure supplements and source data files.

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
