## [Editor Report · eLife assessment]

This is an **important** study that describes an elegant modelling driven approach to design of allosteric antagonists for CXCR4 that have a selective effect on receptor nanocluster formation, cell polarisation and chemotaxis, but spare binding of CXCL12 to the receptor and inhibition of adenylate cyclase. This enables selective targeting of processes dependent upon cell polarisation and chemotaxis without impacting signalling effects and may avoid some of the toxicity associated with antagonists that target CXCL12 binding and thus block all CXCR4 signalling. The revised manuscript offers **convincing** evidence to support the claims. The modelling work is better described and additional data has been presented that better illustrates the unique features of the new antagonist. The in vivo studies in the zebrafish model open a path to studies in mammalian models.

---

## [Referee Report · Reviewer #2 (Public Review)]

Summary:

This work describes a new pharmacological targeting approach to inhibit selective functions of the ubiquitously expressed chemokine receptor CXCR4, a potential target of immunomodulatory or anti-cancer treatments. Overall, the results build a strong case for the potential of this new compound to target specific functions of CXCR4, particularly linked to tumorigenesis. However, a more thorough evaluation of the function of the compound as well as future studies in mammalian model systems are needed to better assess the promise of the compound.

Strengths:

The work elegantly utilizes in silico drug modelling to propose new small molecule compounds with specific features. This way, the authors designed compound AGR1.137, which abolishes ligand-induced CXCR4 receptor nanoclustering and the subsequent directed cell migration without affecting ligand-binding itself or some other ligand-induced signaling pathways. The authors have used a relatively broad set of experiments to validate and demonstrate the effects of the drug. Importantly, the authors also test AGR1.137 in vivo, using a zebra fish model of tumorigenesis and metastasis. A relatively strong inhibitory effect of the compound is reported.

Weaknesses:

The authors have been able to significantly strengthen their data from the first submission. The content of this manuscript is pretty solid, although studies in mammalian model systems are naturally needed in the future to better assess the promise of the compound.

---

## [Author Response]

The following is the authors’ response to the original reviews.

**Reviewer #1**
(1) In the "Introduction" section, an important aspect that requires attention pertains to the discussion surrounding the heterodimerization of CXCR4 and CCR5. Notably, the manuscript overlooks a recent study (https://doi.org/10.1038/s41467-023-42082-z) elucidating the mechanism underlying the formation of functional dimers within these G protein-coupled receptors (GPCRs)…The inclusion of this study within the manuscript would significantly enrich the contextual framework of the work, offering readers a comprehensive understanding of the current knowledge surrounding the structural dynamics and functional implications of CXCR4 and CCR5 heterodimerization.

We thank the reviewer for his/her recommendation to enrich the contextual framework of our study. The Nature Communications paper by Di Marino et al. was published after we sent the first version of our manuscript to eLife, and therefore was not included in the discussion. As the reviewer rightly indicates, this paper elucidates the mechanism underlying the formation of functional dimers within CCR5 and CXCR4. Using metadynamics approaches, the authors emphasize the importance of distinct transmembrane regions for dimerization of the two receptors. In particular, CXCR4 shows two low energy dimer structures and the TMVI-TMVII helices are the preferred interfaces involved in the protomer interactions in both cases. Although the study uses in silico techniques, it also includes the molecular binding mechanism of CCR5 and CXCR4 in the membrane environment, as the authors generate a model in which the receptors are immersed in a 1-palmitoyl-2-oleoyl-sn-glycero-3-phosphocholine (POPC) phospholipid bilayer with 10% cholesterol. This is an important point in this study, as membrane lipids also interact with membrane proteins, and the lipid composition affects CXCR4 oligomerization (Gardeta S.R. et al. Front. Immunol. 2023). In particular, Di Marino et al. find a cholesterol molecule placed in-between the two CXCR4 protomers where it engages a series of hydrophobic interactions with residues including Leu132, Val214, Leu216 and Phe249. Then, the polar head of cholesterol forms an H-bond with Tyr135 that further stabilizes protomer binding. In our hands, the F249L mutation in CXCR4 reverted the antagonism of AGR1.137, suggesting that the compound binds, among others, this residue. We should, nonetheless, indicate that we analyzed receptor oligomerization and not CXCR4 dimerization, which was the main object of the Di Marino et al. study. It is therefore also plausible that other residues than those described as essential for CXCR4 dimerization might participate in receptor oligomerization. We can speculate that AGR1.137 might affect cholesterol binding to CXCR4 and, therefore, alter dimerization/oligomerization. Additionally, the CXCR4 x-ray structure with PDB code 3ODU (Wu B. et al. Science, 2010) experimentally shows the presence of two fatty acid molecules in contact with both TMV and TMVI. These molecules closely interact with hydrophobic residues in the protein, thereby stabilizing it in a hydrophobic environment. Although more experiments will be needed to clarify the mechanism involved, our results suggest that cholesterol and/or other lipids also play an important role in CXCR4 oligomerization and function, as seen for other GPCRs (Jakubik J. & ElFakahani E.E. Int J Mol Sci. 2021). However, we should also consider that other factors not included in the analysis by Di Marino et al. can also affect CXCR4 oligomerization; for instance, the co-expression of other chemokine receptors and/or other GPCRs that heterodimerize with CXCR4 might affect CXCR4 dynamics at the cell membrane, similar to other membrane proteins such as CD4, which also forms complexes with CXCR4 (Martinez-Muñoz L. et al. Mol. Cell 2018).

The revised discussion contains references to the study by Di Marino et al. to enrich the contextual framework of our data.

(2) In "various sections" of the manuscript, there appears to be confusion surrounding the terminology used to refer to antagonists. It is recommended to provide a clearer distinction between allosteric and orthosteric antagonists to enhance reader comprehension. An orthosteric antagonist typically binds to the same site as the endogenous ligand, directly blocking its interaction with the receptor. On the other hand, an allosteric antagonist binds to a site distinct from the orthosteric site, inducing a conformational change in the receptor that inhibits the binding of the endogenous ligand. By explicitly defining the terms "allosteric antagonist" and "orthosteric antagonist" within the manuscript, readers will be better equipped to discern the specific mechanisms discussed in the context of the study.

The behavior of the compounds described in our manuscript (AGR1.35 and AGR1.137) fits with the definition of allosteric antagonists, as they bind on a site distinct from the orthosteric site, although they only block some ligand-mediated functions and not others. This would mean that they are not formally antagonists and should be not considered as allosteric compounds, as their binding on CXCR4 does not alter CXCL12 binding, although they might affect its affinity. In this sense, our compounds respond much better to the concept of negative allosteric modulators (Gao Z.-G. & Jacobson K.A. Drug Discov. Today Technol. 2013). They act by binding on a site distinct from the orthosteric site and selectively block some downstream signaling pathways but not others induced by the same endogenous agonist.

To avoid confusion and to clarify the role of the compounds described in this study, we now refer to them as negative allosteric modulators along the manuscript.

(3) In the Results section, the computational approach employed for "screening small compounds targeting CXCR4, particularly focusing on the inhibition of CXCL12-induced CXCR4 nanoclustering", requires clarification due to several points of incomprehension. The following recommendations aim to address these concerns and enhance the overall clarity of the section:(1) Computational Approach and Binding Mode Description:-Explicitly describe the methodology for identifying the pocket/clef area in angstroms (Å) on the CXCR4 protein structure. Include details on how the volume of the cleft enclosed by TMV and TMVI was determined, as this information is not readily apparent in the provided reference (https://doi.org/10.1073/pnas.1601278113).

The identification of the cleft was based on the observations by Wu et al. (Wu B. et al. Science 2010) who described the presence of bound lipids in the area formed by TMV and VI, and those of Wescott et al. (Wescott M.P. et al. Proc. Natl. Acad. Sci. 2016) on the importance of TMVI in the transmission of conformational changes promoted by CXCL12 on CXCR4 towards the cytoplasmic surface of the receptor to link the binding site with signaling activation. Collectively, these results, and our previous data on the critical role of the N-terminus region of TMVI for CXCR4 oligomerization (Martinez-Muñoz L. et al. Mol. Cell 2018), focused our in silico screening to this region. Once we detected that several compounds bound CXCR4 in this region, the cleavage properties were calculated by subtracting the compound structure. The resulting PDB was analyzed using the PDBsum server (Laskowski R.A. et. al. Protein Sci. 2018). Volume calculations were obtained using the server analyzing surface clefts by SURFNET (Laskowski R. A. J. Mol. Graph. 1995). The theoretical interaction surface between the selected compounds and CXCR4 and the atomic distances between the protein residues and the compounds was calculated using the PISA server (Krissinel E. & Henrick K. J. Mol. Biol. 2007) (Fig. I, only for review purposes). The analysis of the cleft occupied by AGR1.135 showed two independent cavities of 434 Å3 and 1,381 Å3 that were not connected to the orthosteric site. In the case of AGR1.137, the data revealed two distinct clefts of 790 Å3 and 580 Å3 (Fig. I, only for review purposes). These details have been included in the revised manuscript (New Fig. 1A, Supplementary Fig 8A, B).

(4) Clarify the statement regarding the cleft being "surface exposed for interactions with the plasma membrane," particularly in the context of its embedding within the membrane.

For GPCRs, transmembrane domains represent binding sites for bioactive lipids that play important functional and physiological roles (Huwiler A. & Zangemeister-Wittke U. Pharmacol. Ther. 2018). The channel between TMV and TMVI connects the orthosteric chemokine binding pocket to the lipid bilayer and is occupied by an oleic acid molecule, according to the CXCR4 structure published in 2010 (Wu B. et al. Science 2010). In addition, the target region contains residues involved in cholesterol (and perhaps other lipids) engagement (Di Marino et al. Nat. Commun. 2023). Taken together, these data support our statement that the cleft supports interactions between CXCR4 molecules and the plasma membrane.

Moreover, the data of Di Marino et al. also support that CCR5 and CXCR4 have a symmetric and an asymmetric binding mode. Therefore, either dimeric structure has the possibility to form trimers, tetramers, and even oligomers by using the free binding interface to complex with another protomer. This hypothesis suggests that the interaction of dimers to form oligomers should involve residues distinct from those included in the dimeric conformation.

The sentence has been modified in the revised manuscript to clarify comprehension.

(5) Discuss the rationale behind targeting the allosteric binding pocket instead of the orthosteric pocket, outlining potential advantages and disadvantages.

The advantages and disadvantages of using negative allosteric modulators vs orthosteric antagonists have been now included in the revised discussion.

The majority of GPCR-targeted drugs function by binding to the orthosteric site of the receptor, and are agonists, partial agonists, antagonists or inverse agonists. These orthosteric compounds can have off-target effects and poor selectivity due to highly homologous receptor orthosteric sites and to abrogation of spatial and/or temporal endogenous signaling patterns.

The alternative is to use allosteric modulators, which can tune the functions associated with the receptors without affecting the orthosteric site. They can be positive, negative or neutral modulators, depending on their effect on the functionality of the receptor (Foster D.J. & Conn P.J. Neuron 2017). For example, the use of a negative allosteric modulator of a chemokine receptor to dampen pathological signaling events, while retaining full signaling for non-pathological activities might limit adverse effects (Kohout T.A.et al. J. Biol. Chem. 2004). In this case, the negative allosteric modulator 873140 blocks CCL3 binding on CCR5 but does not alter CCL5 binding (Watson C. et al. Mol. Pharmacol. 2005). In other cases, allosteric modulators can stabilize a particular receptor conformation and block others. The mechanism of action of the anti-HIV-1, FDAapproved, CCR5 allosteric modulator, maraviroc (Jin J. et al. Sci. Signal. 2018) is attributed to its ability to modulate CCR5 dimer populations and their subsequent subcellular trafficking and localization to the cell membrane (Jin J .et al. Sci. Signal. 2018). Two CCR5 dimeric conformations that are imperative for membrane localization were present in the absence of maraviroc; however, an additional CCR5 dimer conformation was discovered after the addition of maraviroc, and all homodimeric conformations were further stabilized. This finding is consistent with the observation that CCR5 dimers and oligomers inhibit HIV host-cell entry, likely by preventing the HIV-1 co-receptor formation.

It is well known that GPCRs activate G proteins, but they also recruit additional proteins (e.g., β-arrestins) that induce signaling cascades which, in turn, can direct specific subsets of cellular responses independent of G protein activation (Eichel K. et al. Nature 2018) and are responsible for either therapeutic or adverse effects. Allosteric modulators can thus be used to block these adverse effects without influencing the therapeutic benefits. This was the case in the design of G protein-biased agonists for the kappa opioid receptor, which maintain the desirable antinociceptive and antipruritic effects and eliminate the sedative and dissociative effects in rodent models (Brust T.F. et al. Sci. Signal 2016).

(6) Provide the PDB ID of the CXCR4 structure used as a template for modeling with SwissModel. Explain the decision to model the structure from the amino acid sequence and suggest an alternative approach, such as utilizing AlphaFold structures and performing classical molecular dynamics with subsequent clustering for the best representative structure.

The PDB used as a template for modeling CXCR4 was 3ODU. This information was already included in the material and methods section. At the time we performed these analyses, there were several crystallographic structures of CXCR4 in complex with different molecules and peptides deposited at the PDB. None of them included a full construct containing the complete receptor sequence to provide a suitable sample for Xray structure resolution, as the N- and C-terminal ends of CXCR4 are very flexible loops. In addition, the CXCR4 constructs contained T4 lysozyme inserted between helices TMV and TMVI to increase the stability of the protein––a common strategy used to facilitate crystallogenesis of GPCRs (Zou Y. et al. PLoS One 2012). Therefore, we generated a CXCR4 homology model using the SWISS-MODEL server (Waterhouse A. et al. Nucleic Acids Res. 2018). This program reconstructed the loop between TMV and TMVI, a domain particularly important in this study that was not present in any of the crystal structure available in PDB. The model structure was, nonetheless, still incomplete, as it began at P27 and ended at S319 because the terminal ends were not resolved in the crystal structure used as a template. Nevertheless, we considered that these terminal ends were not involved in CXCR4 oligomerization.

As Alphafold was not available at the time we initiated this project, we didn’t use it. However, we have now updated our workflow to current methods and predicted the structure of the target using AlphaFold (Jumper J. et al. Nature 2021) and the sequence available under UniProt entry P61073. We prepared the ligands using OpenBabel (O’Boyle N.M. et al., J. Cheminformatics 2011), with a gasteiger charge assignment, and generated 10 conformers for each input ligand using the OpenBabel genetic algorithm. We then prepared the target structure with Openmm, removing all waters and possible heteroatoms, and adding all missing atoms. We next predicted the target binding pockets with fPocket (Le Guilloux V. et al. BMC Bioinformatics 2009), p2rank (Krivak R. & Hoksza, J. Cheminformatics 2018), and AutoDock autosite (Ravindranath P.A. & Sanner M.F. Bioinformatics 2016). We chose only those pockets between TMV and TMVI (see answer to point 3). We merged the results of the three programs into so-called consensus pockets, as two pockets are said to be sufficiently similar if at least 75% of their surfaces are shared (del Hoyo D. et al. J. Chem. Inform. Model. 2023). From the consensus pockets, there was one pocket that was significantly larger than the others and was therefore selected. We then docked the ligand conformers in this pocket using AutoDock GPU (Santos-Martins D. et al. J. Chem. Theory Comput. 2021), LeDock (Liu N & Xu Z., IOP Conf. Ser. Earth Environ. Sci. 2019), and Vina (Eberhardt J. et al. J. Chem. Inf. Model. 2021). The number of dockings varied from 210 to 287 poses. We scored each pose with the Vina score using ODDT (Wójcikowski M. et al. J. Cheminform. 2015). Then, we clustered the different solutions into groups whose maximum RMSD was 1Å. This resulted in 40 clusters, the representative of each cluster was the one with maximum Vina score and confirmed that the selected compounds bound this pocket (Author response image 1). When required, we calculated the binding affinity using Schrodinger’s MM-GBSA procedure (Greenidge P.A. et al. J. Chem. Inf. Model. 2013), in two ways: first, assuming that the ligand and target are fixed; second, with an energy minimization of all the atoms within a distance of 3Å from the ligand. This information has now been included in the revised version of the manuscript.

**Author response image 1. sa2fig1:** AGR1. 135 docking in CXCR4 using the updated protocol for ligand docking. Cartoon representation colored in gray with TMV and TMVI shown in blue and pink, respectively. AGR1.135 is shown in stick representation with carbons in yellow, oxygens in red and nitrogens in blue.

(7) Specify the meaning of "minimal interaction energy" and where (if present) the interaction scores are reported in the text.

We refer to minimal interaction energy, the best docking score, that is, the best score obtained in our docking studies. These data were not included in the previous manuscript due to space restrictions but are now included in the reviewed manuscript.

(8) You performed docking studies using GLIDE to identify potential binding sites for the small compounds on the CXCR4 protein. The top-scoring binders were then subjected to further refinement using PELE simulations. However, I realize that a detailed description of the specific binding modes of these compounds was not provided in the text. Please make the description of binding poses more detailed

Firstly, to assess the reliability of this method, a PELE study was carried out for the control molecule IT1t, which is a small drug-like isothiourea derivative that has been crystallized in complex with CXCR4 (PDB code: 3ODU). IT1t is a CXCR4 antagonist that binds to the CXCL12 binding cavity and inhibits HIV-1 infection (Das D. Antimicrob. Agents Chemother. 2015; Dekkers S. et al. J. Med. Chem. 2023). From the best five trajectories, two of them had clearly better binding energies, and corresponded to almost the same predicted pose of the molecule. Although the predicted binding mode was not exactly the same as the one in the crystal structure, the approximation was very good, giving validation to the approach. Although PELE is a suitable technique to find potential binding sites, the predicted poses must be subsequently refined using docking programs.

Analyzing the best trajectories for the remaining ligands, at least one of the best-scored poses was always located at the orthosteric binding site of CXCR4. Even though these poses showed good binding energies, they were discarded as the in vitro biological experiments indicated that the compounds were unable to block CXCL12 binding or CXCL12-mediated inhibition of cAMP release or CXCR4 internalization. Collectively, these data indicated that the selected compounds did not behave as orthosteric inhibitors of CXCR4. The CXCL12 binding pocket is the biggest cavity in CXCR4, and so PELE may tend to place the molecules near it. However, all the compounds presented other feasible binding sites with a comparable binding energy.

AGR1.135 and AGR1.137 showed interesting poses between TMV and TMVI with very good binding energy (-51.4 and -37.2 kcal/mol, respectively). This was precisely the region we had previously selected for the in silico screening, as previously described (see response to point 3).

AGR1.131 showed two poses with low binding energy that were placed between helices TMI and TMVII (-43.6 kcal/mol) and between helices TMV and TMVI (-39.8 kcal/mol). This compound was unable to affect CXCL12-mediated chemotaxis and was therefore used as an internal negative control as it was selected in the in silico screening with the same criteria as the other compounds but failed to alter any CXCL12-mediated functions. PELE studies nonetheless provided different binding sites for each molecule, which had to be further studied using docking to obtain a more accurate binding mode. In agreement with the previous commentary, we repeated the analysis using AlphaFold and the rest of the procedure described (see our response to point 6) and calculated the binding energies for all the compounds using Schrodinger’s MM-GBSA procedure (Greenidge P.A. et al. J. Chem. Inf. Model. 2013). Calculations were performed in two ways: first, assuming that the ligand and target are fixed; second, with an energy minimization of all the atoms within a distance of 3Å from the ligand. The results using the first method indicated that AGR1.135 and AGR1.137 showed poses between TMV and TMVI with - 56.4 and -62.4 kcal/mol, respectively and AGR1.131 had a pose between TMI and TMVII with -61.6kcal/mol. In the second method AGR1.135 and AGR1.137 showed poses between TMV and TMVI with -57.9, and -67.6 kcal/mol, respectively, and AGR1.131 of -62.2 kcal/mol between TMI and TMVII.

This information is now included in the text.

(9) (2) Experimental Design:-Justify the choice of treating Jurkat cells with a concentration of 50 μM of the selected compound. Consider exploring different concentrations and provide a rationale for the selected dosage. Additionally, clearly identify the type of small compound used in the initial experiment.

The revised version contains a new panel in Fig. 1B to show a more detailed kinetic analysis with different concentrations (1-100 µM) of the compounds in the Jurkat migration experiments. In all cases, 100 µM nearly completely abrogated cell migration, but in order to reduce the amount of DMSO added to the cells we selected 50 µM for further experiments, as it was the concentration that inhibits 50-75% of ligand-induced cell migration. Regarding the type of small compounds used in the initial experiments, they were compounds included in the library described in reference #24 (Sebastian-Pérez V. et al Med. Biol. Chem. 2017), which contains heterocyclic compounds. We would note that we do not consider AGR1.137 a final compound. We think that there is scope to develop AGR1.137-based second-generation compounds with greater solubility in water, greater specificity or affinity for CXCR4, and to evaluate delivery methods to hopefully increase activity.

(10) Avoid reporting details in rounded parentheses within the text; consider relocating such information to the Materials and Methods section or figure captions for improved readability.

Most of the rounded parentheses within the text have been eliminated in the revised version of the manuscript to improve readability.

(11) Elaborate on the virtual screening approach using GLIDE software, specifying the targeted site and methodology employed.

For the virtual screening, we used the Glide module (SP and XP function scoring) included in the Schrödinger software package, utilizing the corresponding 3D target structure and our MBC library (Sebastián-Pérez V et al. J. Chem. Inf. Model. 2017). The center of the catalytic pocket was selected as the centroid of the grid. In the grid generation, a scaling factor of 1.0 in van der Waals radius scaling and a partial charge cutoff of 0.25 were used. A rescoring of the SP poses of each compound was then performed with the XP scoring function of the Glide. The XP mode in Glide was used in the virtual screening, the ligand sampling was flexible, epik state penalties were added and an energy window of 2.5 kcal/mol was used for ring sampling. In the energy minimization step, the distance-dependent dielectric constant was 4.0 with a maximum number of minimization steps of 100,000. In the clustering, poses were considered as duplicates and discarded if both RMS deviation is less than 0.5 Å and maximum atomic displacement is less than 1.3 Å.

(12) Provide clarity on the statement that AGR1.131 "theoretically" binds the same motif, explaining the docking procedure used for this determination.

In the in silico screening, AGR1.131 was one of the 40 selected compounds that showed, according to the PELE analysis (see answer to point 8), a pose with low binding energy (-39.8 kcal/mol) between TMV and TMVI helices, which is the selected area for the screening. It, nonetheless, also showed a best pose placed between helices TM1 and TM7 (-43.7 kcal/mol) using the initial workflow. In conclusion, although AGR1.131 also faced to the TMV-TMVI, the most favorable pose was in the area between TMI and TMVII. In addition, the compound was included in the biological screening, where it did not affect CXCL12-mediated chemotaxis. We thus decided to use it as an internal negative control, as it has a skeleton very similar to AGR1.135 and AGR1.137 and can interact with the TM domains of CXCR4 without promoting biological effects. This statement has been clarified in the revised text.

(13) Toxicity Testing:-Enhance the explanation of the approach to testing the toxicity of the compound in Jurkat cells. Consider incorporating positive controls to strengthen the assessment and clarify the experimental design.

All the selected compounds in the in silico screening were initially tested for propidium iodide incorporation in treated cells in a toxicity assay, and some of them were discarded for further experiments (e.g., AGR1.103 and VSP3.1).

Further evaluation of Jurkat cell viability was determined by cell cycle analysis using propidium iodide. Supplementary Fig. 1B included the percentage of each cell cycle phase, and data indicated no significant differences between the treatments tested. Nevertheless, at the suggestion of the reviewer, and to clarify this issue, positive controls inducing Jurkat cell death (staurosporine and hydrogen peroxide) have also been included in the new Supplementary Fig. 2. The new figure also includes a table showing the percentage of cells in each cell-cycle phase.

(14) In the Results section concerning "AGR1.135 and AGR1.137 blocking CXCL12-mediated CXCR4 nanoclustering and dynamics", several points can be improved to enhance clarity and coherence: 1. Specificity of Low Molecular Weight Compounds:-Clearly articulate how AGR1.135 and AGR1.137 specifically target homodimeric CXCR4 and provide an explanation for their lack of impact on heterodimeric CXCR4-CCR5 in that region.

First of all, we should clarify that when we talk about receptor nanoclustering, oligomers refer to complexes including 3 or more receptors and, therefore, the residues involved in these interactions can differ from those involved in receptor dimerization. Moreover, our FRET experiments did not indicate that the compounds alter receptor dimerization (see new Supplementary Fig. 7). Of note, mutant receptors unable to oligomerize can still form dimers (Martínez-Muñoz L. et al. Mol. Cell 2018; García-Cuesta E.M .et al. Proc. Natl. Acad. Sci. USA 2022). Additionally, we believe that these oligomers can also include other chemokine receptors/proteins expressed at the cell membrane, which we are currently studying using different models and techniques.

We have results supporting the existence of CCR5/CXCR4 heterodimers (Martínez-Muñoz L et al. Proc. Natl. Acad. Sci. USA 2014), in line with the data published by Di Marino et al. However, in the current study we have not evaluated the impact of the selected compounds on other CXCR4 complexes distinct from CXCR4 oligomers. Our Jurkat cells do not express CCR5 and, therefore, we cannot discuss whether AGR1.137 affects CCR5/CXCR4 heterodimers. The chemokine field is very complex and most receptors can form dimers (homo- and heterodimers) as well as oligomers (Martinez-Muñoz L., et al Pharmacol & Therap. 2011) when co-expressed. To evaluate different receptor combinations in the same experiment is a complex task, as the number of potential combinations between distinct expressed receptors makes the analysis very difficult. We started with CXCR4 as a model, to continue later with other possible CXCR4 complexes. In addition, for the analysis of CCR5/CXCR4 dynamics, it is much better to use dual-TIRF techniques, which allow the simultaneous detection of two distinct molecules coupled to different fluorochromes.

Regarding the data of Di Marino et al., it is possible that the compounds might also affect heterodimeric conformations of CXCR4. This aspect has also been broached in the revised discussion. We would again note that we evaluated CXCR4 oligomers and not monomers or dimers; this is especially relevant when we compare the residues involved in these processes as they might differ depending on the receptor conformation considered. This issue was also hypothesized by Di Marino et al. (see our response to point 4).

(15) When referring to "unstimulated" cells, provide a more detailed explanation to elucidate the experimental conditions and cellular state under consideration.

Unstimulated cells refer to the cells in basal conditions, that is, cells in the absence of CXCL12. For TIRF-M experiments, transiently-transfected Jurkat cells were plated on glass-bottomed microwell dishes coated with fibronectin; these are the unstimulated cells. To observe the effect of the ligand, dishes were coated as above plus CXCL12 (stimulated cells). We have clarified this point in the material and methods section of the revised version.

(16) 2. Paragraph Organization-Reorganize the second paragraph to eliminate redundancy and improve overall flow. A more concise and fluid presentation will facilitate reader comprehension and engagement.

The second paragraph has been reorganized to improve overall flow.

(17) Ensure that each paragraph contributes distinct information, avoiding repetition and redundancy.

We have carefully revised each paragraph of the manuscript to avoid redundancy.

(18) 3. Claim of Allosteric Antagonism:-Exercise caution when asserting that "AGR1.135 and AGR1.137 behave as allosteric antagonists of CXCR4" based on the presented results. Consider rephrasing to reflect that the observed effects suggest the potential allosteric nature of these compounds, acknowledging the need for further investigations and evidence.

To avoid misinterpretations on the effect of the compounds on CXCR4, as we have commented in our response to point 2, we have substituted the term allosteric inhibitors with negative allosteric modulators, which refer to molecules that act by binding a site distinct from the orthosteric site, and selectively block some downstream signaling pathways, whereas others induced by the same endogenous or orthosteric agonist are unaffected (Gao Z.-G. & Jacobson K.A. Drug Discov. Today Technol. 2013). Our data indicate that the selected small compounds do not block ligand binding or G protein activation or receptor internalization, but inhibit receptor oligomerization and ligand-mediated directed cell migration.

(19) In the Results section discussing the "incomplete abolition of CXCR4-mediated responses in Jurkat cells by AGR1.135 and AGR1.137", several points can be refined for better clarity and completeness: 1. Inclusion of Positive Controls:-Consider incorporating positive controls in relevant experiments to provide a comparative benchmark for assessing the impact of AGR1.135 and AGR1.137. This addition will strengthen the interpretation of results and enhance the experimental rigor.

The in vivo experiments (Fig. 7E,F) used AMD3100, an orthosteric antagonist of CXCR4, as a positive control. We also included AMD3100, as a positive control of inhibition when evaluating the effect of the compounds on CXCL12 binding (Fig. 3, new Supplementary Fig. 3). The revised version of the manuscript also includes the effect of this inhibitor on other relevant CXCL12-mediated responses such as cell migration (Fig. 1B), receptor internalization (Fig. 3A), cAMP production (Fig. 3C), ERK1/2 and AKT phosphorylation (Supplementary Fig. 4), actin polymerization (Fig. 4A), cell polarization (Fig. 4B, C) and cell adhesion (Fig. 4D), to facilitate the interpretation of the results and improve the experimental rigor.

(20) 2. Clarification of Terminology:-Clarify the term "CXCR4 internalizes" by providing context, perhaps explaining the process of receptor internalization and its relevance to the study.

We refer to CXCR4 internalization as a CXCL12-mediated endocytosis process that results in reduction of CXCR4 levels on the cell surface. We use CXCR4 internalization in this study with two purposes: First, for CXCR4 and other chemokine receptors, internalization processes are mediated by ligand-induced clathrin vesicles (Venkatesan et al 2003) a process that triggers CXCR4 aggregation in these vesicles. We have previously determined that the oligomers of receptors detected by TIRF-M remain unaltered in cells treated with inhibitors of clathrin vesicle formation and of internalization processes (Martinez-Muñoz L. et al. Mol. Cell 2018). Moreover, we have described a mutant CXCR4 that cannot form oligomers but internalizes normally in response to CXCL12 (Martinez-Muñoz L. et al. Mol. Cell 2018). The observation in this manuscript of normal CXCL12-mediated endocytosis in the presence of the negative allosteric inhibitors of CXCR4 that abrogate receptor oligomerization reinforces the idea that the oligomers detected by TIRF are not related to receptor aggregates involved in endocytosis; Second, receptor internalization is not affected by the allosteric compounds, indicating that they downregulate some CXCL12-mediated signaling events but not others (new Fig. 3).

All these data have been included in the revised discussion of the manuscript.

(21) Elaborate on the meaning of "CXCL12 triggers normal CXCR4mut internalization" to enhance reader understanding.

We have previously described a triple-mutant CXCR4 (K239L/V242A/L246A; CXCR4mut). The mutant residues are located in the N-terminal region of TMVI, close to the cytoplasmic region, thus limiting the CXCR4 pocket described in this study (see our response to point 3). This mutant receptor dimerizes but neither oligomerizes in response to CXCL12 nor supports CXCL12-induced directed cell migration, although it can still trigger some Ca2+ flux and is internalized after ligand activation (Martinez-Muñoz L. et al. Mol. Cell 2018). We use the behavior of this mutant (CXCR4mut) to show that the CXCR4 oligomers and the complexes involved in internalization processes are not the same and to explain why we evaluated CXCR4 endocytosis in the presence of the negative allosteric modulators.

As we indicated in a previous answer to the reviewer, these issues have been re-elaborated in the revised version.

(22) 3. Discrepancy in CXCL12 Concentration:-Address the apparent discrepancy between the text stating, "...were stimulated with CXCL12 (50 nM, 37{degree sign}C)," and the figure caption (Fig. 3A) reporting a concentration of 12.5 nM. Rectify this inconsistency and provide an accurate and clear explanation.

We apologize for this error, which is now corrected in the revised manuscript. With the exception of the cell migration assays in Transwells, where the optimal concentration was established at 12.5 nM, in the remaining experiments the optimal concentration of CXCL12 employed was 50 nM. These concentrations were optimized in previous works of our laboratory using the same type of experiment. We should also remark that in the experiments using lipid bilayers or TIRF-M experiments, CXCL12 is used to coat the plates and therefore it is difficult to determine the real concentration of the ligand that is retained in the surface of the plates after the washing steps performed prior to adding the cells. In addition, we use 100 nM CXCL12 to create the gradient in the chambers used to perform the directed-cell migration experiments.

(23) 4. Speculation on CXCL12 Binding:-Refrain from making speculative statements, such as "These data suggest that none of the antagonists alters CXCL12 binding to CXCR4," unless there is concrete evidence presented up to that point. Clearly outline the results that support this conclusion.

Figure 3B and Supplementary Figure 3 show CXCL12-ATTO700 binding by flow cytometry in cells pretreated with the negative allosteric modulators. We have also included AMD3100, the orthosteric antagonist, as a control for inhibition. While these experiments showed no major effect of the compounds on CXCL12 binding, we cannot discard small changes in the affinity of the interaction between CXCL12 and CXCR4. In consequence we have re-written these statements.

(24) 5. Corroboration of Data:-Specify where the corroborating data from immunostaining and confocal analysis are reported, ensuring readers can access the relevant information to support the conclusions drawn in this section.

In agreement with the suggestion of the reviewer, the revised manuscript includes data from immunostaining and confocal analysis to complement Fig. 4B (new Fig. 4C). The revised version also includes some representative videos for the TIRF experiments showed in Figure 2 to clarify readability.

(25) In the Results section concerning "AGR1.135 and AGR1.137 antagonists and their direct binding to CXCR4", several aspects need clarification and refinement for a more comprehensive and understandable presentation: 1. Workflow Clarification:-Clearly articulate the workflow used for assessing the binding of AGR1.135 and AGR1.137 to CXCR4. Address the apparent contradiction between the inability to detect a direct interaction and the utilization of Glide for docking in the TMV-TMVI cleft.

To address the direct interaction of the compounds with CXCR4, we intentionally avoided the modification of the small compounds with different labels, which could affect their properties. We therefore attempted a fluorescence a spectroscopy strategy to formally prove the ability of the small compounds to bind CXCR4, but this failed because the AGR1.135 is yellow in color, which interfered with the determinations. We also tried a FRET strategy (see new Supplementary Fig. 7) and detected a significant increase in FRET efficiency of CXCR4 homodimers when AGR1.135 was evaluated, but again the yellow color interfered with FRET determinations. Moreover, AGR1.137 did not modify FRET efficiency of CXCR4 dimers. Therefore, we were unable to detect the interaction of the compounds with CXCR4.

We elected to develop an indirect strategy; in silico, we evaluated the binding-site using docking and molecular dynamics to predict the most promising CXCR4 binding residues involved in the interaction with the selected compounds. Next, we generated point mutant receptors of the predicted residues and re-evaluated the behavior of the allosteric antagonists in a CXCL12-induced cell migration experiment. Obviously, we first discarded those CXCR4 mutants that were not expressed on the cell membrane as well as those that were not functional when activated with CXCL12. Using this strategy, we eliminated the interference due to the physical properties of the compounds and demonstrated that if the antagonism of a compound is reversed in a particular CXCR4 mutant it is because the mutated residue participates or interferes with the interaction between CXCR4 and the compound, thus assuming (albeit indirectly) that the compound binds CXCR4.

To select the specific mutations included in the analysis, our strategy was to generate point mutations in residues present in the TMV-TMVI pocket of CXCR4 that were not directly proposed as critical residues involved in chemokine engagement, signal initiation, signal propagation, or G protein-binding, based on the extensive mutational study published by Wescott MP et. al. (Wescott M.P. et. al. Proc. Natl. Acad. Sci. U S A. 2016).

(26) Provide a cohesive explanation of the transition from docking evaluation to MD analysis, ensuring a transparent representation of the methodology.

Based on the aim of this work, the workflow shown in Author response image 2, was proposed to predict the binding mode of the selected molecules. Firstly, a CXCR4 model was generated to reconstruct some unresolved parts of the protein structure; then a binding site search using PELE software was performed to identify the most promising binding sites; subsequently, docking studies were performed to refine the binding mode of the molecules; and finally, molecular dynamics simulations were run to determine the most stable poses and predict the residues that we should mutate to test that the compounds interact with CXCR4.

**Author response image 2. sa2fig2:** Workflow followed to determine the binding mode of the studied compounds.

(27) 2. Choice of Software and Techniques:-Justify the use of "AMBER14" and the PELE approach, considering their potential obsolescence.

These experiments were performed five years ago when the project was initiated. As the reviewer indicates, AMBER14 and PELE approaches might perhaps be considered obsolescent. Thus, we have predicted the structure of the target using AlphaFold (Jumper J. et al, Nature 2021) and the sequence available under UniProt entry P61073. The complete analysis performed (see our response to point 4) confirmed that the compounds bound the selected pocket, as we had originally determined using PELE. These new analyses have been incorporated into the revised manuscript.

(28)-Discuss the role of the membrane in the receptor-ligand interac7on. Elaborate on how the lipidic double layer may influence the binding of small compounds to GPCRs embedded in the membrane.

Biological membranes are vital components of living organisms, providing a diffusion barrier that separates cells from the extracellular environment, and compartmentalizing specialized organelles within the cell. In order to maintain the diffusion barrier and to keep it electrochemically sealed, a close interaction of membrane proteins with the lipid bilayer is necessary. It is well known that this is important, as many membrane proteins undergo conformational changes that affect their transmembrane regions and that may regulate their activity, as seen with GPCRs (Daemen F.J. & Bonting S.L., Biophys. Struct. Mech. 1977; Gether U. et al. EMBO J. 1997). The lateral and rotational mobility of membrane lipids supports the sealing function while allowing for the structural rearrangement of membrane proteins, as they can adhere to the surface of integral membrane proteins and flexibly adjust to a changing microenvironment. In the case of the first atomistic structure of CXCR4 (Wu B. et al. Science 2010), it was indicated that for dimers, monomers interact only at the extracellular side of helices V and VI, leaving at least a 4-Å gap between the intracellular regions, which is presumably filled by lipids. In particular, they indicated that the channel between TMV and TMVI that connects the orthosteric chemokine binding pocket to the lipid bilayer is occupied by an oleic acid molecule. Recently, Di Marino et al., analyzing the dimeric structure of CXCR4, found a cholesterol molecule placed in between the two protomers, where it engages a series of hydrophobic interactions with residues located in the area between TMI and TMVI (Leu132, Val214, Leu216, Leu246, and Phe249). The polar head of cholesterol forms an H-bond with Tyr135 that further stabilizes its binding mode. This finding confirms that cholesterol might play an important role in mediating and stabilizing receptor dimerization, as seen in other GPCRs (Pluhackova, K., et al. PLoS Comput. Biol. 2016). In addition, we have previously observed that, independently of the structural changes on CXCR4 triggered by lipids, the local lipid environment also regulates CXCR4 organization, dynamics and function at the cell membrane and modulates chemokine-triggered directed cell migration. Prolonged treatment of T cells with bacterial sphingomyelinase promoted the complete and sustained breakdown of sphingomyelins and the accumulation of the corresponding ceramides, which altered both membrane fluidity and CXCR4 nanoclustering and dynamics. Under these conditions, CXCR4 retained some CXCL12-mediated signaling activity but failed to promote efficient directed cell migration (Gardeta S.R. et al. Front. Immunol. 2022). Collectively, these data demonstrate the key role that lipids play in the stabilization of CXCR4 conformations and in regulating its lateral mobility, influencing their associated functions. These considerations have been included in the revised version of the manuscript.

(29) 3. Stable Trajectories and Binding Mode Superimposi7on -Specify the criteria for defining "stable trajectories" to enhance reader understanding

There could be several ways to describe the stability of a MD simulation, based on the convergence of energies, distances or ligand-target interactions, among others. In this work, we use the expression “stable trajectories” to refer to simulations in which the ligand trajectory converges and the ligand RMSD does not fluctuate more than 0.25Å. This definition is now included in the revised text.

(30) Clarify the meaning behind superimposing the two small compounds and ensure that the statement in the figure caption aligns with the information presented in the main text.

We apologize for the error in the previous Fig. 5A and in its legend. The figure was created by superimposing the protein component of the poses for the two compounds, AGR1.135 and AGR1.137, rather than the compounds themselves. As panel 5A was confusing, we have modified all Fig. 5 in the revised manuscript to improve clarity.

(31) 4. Volume Analysis and Distances:-Provide details on how the volume analysis was computed and how distances were accounted for. Consider adding a figure to illustrate these analyses, aiding reader comprehension.

The cleft search and analysis were performed using the default settings of SURFNET (Laskowski R.A. J. Mol. Graph. 1995) included in the PDBsum server (Laskowski R.A. et. al. Trends Biochem. Sci. 1997). The first run of the input model for CXCR4 3ODU identified a promising cleft of 870 Å3 in the lower half of the region flanked by TMV and TMVI, highlighting this area as a possible small molecule binding site (Fig. I, only for review purposes). Analysis of the cleft occupied by AGR1.135 showed two independent cavities of 434 Å3 and 1381 Å3 that were not connected to the orthosteric site. The same procedure for AGR1.137 revealed two distinct clefts of 790 Å3 and 580 Å3, respectively (Fig. I, only for review purposes). Analysis of the atomic distances between the protein residues and the compounds was performed using the PISA server. (Krissinel E. & Henrick K. J. Mol. Biol. 2007). (Please see our response to point 3 and the corresponding figure).

(32) 5. Mutant Selection and Relevance:-Clarify the rationale behind selecting the CXCR4 mutants used in the study. Consider justifying the choice and exploring the possibility of performing an alanine (ALA) scan for a more comprehensive mutational analysis.

The selection of the residues to be mutated along the cleft was first based on their presence in the proposed cleft and the direct interaction of the compounds with them, either by hydrogen bonding or by hydrophobic interactions. Secondly, all mutated residues did not belong to any of the critical residues involved in transmitting the signal generated by the interaction of CXCL12 with the receptor. In any case, mutants producing a non-functional CXCR4 at the cell membrane were discarded after FACS analysis and chemotaxis experiments. Finally, the length and nature of the resulting mutations were designed mainly to occlude the cleft in case of the introduction of long residues such as lysines (I204K, L208K) or to alter hydrophobic interactions by changing the carbon side chain composition of the residues in the cleft. Indeed, we agree that the alanine scan mutation analysis would have been an alternative strategy to evaluate the residues involved in the interactions of the compounds.

(33) Reevaluate the statement regarding the relevance of the Y256F muta7on for the binding of AGR1.137. If there is a significant impact on migra7on in the mutant (Fig. 6B), elaborate on the significance in the context of AGR1.137 binding.

In the revised discussion we provide more detail on the relevance of Y256F mutation for the binding of AGR1.137 as well as for the partial effect of G207I and R235L mutations. The predicted interactions for each compound are depicted in new Fig. 6 C, D after LigPlot+ analysis (Laskowski R.A. & Swindells M.B. J. Chem. Inf. Model. 2011), showing that AGR1.135 interacted directly with the receptor through a hydrogen bond with Y256. When this residue was mutated to F, one of the anchor points for the compound was lost, weakening the potential interaction in the region of the upper anchor point.

It is not clear how the Y256F mutation will affect the binding of AGR1.137, but other potential contacts cannot be ruled out since that portion of the compound is identical in both AGR1.135 and AGR1.137. This is especially true for its neighboring residues in the alpha helix, F249, L208, as shown in 3ODU structure (Fig. 6D), which are shown to be directly implicated in the interaction of both compounds. Alternatively, we cannot discard that Y256 interacts with other TMs or lipids stabilizing the overall structure, which could reverse the effect of the mutant at a later stage (Author response image 3).

**Author response image 3. sa2fig3:** Cartoon representation of Y256 and its intramolecular interactions in the CXCR4 Xray solved structure 3ODU. TMV helix is colored in blue and TMVI in pink.

(34) Address the apparent discrepancy in residue involvement between AGR1.135 and AGR1.137, particularly if they share the same binding mode in the same clef.

AGR1.135 and AGR1.137 exhibit comparable yet distinct binding modes, engaging with CXCR4 within a molecular cavity formed by TMV and TMVI. AGR1.135 binds to CXCR4 through three hydrogen bonds, two on the apical side of the compound that interact with residues TMV-G207 and TMVI-Y256 and one on the basal side that interacts with TMVI-R235 (Fig. 5A). This results in a more extended and rigid conformation when sharing hydrogen bonds, with both TMs occupying a surface area of 400 Å2 and a length of 20 Å in the cleft between TMV and TMVI (Supplementary Fig. 8A). AGR1.137 exhibits a distinct binding profile, interacting with a more internal region of the receptor. This interaction involves the formation of a hydrogen bond with TMIIIV124, which induces a conformational shift in the TMVI helix towards an active conformation (Fig. 5B; Supplementary Fig. 13). Moreover, AGR1.137 may utilize the carboxyl group of V124 in TMIII and overlap with AGR1.135 binding in the cavity, interacting with the other 19 residues dispersed between TMV and VI to create an interaction surface of 370 Å2 along 20 Å (Supplementary Fig. 8B). This is illustrated in the new Fig. 5B. AGR1.137 lacks the phenyl ring present in AGR1.135, resulting in a shorter compound with greater difficulty in reaching the lower part of TMVI where R235 sits.

**Author response image 4. sa2fig4:** AGR1. 135 and AGR1.137 interaction with TMV and TMVI. The model shows the location of the compounds within the TMV-VI cleft, illustrated by a ribbon and stick representation. The CXCR4 segments of TMV and TMVI are represented in blue and pink ribbons respectively, and side chains for some of the residues defining the cavity are shown in sticks. AGR1.135 and AGR1.137 are shown in stick representation with carbon in yellow, nitrogen in blue, oxygen in red, and fluorine in green. Hydrogen bonds are indicated by dashed black lines, while hydrophobic interactions are shown in green. The figure reproduces the panels A, B of Fig. 5 in the revised manuscript.

(35) In the Results sec7on regarding "AGR1.137 treatment in a zebrafish xenograf model", the following points can be refined for clarity and completeness: 1. Cell Line Choice for Zebrafish Xenograft Model:-Explain the rationale behind the choice of HeLa cells for the zebrafish xenograft model when the previous experiments primarily focused on Jurkat cells. Address any specific biological or experimental considerations that influenced this decision.

As far as we know, there are no available models of tumors in zebrafish using Jurkat cells. We looked for a tumoral cell system that expresses CXCR4 and could be transplanted into zebrafish. HeLa cells are derived from a human cervical tumor, express a functional CXCR4, and have been previously used for tumorigenesis analyses in zebrafish (Brown H.K. et al. Expert Opin. Drug Discover. 2017; You Y. et al Front. Pharmacol. 2020). These cells grow in the fish and disseminate through the ventral area and can be used to determine primary tumor growth and metastasis. Nonetheless, we first analyzed in vitro the expression of a functional CXCR4 in these cells (Supplementary Fig. 10A), whether AGR1.137 treatment specifically abrogated CXCL12-mediated direct cell migration (Fig. 7A, B), as whether it affected cell proliferation (Supplementary Fig. 10B). As HeLa cells reproduce the in vitro effects detected for the compounds in Jurkat cells, we used this model in zebrafish. These issues were already discussed in the first version of our manuscript.

(36) 2. Toxicity Assessment in Zebrafish Embryos:-Clarify the basis for stating that AGR1.137 is not toxic to zebrafish embryos. Consider referencing the Zebrafish Embryo Acute Toxicity Test (ZFET) and provide relevant data on lethal concentration (LC50) and non-lethal toxic phenotypes such as pericardial edema, head and tail necrosis, malformation, brain hemorrhage, or yolk sac edema.

Tumor growth and metastasis kinetics within the zebrafish model have been extensively evaluated in many publications (White R. et al. Nat. Rev. Cancer. 2013; Astell K.R. and Sieger D. Cold Spring Harb. Perspect. Med. 2020; Chen X. et al. Front. Cell Dev. Biol. 2021; Weiss JM. Et al. eLife 2022; Lindhal G. et al NPJ Precis. Oncol. 2024). Our previous experience using this model shows that tumors start having a more pronounced proliferation and lower degree of apoptosis from day 4 onwards, but we cannot keep the tumor-baring larvae for that long due to ethical reasons and also because we don’t see much scientific benefit of unnecessarily extending the experiments. Anti-proliferative or pro-apoptotic effects of drugs can still be observed within the three days, even if this is then commonly seen as larger reduction (instead of a smaller growth as it is commonly seen in for example mouse tumor models) compared to controls. Initially we characterized the evolution of implanted tumors in our system and how much they metastasize over time in the absence of treatment before to test the compounds (Author response image 5).

The in vivo experiments were planned to validate efficacious concentrations of the investigated drugs rather than to derive in vivo IC50 or other values, which require testing of multiple doses. We have, however, included an additional concentration to show concentration-dependence and therefore on-target specificity of the drugs in the revised version of the manuscript (data also being elaborated in ongoing experiments). At this stage, we believe that adding the LC50 does not provide interesting new knowledge, and it is standard to only show results from the experimental endpoint (in our case 3 days post implantation). We agree that showing these new data points strengthens the manuscript and facilitates independent evaluation and conclusions to be drawn from the presented data. We have created new graphs where datapoints for each compound dose are shown.

**Author response image 5. sa2fig5:** Evolution of the tumors and metastasis along the time in the absence of any treatment. HeLa cells were labeled with 8 µg/mL Fast-DiI oil and then implanted in the dorsal perivitelline space of 2-days old zebrafish embryos. Tumors were imaged within 2 hours of implantation and re-imaged each 24 h for three days. Changes in tumor size was evaluated as tumor area at day 1, 2 and 3 divided by tumor area at day 0, and metastasis was evaluated as the number of cells disseminated to the caudal hematopoietic plexus at day 1, 2 and 3 divided by the number of cells at day 3.

Regarding the statement that AGR1.137 was not toxic, this was based on visual inspection of the zebrafish larvae at the end of the experiment, which also revealed a lack of drug-related mortality in these experiments. There are a number of differences in how our experiment was run compared with the standardized ZFET. ZFET evaluates toxicity from 0 hours post-fertilization to 1 or 2 days post-fertilization, whereas here we exposed zebrafish from 2 days post-fertilization to 5 days post-fertilization. The ZFET furthermore requires that the embryos are raised at 26ºC whereas kept the temperature as close as possible to a physiologically relevant temperature for the tumor cells (36ºC). In the ZFET, embryos are incubated in 96-well plates whereas for our studies we required larger wells to be able to manipulate the larvae and avoid well edge-related imaging artefacts, and we therefore used 24-well plates. As such, the ZFET was for various reasons not applicable to our experimental settings. As we were not interested in rigorously determining the LD50 or other toxicity-related measurements, as our focus was instead on efficacy and we found that the targeted dose was tolerated, we did not evaluate multiple doses, including lethal doses of the drug, and are therefore not able to determine an LD50/LC50. We also did not find drug-induced non-lethal toxic phenotypes in this study, and so we cannot elaborate further on such phenotypes other than to simply state that the drug is well tolerated at the given doses. Therefore, the reference to ZFET in the manuscript was eliminated.

(37) If supplementary information is available, consider providing it for a comprehensive understanding of toxicity assessments.

The effective concentration used in the zebrafish study was derived from the in vitro experiments. That being said, and as elaborated in our response to comment 36, we have added data for one additional dose to show the dose-dependent regulation of tumor growth and metastasis.

(38) 3. Optimization and Development of AGR1.137:-Justify the need for further optimization and development of AGR1.137 if it has a comparable effect to AMD3100. Explain the specific advantages or improvements that AGR1.137 may offer over AMD3100.

AGR1.137 is highly hydrophobic and is very difficult to handle, particularly in in vivo assays; thus, for the negative allosteric modulators to be used clinically, it would be very important to increase their solubility in water. Contrastingly, AMD3100 is a water-soluble compound. Before using the zebrafish model, we performed several experiments in mice using AGR1.137, but the inhibitory results were highly variable, probably due to its hydrophobicity. We also believe that it would be important to increase the affinity of AGR1.137 for CXCR4, as the use of lower concentrations of the negative allosteric modulator would limit potential in vivo side effects of the drug. On the other hand, we are also evaluating distinct administration alternatives, including encapsulation of the compounds in different vehicles. These alternatives may also require modifications of the compounds.

AMD3100 is an orthosteric inhibitor and therefore blocks all the signaling cascades triggered by CXCL12. For instance, we observed that AMD3100 treatment blocked CXCL12 binding, cAMP inhibition, calcium flux, cell adhesion and cell migration (Fig. 3, Fig. 4), whereas the effects of AGR1.137 were restricted to CXCL12-mediated directed cell migration. Although AMD3100 was well tolerated by healthy volunteers in a singledose study, it also promoted some mild and reversible events, including white blood cells count elevations and variations of urine calcium just beyond the reported normal range (Hendrix C.W. et al. Antimicrob. Agents Chemother. 2000). To treat viral infections, continuous daily dosing requirements of AMD3100 were impractical due to severe side effects including cardiac arrhythmias (De Clercq E. Front Immunol. 2015). For AMD3100 to be used clinically, it would be critical to control the timing of administration. In addition, side effects after long-term administration have potential problems. Shorter-term usage and lower doses would be fundamental keys to its success in clinical use (Liu T.Y. et al. Exp. Hematol. Oncol. 2016). The use of a negative allosteric modulator that block cell migration but do not affect other signaling pathways triggered by CXCL12 would be, at least in theory, more specific and produce less side effects. These ideas have been incorporated into the revised discussion to reflect potential advantages or improvements that AGR1.137 may offer over AMD3100.

(39) 4. Discrepancy in AGR1.137 and AMD3100 Effects:-Discuss the observed discrepancy where AGR1.137 exhibits similar effects to AMD3100 but only after 48 hours. Provide insights into the temporal dynamics of their actions and potential implications for the experimental design.

Images and data shown in Fig. 7E, F correspond to days 0 and 3 after HeLa cell implantation (tumorigenesis) and only to day 3 in the case of metastasis data. The revised version contains the effect of two distinct doses of the compounds (10 and 50 µM, for AGR1.135 and AGR1.137 and 1 and 10 µM for AMD3100).

(40) In the "Discussion" section, there are several points that require clarifica7on and refinement to enhance the overall coherence and depth of the analysis: 1. Reduction of Side-Effects:-Provide a more detailed explanation of how the identified compounds, specifically AGR1.135 and AGR1.137, contribute to the reduction of side effects. Consider discussing specific mechanisms or characteristics that differentiate these compounds from existing antagonists.

The sentence indicating that AGR1.135 and AGR1.137 contribute to reduce side effects is entirely speculative, as we have no experimental evidence to support it. We have therefore corrected this in the revised version. The origin of the sentence was that orthosteric antagonists typically bind to the same site as the endogenous ligand, thus blocking its interaction with the receptor. Therefore, orthosteric inhibitors (i.e. AMD3100) block all signaling cascades triggered by the ligand and therefore their functional consequences. However, the compounds described in this project are essentially negative allosteric modulators, that is, they bind to a site distinct from the orthosteric site, inducing a conformational change in the receptor that does not alter the binding of the endogenous ligand, and therefore block some specific receptor-associated functions without altering others. We observed that AGR1.137 blocked receptor oligomerization and directed cell migration whereas CXCL12 still bound CXCR4, triggered calcium mobilization, did not inhibit cAMP release or promoted receptor internalization. This is why we speculated on the limitation of side effects. The statements have been nonetheless revised in the new version of the manuscript.

(41) 2. Binding Site Clarification:-Address the apparent discrepancy between docking the small compounds in a narrow cleft formed by TMV and TMVI helices and the statement that AGR1.131 binds elsewhere. Clarify the rationale behind this assertion

After the in silico screening, a total of 40 compounds were selected. These compounds showed distinct degrees of interaction with the cleft formed by TMV and TMVI and even with other potential interaction sites on CXCR4, with the exception of the ligand binding site according to the data described by Wescott et al. (PNAS 2016 113:9928-9933), as this possibility was discarded in the initial approach of the in silico screening. According to PELE analysis, AGR1.131 was one of the 40 selected compounds that showed a pose with low binding energy, -39.8 kcal/mol, between TMV and TMVI helices, that is, it might interact with CXCR4 through the selected area for the screening. It nonetheless also showed a best pose placed between helices TMI and TMVII, -43.7 kcal/mol. In any case, the compound was included in the biological screening, where it was unable to impact CXCL12-mediated chemotaxis (Fig. 1B). We then focused on AGR1.135 and AGR1.137, as showed a higher inhibitory effect on CXCL12-mediated migration, and on AGR1.131 as an internal negative control. AGR1.131 has a skeleton very similar to the other compounds (Fig. 1C) and can interact with the TM domains of CXCR4 without promoting effects. None of the three compounds affected CXCL12 binding, or CXCL12mediated inhibition of cAMP release, or receptor internalization. However, whereas AGR1.135 and AGR1.137, blocked CXCL12-mediated CXCR4 oligomerization and directed cell migration towards CXCL12 gradients, AGR1.131 had no effect in these experiments (Fig. 3, Fig. 4).

Next, we performed additional theoretical calculations (PELE, docking, MD) to inspect in detail the potential binding modes of active and inactive molecules. Based on these additional calculations, we identified that whereas AGR1.135 and AGR1.137 showed preferent binding on the molecular pocket between TMV and TMVI, the best pose for AGR1.131 was located between TMI and TMVII, as the initial experiments indicated. These observations and data have been clarified in the revised discussion.

(42) 3. Impact of Chemical Modifications:-Discuss the consequences of the distinct chemical groups in AGR1.135, AGR1.137, and AGR1.131, specifically addressing how variations in amine length and chemical nature may influence binding affinity and biological activity. Provide insights into the potential effects of these modifications on cellular responses and the observed outcomes in zebrafish.

The main difference between AGR1.131 and the other two compounds is the higher flexibility of AGR1.131 due to the additional CH2 linker, together with the lack of a piperazine ring. The additional CH2 linking the phenyl ring increases the flexibility of AGR1.131 when compared with AGR1.135 and AGR1.137, and the absence of the piperazine ring might be responsible for its lack of activity, as it makes this compound able to bind to CXCR4 (Fig. 1C).

AGR1.137 was chosen in a second round. The additional presence of the tertiary amine (in the piperazine ring) allows the formation of quaternary ammonium salts in the aqueous medium and its substituents to increase its solubility (Fig 1C). This characteristic might be related to the absence of toxic effects of the compound in the zebrafish model.

(43) 4. Existence of Distinct CXCR4 Conformational States:-Provide more detailed support for the statement suggesting the "existence of distinct CXCR4 conformational states" responsible for activating different signaling pathways. Consider referencing relevant studies or experiments that support this claim.

Classical models of GPCR allostery and activation, which describe an equilibrium between a single inactive and a single signaling-competent active conformation, cannot account for the complex pharmacology of these receptors. The emerging view is that GPCRs are highly dynamic proteins, and ligands with varying pharmacological properties differentially modulate the balance between multiple conformations.

Just as a single photograph from one angle cannot capture all aspects of an object in movement, no one biophysical method can visualize all aspects of GPCR activation. In general, there is a tradeoff between high-resolution information on the entire protein versus dynamic information on limited regions. In the former category, crystal and cryo-electron microscopy (cryoEM) structures have provided comprehensive, atomic-resolution snapshots of scores of GPCRs both in inactive and active conformations, revealing conserved conformational changes associated with activation. However, different GPCRs vary considerably in the magnitude and nature of the conformational changes in the orthosteric ligand-binding site following agonist binding (Venkatakrishnan A.J.V. et al. Nature 2016). Spectroscopic and computational approaches provide complementary information, highlighting the role of conformational dynamics in GPCR activation (Latorraca N.R.V. et al. Chem. Rev 2017). In the absence of agonists, the receptor population is typically dominated by conformations closely related to those observed in inactive-state crystal structures (Manglik A. et al. Cell 2015). While agonist binding drives the receptor population towards conformations similar to those in activestate structures, a mixture of inactive and active conformations remains, reflecting “loose” or incomplete allosteric coupling between the orthosteric and transducer pockets (Dror R.O. et al. Proc. Natl. Acad. Sci. USA 2011). Surprisingly, for some GPCRs, and under some experimental conditions, a substantial fraction of unliganded receptors already reside in an active-like conformation, which may be related to their level of basal or constitutive signaling (Staus D.P. et al. J. Biol. Chem. 2019; Ye L. et al. Nature 2016). In our case, the negative allosteric modulators, (Staus DP, et al. J. Biol. Chem 2019; Ye L. et al. Nature 2016) did not alter ligand binding and had only minor effects on specific CXCL12-mediated functions such as inhibition of cAMP release or receptor internalization, among others, but failed to regulate CXCL12-mediated actin dynamics and receptor oligomerization. Collectively, these data suggest that the described compounds alter the active conformation of CXCR4 and therefore support the presence of distinct receptor conformations that explain a partial activation of the signaling cascade.

All these observations are now included in the revised discussion of the manuscript.

(44) 5. Equilibrium Shift and Allosteric Ligands:-Clarify the statement about "allosteric ligands shifting the equilibrium to favor a particular receptor conformation". Support this suggestion with references or experimental evidence

In a previous answer (see our response to point 2), we explain why we define the compounds as negative allosteric modulators. These compounds do not bind the orthosteric binding site or a site distinct from the orthosteric site that alters the ligand-binding site. Their effect should be due to changes in the active conformation of CXCR4, which allow some signaling events whereas others are blocked. Our functional data thus support that through the same receptor the compounds separate distinct receptor-mediated signaling cascades, that is, our data suggest that CXCR4 has a conformational heterogeneity. It is known that GPCRs exhibit more than one “inactive” and “active” conformation, and the endogenous agonists stabilize a mixture of multiple conformations. Biased ligands or allosteric modulators can achieve their distinctive signaling profiles by modulating this distribution of receptor conformations. (Wingler L.M. & Lefkowitz R.J. Trends Cell Biol. 2020). For instance, some analogs of angiotensin II do not appreciably activate Gq signaling (e.g., increases in IP3 and Ca2+) but still induce receptor phosphorylation, internalization, and mitogen-activated protein kinase (MAPK) signaling (Wei H, et al. Proc. Natl. Acad. Sci. USA 2003). Some of these ligands activate Gi and G12 in bioluminescence resonance energy transfer (BRET) experiments (Namkung Y. et al. Sci. Signal. 2018). A similar observation was described in the case of CCR5, where some chemokine analogs promoted G protein subtype-specific signaling bias (Lorenzen E. et al. Sci. Signal 2018). Structural analysis of distinct GPCRs in the presence of different ligands vary considerably in the magnitude and nature of the conformational changes in the orthosteric ligand-binding site following agonist binding (Venkatakrishnan A.J.V. et al. Nature 2016). Yet, these changes modify conserved motifs in the interior of the receptor core and induce common conformational changes in the intracellular site involved in signal transduction. That is, these modifications might be considered distinct receptor conformations.

The revised discussion contains some of these interpretations to support our statement about the stabilization of a particular receptor conformation triggered by the negative allosteric modulators.

(45) 6. Refinement of Binding Mode:-Clarify the workflow for obtaining the binding mode, particularly the role of GLIDE and PELE. Clearly explain how these software tools were used in tandem to refine the binding mode.

The computational sequential workflow applied in this project included, (i) Protein model construction, (ii) Virtual screening (Glide), (iii) PELE, (iv) Docking (AutoDock and Glide) and (v) Molecular Dynamics (AMBER).

Glide was applied for the structure-based virtual screening to explore which compounds could fit and interact with the previously selected binding site.

After the identification of theoretically active compounds (modulators of CXCR4), additional calculations were done to identify a potential binding site. PELE was used in this sense, to study how the compounds could bind in the whole surface of the target (TMV-TMVI). By applying PELE, we avoided biasing the calculation, and we found that the trajectories with better interaction energies identified the cleft between TMV and TMVI as the binding site for AGR1.135 and AGR1.137, and not for AGR1.131. AGR1.131 showed a pose with low binding energy, -39.8 kcal/mol, between TMV and TMVI helices, that is, it might interact with CXCR4 in the selected area for the screening. But it also showed a better pose placed between helices TMI and TMVII, - 43.7 kcal/mol (see our response to point 41). These data have been now confirmed using Schrodinger’s MM-GBSA procedure (see our response to points 6 and 8). In any case, the compound was included in the biological screening, where it was unable to affect CXCL12-mediated chemotaxis (Fig. 1B). Docking and MD simulations were then performed to study and refine the specific binding mode in this cavity. These data were important to choose the mutations on CXCR4 required, to test whether the compounds reversed its behavior. In these experiments we also confirmed that AGR1.131 had a better pose on the TMI-TMVII region.

(46) 7. Impact of Compound Differences on CXCR4-F249L mutant:-Provide visual aids, such as figures, and additional experiments to support the statement about differences in the behavior of AGR1.135 and AGR1.137 on cells expressing CXCR4-F249L mutant. Elaborate on the closer interaction suggested between the triazole group of AGR1.137 and the F249 residue

At the reviewer’s suggestion, Fig. 5 has been modified to incorporate a closer view of the interactions identified and new panels in new Fig. 6 have been added to show in detail the effect of the mutations selected on the structure of the cleft between TMV and TMVI. The main difference between AGR1.135 and AGR1.137 is how the triazole group interacts with F249 and L216 (Author response image 6). In AGR1.137, the three groups are aligned in a parallel organization, which appears to be more effective: This might be due to a better adaptation of this compound to the cleft since there is only one hydrogen bond with V124. In AGR1.135, the compound interacts with the phenyl ring of F249 and has a stronger interaction at the apical edge to stabilize its position in the cleft. However, there is still an additional interaction present. When changing F249

**Author response image 6. sa2fig6:** Cartoon representation of the interaction of CXCR4 F249L mutant with AGR1. 135 (A) and AGR1.137 (B). The two most probable conformations of Leucine rotamers are represented in cyan A and B conformations. Van der Waals interactions are depicted in blue cyan dashed lines, hydrogen bonds in black dashed lines. CXCR4 segments of TMV and TMVI are colored in blue and pink, respectively

to L (Fig. VIIA, B, only for review purposes) and showing the two most likely rotamers resulting from the mutation, it is observed that rotamer B is in close proximity to the compound, which may cause the binding to either displace or adopt an alternative conformation that is easier to bind into the cleft. As previously mentioned, it is likely that AGR1.135 can displace the mutant rotamer and bind into the cleft more easily due to its higher affinity.

(47) In the "Materials and Methods" section, the computational approach for the "discovery of CXCR4 modulators" requires significant revision and clarification. The following suggestions aim to address the identified issues: 1. Structural Modeling:-Reconsider the use of SWISS-MODEL if there is an available PDB code for the entire CXCR4 structure. Clearly articulate the rationale for choosing one method over the other and explain any limitations associated with the selected approach.

The SWISS-model server allows for automated comparative modeling of 3D protein structures that was pioneered in the fields of automated modeling. At the time we started this project. it was the most accurate method to generate reliable 3D protein structure models.

As explained above, we have now predicted the structure of the target using AlphaFold (Jumper J. et al, Nature 2021) and performed several additional experiments that confirm that the small compounds bind the selected pocket as the original strategy indicated (see our response to point 6). (Fig. II, only for review purposes).

(48) 2. Parametriza7on of Small Compounds:-Provide a detailed description of the parametrization process for the small compounds used in the study. Specify the force field and parameters employed, considering the obsolescence of AMBER14 and ff14SB. Consider adopting more contemporary force fields and parameterization strategies.

When we performed these experiments, some years ago, the force fields applied (ff14SB, AMBER14 used in MD or OPLS2004 in docking with Glide) were well accepted and were gold standards. It is, however, true that the force fields have evolved in the past few years, Moreover, in the case of the MD simulations, to consider the parameters of the ligands that are not contained within the force field, we performed an additional parameterization as a standard methodology. We then generated an Ab initio optimization of the ligand geometry, defining as basis sets B3LYP 6-311+g(d), using Gaussian 09, Revision A.02, and then a single point energy calculation of ESP charges, with HF 6311+g(d) on the optimized structure. As the last step of the parametrization, the antechamber module was used to adapt these charges and additional parameters for MD simulations.

(49) 3. Treatment of Lipids and Membrane:-Elaborate on how lipids were treated in the system. Clearly describe whether a membrane was included in the simulations and provide details on its composition and structure. Address the role of the membrane in the study and its relevance to the interactions between CXCR4 and small compounds

To stabilize CXCR4 and more accurately reproduce the real environment in the MD simulation, the system was embedded in a lipid bilayer using the Membrane Builder tool (Sunhwan J. et al. Biophys. J. 2009) from the CHARMM-GUI server. The membrane was composed of 175 molecules of the fatty acid 1-palmitoyl-2-oleoyl-sn-glycero-3phosphocholine (POPC) in each leaflet. The protein-membrane complex was solvated with TIP3 water molecules. Chloride ions were added up to a concentration of 0.15 M in water, and sodium ions were added to neutralize the system. This information was previously described in detail.

(50) 4. Molecular Dynamics Protocol:-Provide a more detailed and coherent explanation of the molecular dynamics protocol. Clarify the specific steps, parameters, and conditions used in the simulations. Ensure that the protocol aligns with established best practices in the field.

Simulations were calculated on an Asus 1151 h170 LVX-GTX-980Ti workstation, with an Intel Core i7-6500 K Processor (12 M Cache, 3.40 GHz) and 16 GB DDR4 2133 MHz RAM, equipped with a Nvidia GeForce GTX 980Ti available for GPU (Graphics Processing Unit) computations. MD simulations were performed using AMBER14 (Case D.A. et al. AMBERT 14, Univ. of California, San Francisco, USA, 2014) with ff14SB (Maier J.A. et al. J. Chem. Theory Comput. 2015) and lipid14 (Dickson C. J. et al. J. Chem. Theory Comput. 2014) force fields in the NPT thermodynamic ensemble (constant pressure and temperature). Minimization was performed using 3500 Steepest Descent steps and 4500 Conjugate Gradient steps three times, firstly considering only hydrogens, next considering only water molecules and ions, and finally minimizing all atoms. Equilibration raises system temperature from 0 to 300 K at a constant volume fixing everything but ions and water molecules. After thermalization, several density equilibration phases were performed. In the production phase, 50 ns MD simulations without position restraints were calculated using a time step of 2 fs. Trajectories of the most interesting poses were extended to 150 ns. All bonds involving hydrogen atoms were constrained with the SHAKE algorithm (Lippert R.A. et al. J. Chem. Phys. 2007). A cutoff of 8 Å was used for the Lennard-Jones interaction and the short-range electrostatic interactions. Berendsen barostat (Berendsen H.J. et al. J. Chem. Phys. 1984) and Langevin thermostat were used to regulate the system pression and temperature, respectively. All trajectories were processed using CPPTRAJ (Roe D.R. & Cheatham III T.E. J. Chem. Theory Comput. 2013) and visualized with VMD (Visual Molecular Dynamics) (Humphrey W. et al. J. Mol. Graphics. 1996). To reduce the complexity of the data, Principal Component Analysis (PCA) was performed on the trajectories using CPPTRAJ.

(51) Consider updating the molecular dynamics protocol to incorporate more contemporary methodologies, considering advancements in simulation techniques and software.

In our answer to points 6 and 47, we describe why we use the technology based on Swiss-model and PELE analysis and how we have now used Alphafold and other more contemporary methodologies to confirm that the small compounds bind the selected pocket.

(52) Figure 1A:• Consider switching to a cavity representation for CXCL12 to enhance clarity and emphasize the cleft.

Fig. 1A has been modified to emphasize the cleft.

(53) Explicitly show the TMV-TMVI cleft in the figure for a more comprehensive visualization.

In Fig. 1A we have added an insert to facilitate TMV-TMVI visualization.

(54) Figure 1B:• Clearly explain the meaning of the second DMSO barplot to avoid confusion.

To clarify this panel, we have modified the figure and the figure legend. Panel B now includes a complete titration of the three compounds analyzed in the manuscript. The first bar shows cell migration in the absence of both treatment with AMD3100 and stimulation with CXCL12. The second bar shows migration in response to CXCL12 in the absence of AMD3100. The third bar shows the effect of AMD3100 on CXCL12-induced migration, as a known control of inhibition of migration. We hope that this new representation of the data results is clearer.

(55) Figure 1C:• Provide a clear legend explaining the significance of the green shading on the small compounds.

The legend for Fig. 1C has been modified accordingly to the reviewer’s suggestion.

(56) Figure 2:• Elaborate on the role of fibronectin in the experiment and explain the specific contribution of CD86-AcGFP.

The ideal situation for TIRF-M determinations is to employ cells on a physiological substrate complemented with or without chemokines. Fibronectin is a substrate widely used in different studies that allows cell adhesion, mimicking a physiological situation. Jurkat cells express alpha4beta1 and alpha5beta1 integrins that mediate adhesion to fibronectin (Seminario M.C. et al. J. Leuk. Biol. 1999).

Regarding the use of CD86-AcGFP in TIRF-M experiments. We currently determine the number of receptors in individual trajectories of CXCR4 using, as a reference, the MSI value of CD86-AcGFP that strictly showed a single photobleaching step (Dorsch S. et al. Nat Methods 2009).

We preferred to use CD86-AcGFP in cells instead of AcGFP on glass, to exclude any potential effect on the different photodynamics exhibited by AcGFP when bound directly to glass. In any case, this issue has been clarified in the revised version.

(57) Figure 3D:• Include a plot for the respective band intensity to enhance data presentation

The plot showing the band intensity analysis of the experiments shown in Fig. 3D was already included in the original version (see old Supplementary Fig. 3). However, in the revised version, we include these plots in the same figure as panels 3E and 3F. As a control of inhibition of CXCL12 stimulation, we have also included a new figure (Supplementary Fig. 4) showing the effect of AMD3100 on CXCL12-induced activation of Akt and ERK as analyzed by western blot.

(58) Consider adding AMD3100 as a control for comparison.

In agreement with the reviewer’s suggestion, we have added the effect of AMD3100 in most of the functional experiments performed.

(59) Figure 4:• Address the lack of positive controls in Figure 4 and consider their inclusion for a more comprehensive analysis.

DMSO bars correspond to the control of the experiment, as they represent the effect of CXCL12 in the absence of any allosteric modulator. As previously described in this point-by-point reply, DMSO bars correspond to the control performed with the solvent with which the small compounds, at maximum concentration, are diluted. Therefore, they show the effect of the solvent on CXCL12 responses. In any case, and in order to facilitate the comprehension of the figure we have also added the controls in the absence of DMSO to demonstrate that the solvent does not affect CXCL12-mediated functions, together with the effect of the orthosteric inhibitor AMD3100. In addition, we have also included representative images of the effect of the different compounds on CXCL12-induced polarization (Fig. 4C).

(60) In Figure 4A, carefully assess overlapping error bars and ensure accurate interpreta7on. If necessary, consider alternative representation.

We have tried alternative representations of data in Fig. 4A, but in all cases the figure was unclear. We believe that the way we represent the data in the original manuscript is the most clear and appropriate. Nevertheless, we have now included significance values as a table annexed to the figure, as well as the effect of AMD3100, as a control of inhibition

(61) Supplementary Figure 1A:• Improve the clarity of bar plots for better understanding. Consider reordering them from the most significant to the least.

This was a good idea, and therefore Supplementary Fig. 1A has been reorganized to improve clarity.

(62) Supplementary Figure 1C:• Clarify the rationale behind choosing the 12.5 nM concentration and explain if different concentrations of CXCL12 were tested.

In old Supplementary Fig. 1C, we used untreated cells, that is, CXCL12 was not present in the assay. These experiments were performed to test the potential toxicity of DMSO (solvent) or the negative allosteric modulators on Jurkat cells. The 12.5 nM concentration of CXCL12 mentioned in the figure legend applied only to panels A and B, as indicated in the figure legend. We previously optimized this concentration for Jurkat cells using different concentrations of CXCL12 between 5 and 100 nM. Nevertheless, we have reorganized old supplementary fig. 1 and clarified the figure legend to avoid misinterpretations (see Supplementary Fig 1A, B and Supplementary Fig. 2A, B).

(63) Explain the observed reduction in fluorescence intensity for AGR1.135.

The cell cycle analysis has been moved from Supplementary Fig. 1C to a new Supplementary Fig. 2. It now includes the flow cytometry panels to show fluorescence intensity as a function of the number of cells analyzed (Panel 1A) as well as a table (panel B) with the percentage of cells in each phase of the cell cycle. We believe that the apparent reduction in fluorescence that the reviewer observes is mainly due to the number of events analyzed. However, we have changed the flow cytometry panels for others that are more representative and included a table with the mean of the different results. When we determined the percentage of cells in each cell cycle phase, we observed that it looks very similar in all the experimental conditions. That is, none of the compounds affected any of the cell cycle phases. We have also included the effect of H2O2 and staurosporine as control compounds inducing cell death and cell cycle alteration of Jurkat cells.

(64) Supplementary Table 1:• Include a column specifying the scoring for each compound to provide a clear reference for readers.

To facilitate references to readers, we have now included the inhibitory effect of each compound on Jurkat cell migration in the revised version of this table.

(65) Minor PointsPage 2 - Abstract: Rephrase the first sentence of the abstract to enhance fluidity.

Although the entire manuscript was revised by a professional English editor, we appreciate the valuable comments of this reviewer and we have corrected these issues accordingly.

(66) Page 2 - Abstract: Explicitly define "CXCR4" as "C-X-C chemokine receptor type 4" the first time it appears.

We have not used C-X-C chemokine receptor type 4 the first time it appears in the abstract. CXCR4 is an acronym normally accepted to identify this chemokine receptor, and it is used as CXCR4 in many articles published in eLife. However, we introduce the complete name the first time it appears in the introduction.

(67) Page 2 - Abstract: Explicitly define "CXCL12" as "C-X-C motif chemokine 12" the first time it is mentioned.

As we have discussed in the previous response, we have not used C-X-C motif chemokine 12 the first time CXCL12 appears in the abstract, as it is a general acronym normally accepted to identify this specific chemokine, even in eLife papers. However, we introduce the complete name the first time it appears in the introduction section.

(68) Page 2 - Abstract: Explicitly define "TMV and TMVI" upon its first mention.

The acronym TM has been defined as “Transmembrane” in the revised version

(69) Page 2 - Abstract: Review the use of "in silico" in the sentence for accuracy and consider revising if necessary.

With the term “in silico” we want to refer to those experiments performed on a computer or via computer simulation software. We have carefully reviewed its use in the new version of the manuscript.

(70) Page 2 - Abstract: Add a comma after "compound" in the sentence, "We identified AGR1.137, a small compound that abolishes...".

A comma after “compound” has been added in the revised sentence.

(71) Page 2 - Significance Statement: Rephrase the first sentence of the "Significance Statement" to avoid duplication with the abstract.

The first sentence of the Significance Statement has been revised to avoid duplication with the abstract.

(72) Page 2 - Significance Statement: Break down the lengthy sentence, "Here, we performed in silico analyses..." for better readability.

The sentence starting by “Here, we performed in silico analyses…” has been broken down in the revised manuscript.

(73) Page 2 - Introduction: Replace "Murine studies" with a more specific term for clarity.

The term “murine studies” is normally used to refer to experimental studies developed in mice. We have nonetheless rephrased the sentence.

(74) Page 3 - Introduction: Rephrase the sentence for clarity: "Finally, using a zebrafish model, ..."

The sentence has been now rephrased for clarity.

(75) Results-AGR1.135 and AGR1.137 block CXCL12-mediated CXCR4 nanoclustering and dynamics:Rephrase the sentence for clarity: "Retreatment with AGR1.135 and AGR1.137, but not with AGR1.131, substantially impaired CXCL12-mediated receptor nanoclustering.”

The sentence has been rephrased for clarity.

(76) Results - AGR1.135 and AGR1.137 incompletely abolish CXCR4-mediated responses in Jurkat cells: Clarify the sentence: "In contrast to the effect promoted by AMD3100, a binding-site antagonist of CXCR4..."

The sentence has been modified for clarity.

(77) Consider using "orthosteric" instead of "binding-site" antagonist.

The term orthosteric is now used throughout to refer to a binding site antagonist.

(78) Discussion: Use the term "in silico" only when necessary.

We have carefully reviewed the use of “in silico” in the manuscript.

(79) Discussion: Clarify the sentence: "...not affect neither CXCR2-mediated cell migration...". Confirm if "CXCL12" is intended.

The sentence refers to the chemokine receptor CXCR2, which binds the chemokine CXCL2. To test the specificity of the compounds for the CXCL12/CXCR4 axis, we evaluated CXCL2-mediated cell migration. The results indicated that CXCL2/CXCR2 axis was not affected by the negative allosteric modulators, whereas CXCL12-mediated cell migration was blocked. The sentence has been clarified in the new version of the manuscript.

(80) Figure 4B: Bold the "B" in the figure label for consistency.

The “B” in Fig. 4B has been bolded.

**Reviewer #2**
(1) Fig 2. The SPT data is sub-optimal in its presentation as well as analysis. Example images should be shown. The analysis and visualization of the data should be reconsidered for improvements. Graphs with several hundreds, in some conditions over 1000 tracks, per condition are very hard to compare. The same (randomly selected representative set) number of data points should be shown for better visualization. Also, more thorough analyses like MSD or autocorrelation functions are lacking - they would allow enhanced overall representation of the data.

In agreement with the reviewer’s commentary, we have modified the representation of Fig. 2. We have carefully read the paper published by Lord S.J. and col. (Lord S. J. et al., J. Cell Biol. 2020) and we apply their recommendations for these type of data. We have also included as supplementary material representative videos for the TIRF-M experiments performed to allow readers to visualize the original images. Regarding the MSD analyses, they were developed to determine all D1-4 values. According to the data published by Manzo & García-Parajo (Manzo C. & García-Parajo M.F. Rep.Prog. Phys. 2015) due to the finite trajectory length the MSD curve at large tlag has poor statistics and deviates from linearity. However, the estimation of the Diffusion Coefficient (D1-4) can be obtained by fitting of the short tlag region of the MSD plot giving a more accurate idea of the behavior of particles. In agreement we show D1-4 values and not MSD data.

Due to the space restrictions, it is very difficult to include all the figures generated, but, only for review purposes, we included in this point-by-point reply some representative plots of the MSD values as a function of the time from individual trajectories showing different types of motion obtained in our experiments (Author response image 7).

**Author response image 7. sa2fig7:** Representative MSD plots from individual trajectories of CXCR4-AcGFP showing different types of motion: (A) confined, (B) Brownian/Free, (C) direct transport of CXCR4-AcGFP particles diffusing at the cell membrane detected by SPT-TIRF in resting JKCD4 cells.

Further analysis, such as the classification based on particle motion, has not been included in this article. This classification uses the moment scaling spectrum (MSS), described by Ewers H. et al. 2005 PNAS, and requires particles with longer trajectories (>50 frames). Only for review purposes, we include a figure showing the percentage of the MSS-based particle motion classification for each condition. As expected, most of long particles are confined, with a slight increase in the percentage upon CXCL12 stimulation in all conditions, except in cell treated with AGR1.137 (Author response image 8).

**Author response image 8. sa2fig8:** Effects of the negative allosteric modulators on the Types of Motion of CXCR4. Percentage of single trajectories with different types of motion, classified by MSS (DMSO: 58 particles in 59 cells on FN; 314 in 63 cells on FN+CXCL12; AGR1.131: 102 particles in 71 cells on FN; 258in 69 cells on FN+CXCL12; AGR1.135: 86 particles in 70 cells on FN; 120 in 77 cells on FN+CXCL12; AGR1.137: 47 particles in 66 cells on FN; 74 in 64 cells on FN+CXCL12) n = 3.

(2) Fig 3. The figure legends have inadequate information on concentrations and incubation times used, both for the compounds and other treatments like CXCL12 and forskolin. For the Western blot data, also the quantification should be added to the main figure. The compounds, particularly AGR1.137 seem to lead to augmented stimulation of pAKT and pERK. This should be discussed

The Fig. 3 legend has been corrected in the revised manuscript. Fig. 3D now contains representative western blots and the densitometry evaluation of these experiments. As the reviewer indicates, we also detected in the western blot included, augmented stimulation of pAKT and pERK in cells treated with AGR1.137. However, as shown in the densitometry analysis, no significant differences were noted between the data obtained with each compound. As a control of inhibition of CXCL12 stimulation we have included a new Supplementary Fig. 4 showing the effect of AMD3100 on CXCL12-induced activation of Akt and ERK as analyzed by western blot.

(3) Fig. 4 immunofluorescence data on polarization as well as the flow chamber data lack the representative images of the data. The information on the source of the T cells is missing. Not clear if this experiment was done on bilayers or on static surfaces.

Representative images for the data shown in Figure 4B have been added in the revised figure (Fig. 4C). The experiments in Fig. 4B were performed on static surfaces. As indicated in the material and methods section, primary T cell blasts were added to fibronectin-coated glass slides and then were stimulated or not with CXCL12 (5 min at 37ºC) prior to fix permeabilize and stain them with Phalloidin. Primary T cell blasts were generated from PBMCs isolated from buffy coats that were activated in vitro with IL-2 and PHA as indicated in the material and methods section.

(4) The data largely lacks titration of different concentrations of the compounds. How were the effective concentration and treatment times determined? What happens at higher concentrations? It is important to show, for instance, if the CXCR12 binding gets inhibited at higher concentrations. most experiments were performed with 50 uM, but HeLa cell data with 100 uM. Why and how was this determined?

The revised version contains a new panel in Fig. 1B to show a more detailed kinetic analysis with different concentrations (1-100 µM) of the compounds in the migration experiments using Jurkat cells. We choose 50 µM for further studies as it was the concentration that inhibits 50-75% of the ligand induced cell migration.

We have also included the effect of two doses of the compounds (10 and 50 µM) in the zebrafish model as well as AMD3100 (1 and 10 µM) as control (new Fig. 7D, E). Tumors were imaged within 2 hours of implantation and tumor-baring embryos were treated with either vehicle (DMSO) alone, AGR1.131 or AGR1.137 at 10 and 50 µM or AMD3100 at 1 and 10 µM for three days, followed by re-imaging.

Regarding the amount of CXCL12 used in these experiments, with the exception of cell migration assays in Transwells, where the optimal concentration was established at 12.5 nM, in all the other experiments the optimal concentration of CXCL12 employed was 50 nM. In the case of the directional cell migration assays, we use 100 nM to create the chemokine gradient in the device. These concentrations have been optimized in previous works of our laboratory using these types of experiments. It should also be noted that in the experiments using lipid bilayers or TIRF-M experiments, CXCL12 is used to coat the plates and therefore it is difficult to determine the real concentration that is retained in the surface after the washing steps performed prior adding the cells.

(5) The authors state that they could not detect direct binding of the compounds and the CXCR14. It should be reported what approaches were tried and discussed why this was not possible.

We attempted a fluorescence spectroscopy strategy to formally prove the ability of AGR1.135 to bind CXCR4, but this strategy failed because the compound has a yellow color that interfered with the determinations. We also tried a FRET strategy (see supplementary Fig. 7) and detected a significant increase in FRET efficiency of CXCR4 homodimers in cells treated with AGR1.135; this effect was due to the yellow color of this compound that interferes with FRET determinations. In the same assays, AGR1.137 did not modify FRET efficiency for CXCR4 homodimers and therefore we cannot assume that AGR1.137 binds on CXCR4. All these data have been considered in the revised discussion.

(6) The proliferation data in Supplementary Figure 1 lacks controls that affect proliferation and indication of different cell cycle stages. What is the conclusion of this data? More information on the effects of the drug to cell viability would be important.

Toxicity in Jurkat cells was first determined by propidium iodide incorporation. Some compounds (i.e., AGR1.103 and VSP3.1) were discarded from further analysis as they were toxic for cells. In a deeper analysis of cell toxicity, even if these compounds did not kill the cells, we checked whether they could alter the cell cycle of the cells. New Supplementary Fig. 2 includes a table (panel B) with the percentage of cells in each cell cycle phase, and no differences between any of the treatments tested were detected.

Nevertheless, to clarify this issue the revised version of the figure also includes H2O2 and staurosporine stimuli to induce cell death and cell cycle alterations as controls of these assays.

(7) The flow data in Supplementary Figure 2 should be statistically analysed.

Bar graphs corresponding to the old Supplementary Fig. 2 (new Supplementary Fig. 3) are shown in Fig. 3B. We have also incorporated the corresponding statistical analysis to this figure.

(8) In general, the authors should revise the figure legends to ensure that critical details are added.

We have carefully revised all the figure legends in the new version of the manuscript.

(9) Bar plots are very poor in showing the heterogeneity of the data. Individual data points should be shown whenever feasible. Superplot-type of representation is strongly advised (https://doi.org/10.1083/jcb.202001064).

We have carefully read the paper published by Lord S.J. and col. (Lord S. J. et al., J. Cell Biol. 2020) and we apply their recommendations for our TIRF-M data (see revised Fig. 2).